# Tuning cyanide coordination electronic structure enables stable Prussian blue analogues for sodium-ion batteries

Yuanheng Wang[1], Jiaxin Yan[1], Bingxing Xie[2]✉, Yan Meng[1], Chuankai Fu[1], Fanpeng Kong[1], Xingyu Wang[1], Qingjie Zhou[1], Xin Chen[1], Jianting Li[1], Chunyu Du[1], Liguang Wang[3]✉ & Pengjian Zuo[1]✉

Prussian blue analogues exhibit significant potential as positive electrode materials for sodium-ion batteries, particularly due to their three-dimensional cyanide-bridged frameworks which facilitate fast charging capabilities. However, the labile chemical bonds coordinated by transition metal ions and cyanide ligands often lead to structural instability, causing serious electrochemical degradations during cycling. Fundamentally understanding and controlling the local electronic structure to mitigate this instability remains challenging. Herein, we approach this problem by modulating the local electronic structure surrounding nitrogen-coordinated transition metal ions to create a uniform electron distribution within the Prussian blue analogues frameworks. The resulting uniform electronic structure enhances the reactivity of both nitrogen-coordinated and carbon-coordinated transition metals. More importantly, the reduction of electronic displacement through regulated coordination significantly improves the crystal structural stability, yielding a capacity retention of over 91% at 5 C after 1000 cycles. These findings provide insights into the local structural chemistry of Prussian blue analogues and offer guidance for the development of positive materials for sodium-ion batteries.

Sodium-ion batteries (SIBs) have emerged as promising candidates for large-scale energy storage applications. Among the key components determining their performance, positive electrode materials play a pivotal role in governing the energy density, cost-effectiveness, and cycling stability of SIBs. Currently, three main categories of positive electrode materials are being extensively investigated: polyanion compounds[1,2], layered oxides[3,4], and Prussian blue analogs (PBAs). Notably, PBAs stand out as attractive positive electrode materials due to their open three-dimensional framework structure, which offers distinct advantages for sodium ion storage[5]. In the coordination framework structure of PBAs, the transition metal cations coordinated with nitrogen ($M^{HS}$) and Fe cations coordinated with carbon ($Fe^{LS}$) are connected by cyanide anions, forming perovskite-type coordination polymers with a formula $Na_xM^{HS}[Fe^{LS}(CN)_6]_y\square_{1-y}\cdot zH_2O$[6,7]. Up to now, the commercialization limitations for PBAs mainly stem from the unsatisfactory cycling performance and the discrepancy between the practical and theoretical capacity, which is mainly related to the poor stability of the coordination structure and incomplete activation of metal active sites[8].

Generally, the coordination environment of transition metal cations is affected by their valence and electron configuration as well as the electronegativity of ligand elements. The activity of $Fe^{LS}$ is difficult to fully activate according to the electronic configuration of $Fe^{LS}$–C[9]. The electron configurations of $Fe^{LS}$ have filled $t_{2g}$ orbitals,

[1]State Key Laboratory of Space Power-Sources, MIIT Key Laboratory of Critical Materials Technology for New Energy Conversion and Storage, School of Chemistry and Chemical Engineering, Harbin Institute of Technology, Harbin, China. [2]School of New Energy, Nanjing University of Science and Technology, Jiangyin, PR China. [3]College of Chemical and Biological Engineering, Zhejiang University, Hangzhou, China. ✉e-mail: bingxingxie@njust.edu.cn; wanglg@zju.edu.cn; zuopj@hit.edu.cn

making it difficult to be oxidized during the charging process[10]. In addition, the electron transfer in PBAs during cycling can also be affected by the cyanide electron cloud distribution[11]. The low electronegativity of C atoms and the spin state of $Fe^{LS}$ cations tend to form inner-orbital coordination structure, enhancing the interaction of $Fe^{LS}$ to the lone-pair electrons in C and hindering the electron transfer process along $Fe^{LS}$–C≡N–$M^{HS}$ coordination frameworks[12,13]. Moreover, the degradation of the crystal structure during cycling is mainly determined by the weak interaction between transition metals and cyanide ligands[14,15]. The $Fe^{LS}$–C≡N–$M^{HS}$ coordination structure tends to be disrupted during the desodiation/sodiation process because of the weak coordination interaction of $M^{HS}$ and N with low bond energy[16]. The difference in coordination bond energy between $Fe^{LS}$–C and $M^{HS}$–N bond is ascribed to the electron cloud distribution of cyanide ions[17]. The bond energy of inner-orbital $Fe^{LS}$–C is relatively stronger than that of outer-orbital $M^{HS}$–N coordination structure, and thus $M^{HS}$–N bonds prefer to be broken before the $Fe^{LS}$–C bonds destruction[18,19]. Moreover, some $M^{HS}$ cations (such as $Mn^{3+}$ and $Cu^{2+}$) with asymmetric 3d valence electron orbital configurations lead to the Jahn–Teller effect, triggering crystal structure degradation[20].

The electronic structure of $Fe^{LS}$–C coordination bonds can be changed by adjusting the electronic distribution of $M^{HS}$–N coordination bonds, thus the electron transfer energy barrier of $Fe^{LS}$ can be effectively reduced[21]. By selecting transition metals preferring to form stronger $M^{HS}$–N bonds, the stability of the cyanide coordination structure ($Fe^{LS}$–C≡N–$M^{HS}$) will be effectively improved[22]. The robust coordination structure of $Fe^{LS}$–C≡N–$M^{HS}$ can tolerate the unit cell volume changes and inhibit irreversible phase transition during cycling[23,24]. More importantly, the cyanide, as a crucial bridge-like function between $Fe^{LS}$ and $M^{HS}$, plays a vital role in determining the redox activity and structure stability of PBAs[25]. Although it has been proven that the π electron interaction between the cyanide anions and transition metal ions can alleviate the lattice volume change[26,27], the impact of cyanide coordination electronic structure on cycling stability and redox activity of PBAs has not been investigated systematically.

Here, we aim to balance reversible capacity and cycling performance by modulating the electronic structure of cyanide coordination frameworks for PBAs. The electronic distribution of $Fe^{LS}$–C≡N–$M^{HS}$ coordination structure is homogenized by optimizing the $M^{HS}$ ions, and the designed PBAs not only reduce the capacity loss caused by cyanide electron cloud displacement to $Fe^{LS}$, but also maintain a satisfactory stability of $Fe^{LS}$–C≡N–$M^{HS}$ coordination structure. The results of density functional theory (DFT) calculations reveal that the uniformly distributed cyanide electronic structure can activate $Fe^{LS}$ and $M^{HS}$ ions simultaneously, thus enhancing the reversible specific capacity of PBAs. In situ Fourier transform infrared spectroscopy (FT-IR) and ex situ extended X-ray absorption fine structure (EXAFS) are used to further confirm that the uniform cyanide electronic distribution between $Fe^{LS}$ and $M^{HS}$ in PBAs helps to maintain the stability of $Fe^{LS}$–C≡N–$M^{HS}$ coordination structure during the desodiation/sodiation process. Consequently, the optimized PBA achieves the simultaneous improvement of capacity and cycling lifetime, delivering the discharge capacity of $142.4\ mAh\cdot g^{-1}$ at 0.1 C and retaining 91.7% of the reversible capacity after 1000 cycles at 5 C.

## Results
### Material design and characterization
To demonstrate the effect of N-coordinated metals on the electronic structure of cyanide, a series of single N-coordinated-metal PBAs was modeled and structurally optimized by first-principles calculations. The electronic distribution maps of cubic (Fig. 1a–e) and

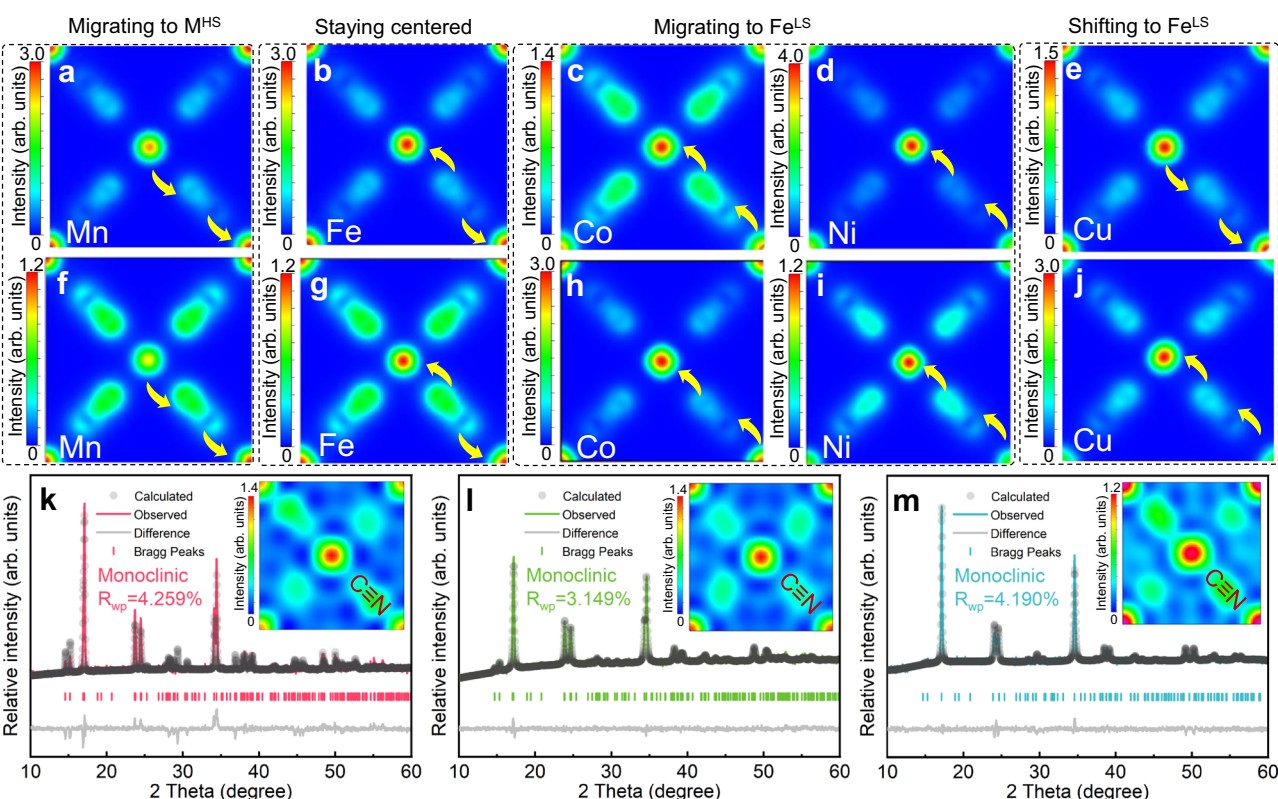

**Fig. 1 | Design of cyanide localized structure and crystal structure characterization.** Cross-sectional charge density distribution diagrams of cubic phase PBAs of various $M^{HS}$ calculated via DFT: **a** Mn-PBA; **b** Fe-PBA; **c** Co-PBA; **d** Ni-PBA; **e** Cu-PBA. Cross-sectional charge density distribution diagrams of rhombohedral phase PBAs: **f** Mn-PBA; **g** Fe-PBA; **h** Co-PBA; **i** Ni-PBA; **j** Cu-PBA. Rietveld refinement PXRD patterns with the schematic of the cross-sectional electronic structure shown in the inset: **k** M2-PBA; **l** M4-PBA; **m** M5-PBA.

rhombohedral phase PBAs (Fig. 1f–j) are obtained. The central position represents the electronic structure of $Fe^{LS}$, while the four corner positions correspond to the electronic structures of N-coordinated $Mn^{2+}$, $Fe^{2+}$, $Co^{2+}$, $Ni^{2+}$, and $Cu^{2+}$, respectively. The electronic structure of cyanide is located between $Fe^{LS}$ and $M^{HS}$. It can be found that the choice of $M^{HS}$ and the differences in phase structure affect the electron distribution of the cyanide coordination structure ($Fe^{LS}$–C≡N–$M^{HS}$). As the atomic number increases, the electron cloud of cyanide ions gradually shifts from being biased toward $M^{HS}$ to $Fe^{LS}$, which is observed from the decreased electronic distribution intensity at the center and the increased intensity at the corner positions (Fig. 1a–e). Furthermore, when comparing the sodium-rich rhombohedral phase and sodium-poor cubic phase of the same $M^{HS}$ PBAs, the cyanide electron displacement appears to be nearly identical (Fig. 1f–i). Differently, in the phase transition process of Cu-PBA from cubic (Fig. 1e) to rhombohedral (Fig. 1j), the cyanide electrons shift toward $Fe^{LS}$. Conventionally, as cyanide electrons become more concentrated toward the transition metal ions, the charge transfer capability of active sites is adversely affected[19], making redox reactions more challenging and consequently restricting the reversible capacity. Therefore, in order to give full play to the theoretical capacity of PBAs, it is crucial to obtain a uniform distribution of cyanide electrons, which can be achieved by regulating the transition metal elements.

To conceptually validate the electronic modulation of the $Fe^{LS}$–C≡N–$M^{HS}$ framework, three PBA samples with varying $M^{HS}$ compositions were synthesized via an optimized co-precipitation method. To systematically probe the influence of different $M^{HS}$ ions on the $Fe^{LS}$–C≡N–$M^{HS}$ electronic structure, the $M^{HS}$ components in each PBA were introduced at equimolar ratios during synthesis. In consideration of the selection number of $M^{HS}$, the samples are denoted as M2-PBA (in which the pre-set $M^{HS}$ are equimolar Mn and Fe), M4-PBA (in which the pre-set $M^{HS}$ are equimolar Mn, Fe, Co, and Ni), and M5-PBA (in which the pre-set $M^{HS}$ are equimolar Mn, Fe, Co, Ni, and Cu), respectively. The morphology of the as-prepared PBAs was characterized by SEM, exhibiting cubic morphology with particle sizes of approximately 1.5 μm (Fig. S1). The comparison of high-resolution transmission electron microscopy (TEM) images accompanied by energy dispersive spectroscopy (EDS) mapping is displayed in Figs. S2–S4. Edge dislocations are detected in the crystals from the TEM testing, which results from the lattice coupling of various PBAs with different $M^{HS}$. Uniform distribution of corresponding elements is displayed by EDS mapping, revealing that all the pre-set $M^{HS}$ are present in these PBAs. Full spectroscopy and the corresponding fine spectroscopy of XPS analyses also confirm the existence of pre-set transition metals in the crystals (Figs. S5–S8). The $g$ values of ~2.03 for three samples from the electron paramagnetic resonance (EPR) tests indicate the presence of $[Fe(CN)_6]^{4-}$ defect[28] (Fig. S9). It has been demonstrated that comparing the amplitude and peak width of EPR curves can reflect the concentration of $[Fe(CN)_6]^{4-}$ defects to some extent[29]. The amplitude difference between M4-PBA and M5-PBA in the curves is small, and the higher amplitude for M2-PBA indicates more crystal defects of $[Fe(CN)_6]^{4-}$ in comparison with M4-PBA and M5-PBA. Thermogravimetric analysis (TGA) in Fig. S10, EA (Table S1) and ICP-OES tests (Table S2) confirm that the chemical formulas of M2-PBA, M4-PBA, and M5-PBA are $Na_{1.84}Mn_{0.50}Fe_{0.50}[Fe(CN)_6]_{0.89}\square_{0.11}\cdot2.74H_2O$, $Na_{1.89}Mn_{0.27}Fe_{0.27}Co_{0.25}Ni_{0.21}[Fe(CN)_6]_{0.91}\square_{0.09}\cdot1.42H_2O$ and $Na_{1.90}Mn_{0.23}Fe_{0.22}Co_{0.20}Ni_{0.18}Cu_{0.17}[Fe(CN)_6]_{0.93}\square_{0.07}\cdot1.14H_2O$, respectively, where ϒ denotes $[Fe(CN)_6]$ vacancies in the crystal lattice and the content of crystal water in PBAs was determined by TGA[30,31]. The contents of transition metal elements in the PBAs materials closely match the initial feeding amounts, with slight deviations likely due to the differences in the complexing capabilities between various transition metal elements and chelating agents[32].

The powder X-ray diffraction (PXRD) patterns of the three samples were analyzed through Rietveld refinement, with atomic site occupancies in agreement with the calculated stoichiometry (Fig. 1k–m). Although there are differences in elemental selection and water content for these PBAs, all samples exhibit a monoclinic structure[33]. The crystallographic information for the three samples is detailed in Tables S3–S5. The insets show the corresponding charge density distribution diagrams of the (200) crystal planes based on the Rietveld refinement PXRD patterns of these PBAs. In M2-PBA, where the $M^{HS}$ sites are exclusively occupied by Mn and Fe, the electron density of the cyanide ligands shifts toward $M^{HS}$ ions. In M4-PBA, the partial substitution from Mn and Fe to Co and Ni at the $M^{HS}$ sites results in a symmetrical distribution of cyanide electron clouds between $M^{HS}$ and $Fe^{LS}$. The incorporation of Cu at $M^{HS}$ sites (M5-PBA) induces an electron density redistribution of the cyanide coordination structure toward the $Fe^{LS}$ centers. Additionally, the cyanide electron clouds with a pear shape in M2-PBA and M5-PBA can be observed in the insets of Fig. 1k, m, while M4-PBA displays a uniform cyanide electronic distribution with an ellipsoidal shape, aligning with the intended distribution of cyanide electrons according to the results of first-principles calculations in Fig. 1a–j.

## Characterization of the localized cyanide coordination structure

Since both Fe–N and Fe–C coordination bonds exist in all three samples, various structural characterization techniques focused on the Fe element were employed to investigate the effect of the electronic structural modulation on cyanide coordination structure in PBAs. Fe K-edge X-ray absorption fine spectroscopy (XAFS) was utilized to analyze the cyanide electronic environments in the three samples[34,35]. The X-ray absorption near-edge structure (XANES) spectra of the three samples show minimal differences, in which M5-PBA exhibits a relatively lower near-edge absorption energy, indicating less oxidation of $Fe^{2+}$ (Fig. 2a). All three samples exhibit distinct pre-edge peaks in their K-edge spectra of Fe, which can likely be attributed to relatively high sodium content (Fig. S11a)[36]. The distinct profiles of Fourier-transformed EXAFS (FT-EXAFS) in Fig. 2b with their spectra of EXAFS oscillations (Fig. S11b) exhibit different coordination structures of the central Fe atom for the three samples, correspondingly revealing the coordination environment of Fe–C and Fe–N shells in PBAs. The splitting of Fe–N profiles in all samples indicates the influence of $M^{HS}$ on the Fe–N coordination shell. The higher profile intensity for both Fe–C and Fe–N coordination shells reflects an ideal coordination environment for M4-PBA[37]. Besides, the appearance of the shoulder peak at 2061 $cm^{-1}$ detected by the FT-IR test in M2-PBA (Fig. S12) and the blue shift of the Raman peak at about 2071 $cm^{-1}$ representing $M^{HS}$–N for M5-PBA (Fig. S13) provide direct evidence for their relatively lower structural symmetry of coordination environment in comparison with M4-PBA[38,39]. The low coordination environmental symmetry is affected by the reduced shift of cyanide electron cloud toward either $M^{HS}$ or $Fe^{LS}$, namely, an increased distribution nonuniformity of cyanide electron cloud. However, a more symmetrical localized environment of cyanide electron cloud does not necessarily imply stronger cyanide coordination bonds in the crystals, which is confirmed by the electron binding energy between Fe and cyanide according to the XPS results of Fe 2p. As shown in Fig. 2c, M4-PBA exhibits the lowest electron binding energy of Fe 2p in comparison with M2-PBA and M5-PBA. The decreased electron binding energy of Fe 2p in M4-PBA is partly due to the strength abatement of cyanide coordination bonds, which reduces the electron binding force on Fe ions. Besides, the uniform and symmetric distribution of the cyanide electron can also reduce the confinement of electrons on Fe ions.

Further, the electronic environment and oxidation state of Fe ions were revealed by Mössbauer spectroscopy. According to the fitted Mössbauer spectra of the three samples in Fig. 2d–f, the red and green doublet peaks represent the deconvolution results for N-coordinated $Fe^{3+}$ and $Fe^{2+}$, respectively, while the blue singlet peak corresponds to C-coordinated $Fe^{LS}$. The pie charts in the inset reflect the amount of Fe

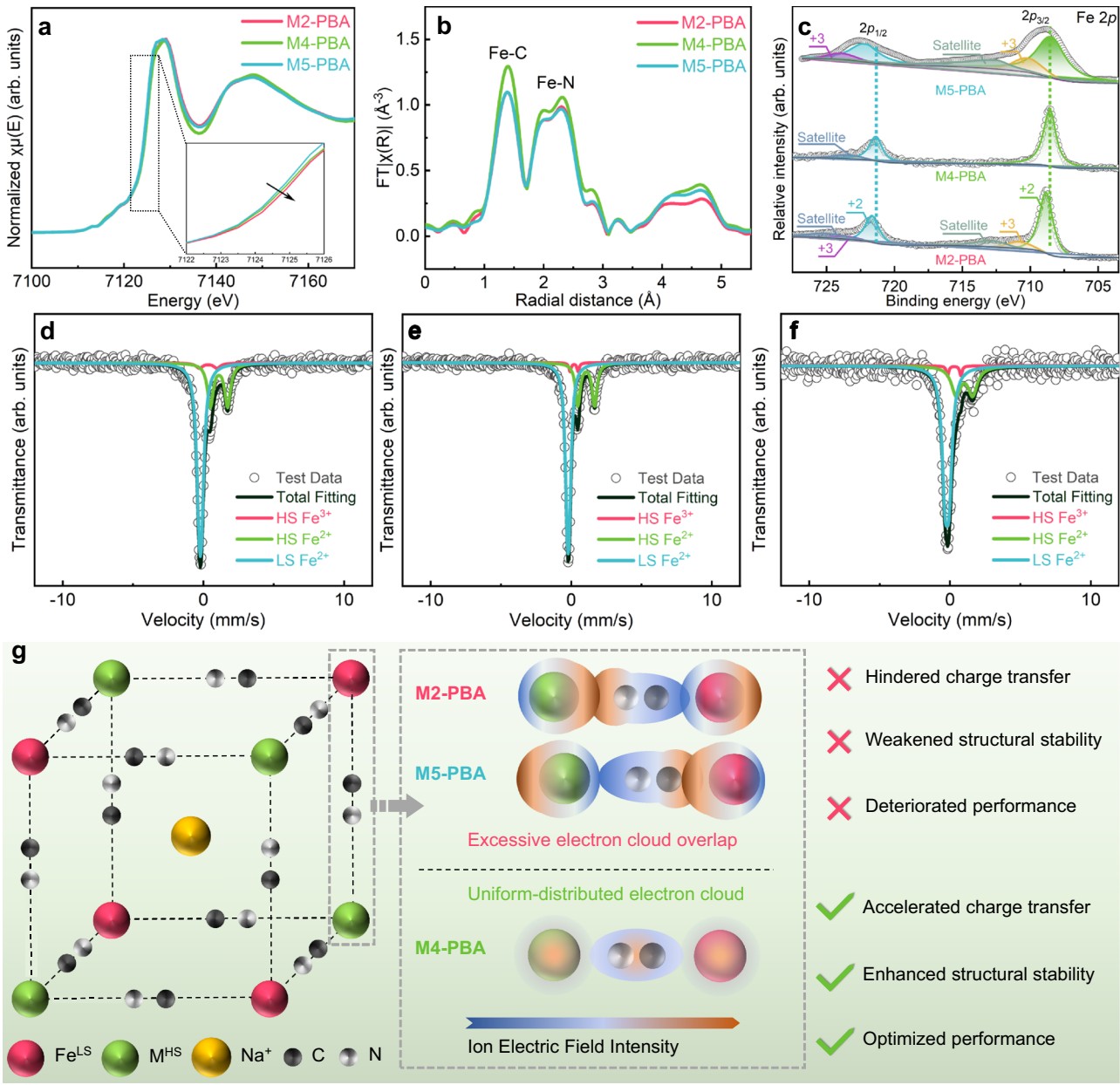

**Fig. 2 | Characterization of cyanide coordination structures. a** XANES. **b** FT-EXAFS. **c** XPS. Mössbauer spectra fitting results with the proportion of Fe ions in different coordination states shown in the inset: **d** M2-PBA; **e** M4-PBA, and **f** M5-PBA; **g** diagrammatic drawing of cyanide coordination electronic structure.

ions with different valence in the three samples, indicating the higher oxidation of N-coordinated Fe ions in M2-PBA. The detailed fitting parameters in Mössbauer spectra are listed in Table S6, where IS (isomer shift) provides information related to oxidation state and coordination bonds of Fe ions, QS (quadrupole splitting) reflects the nuclear charge distribution of Fe ions at the respective sites, and $\Gamma$ (half-peak breadth) indicates the degree of disorder in the local coordination environment of Fe ions. As shown in Fig. S14a, the ISs for N-coordinated Fe ions of the three samples display significant differences. On one hand, the oxidation content of Fe ions in the crystal leads to differences in the IS values. On the other hand, variations in the content and selection of $M^{HS}$ alter the electron environment around Fe ions and the corresponding coordination bond energy, further causing differences in the IS values[14,40]. Regardless of the valence of N-coordinated Fe ions, the IS values in M2-PBA are the highest (1.11 mm/s for $Fe^{2+}$ and 0.34 mm/s for $Fe^{3+}$), indicating that the electron density of N-coordinated Fe ions in M2-PBA is increased due

to the shift of the cyanide electron cloud toward the N-coordinated Fe. In contrast, the IS values of the N-coordinated $Fe^{2+}$ and $Fe^{3+}$ in M4-PBA are 1.06 mm/s and 0.3 mm/s, respectively, which fall in comparison with those of M2-PBA and M5-PBA (1.02 mm/s and 0.2 mm/s in M5-PBA, respectively). Thus, M4-PBA presents a moderate IS value through the modification of the local cyanide coordination structure, which can better keep the even distribution of cyanide electron cloud between $M^{HS}$ and $Fe^{LS}$.

The QS value reflects changes in the electronic environment around Fe ions in these PBAs (Fig. S14b). When the symmetry of the coordination environment around Fe ions decreases, the electric field gradient around the ions increases, leading to an increase in the QS values[41]. Compared to the QS values of N-coordinated Fe ions in M5-PBA (0.95 and 1.18 mm/s for $Fe^{2+}$ and $Fe^{3+}$, respectively), the QS values of M2-PBA are significantly higher (1.26 and 1.23 mm/s for $Fe^{2+}$ and $Fe^{3+}$, respectively). This phenomenon indicates that the N-coordinated Fe ions are influenced by the shifting of the cyanide group electronic

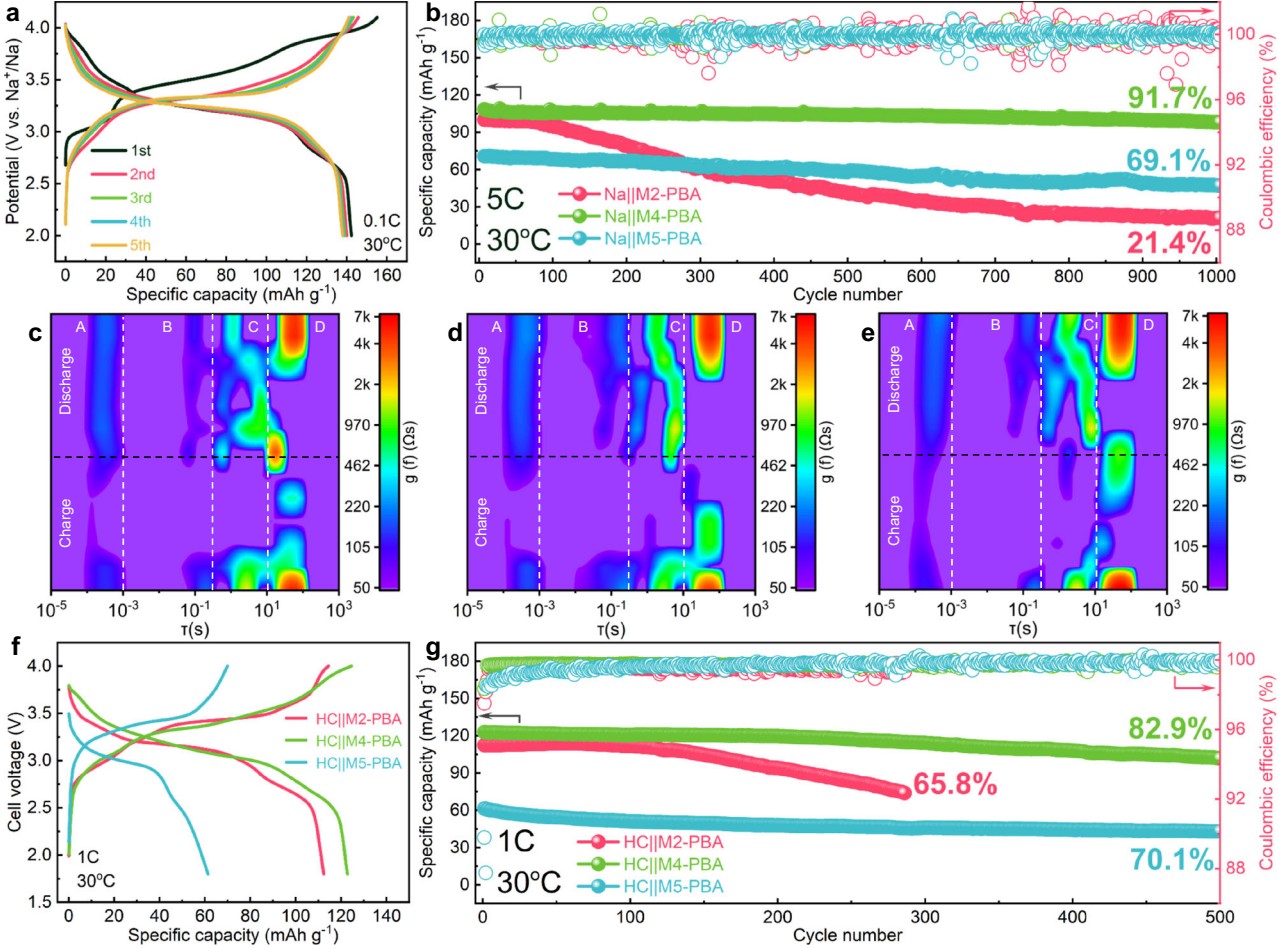

**Fig. 3 | Electrochemical characterization of PBAs with modified cyanide electronic structures at 30 °C. a** the first five times galvanostatic charge and discharge curves of M4-PBA, **b** comparison of cycling performance. DRT results of **c** M2-PBA, **d** M4-PBA, and **e** M5-PBA. Full cell electrochemical performance comparison of **f** galvanostatic charge and discharge curves and **g** cycling performance of 1 C.

structure in M2-PBA, resulting in the symmetry reduction of the electronic environment around the N-coordinated Fe. Correspondingly, the QS value of C-coordinated Fe ions (0.29 mm/s) in M5-PBA is higher than the QS value of M2-PBA (0.17 mm/s), suggesting that the electron cloud of cyanide is more biased toward the C-coordinated Fe ions in M5-PBA. In contrast, the QS values for N-coordinated $Fe^{3+}$ and $Fe^{2+}$ in M4-PBA (0.59 mm/s and 1.19 mm/s, respectively) and the QS value for $Fe^{LS}$ (0.13 mm/s) are smaller, indicating that the electronic distribution of both N-coordinated and C-coordinated Fe ions is relatively symmetrical. It suggests that the cyanide group electrons are more uniformly distributed in the coordination structure of M4-PBA. Similarly, an increase in Γ values suggests an increased disorder of the coordination structure[42] (Fig. S14c). The lower QS and Γ values of M4-PBA indicate that the local electron symmetries of Fe ions are slightly influenced by the cyanide electron cloud in comparison with the other two samples. Additionally, the introduction of other $M^{HS}$ somewhat changes the high symmetry of the C-coordinated Fe ions in M5-PBA via the coordination bonds of cyanide. Thus, the results of Mössbauer spectra indicate that the electron cloud of cyanide in M4-PBA is well-maintained, symmetrical, and centered between the $Fe^{HS}$ and $Fe^{LS}$, consistent with the electronic simulation results for the (200) crystal planes in XRD patterns (insets in Fig. 1k-m). Hereto, by designing the selections and proportions of $M^{HS}$, as shown in the schematic diagram of Fig. 2g, we can modulate the electronic distribution of $M^{HS}$ and $Fe^{LS}$, consequently achieving a uniform distribution of cyanide electronic structure between $M^{HS}$ and $Fe^{LS}$.

## Electrochemical performances of PBAs

Benefiting from the uniform distribution of cyanide electrons in PBAs, the electrochemical performances are improved. The galvanostatic charge and discharge curves of the first five cycles (Fig. 3a and Fig. S15a, b) and the corresponding dQ/dV curves were measured at a current density of 0.1 C (1 C = 170.0, 148.8, and 136.0 mA·$g^{-1}$ for M2-PBA, M4-PBA, and M5-PBA, respectively) for the three samples (Fig. S16). The initial discharge capacities of M2-PBA, M4-PBA and M5-PBA are 142.7, 142.4 and 109.5 mAh·$g^{-1}$, respectively. Notably, M4-PBA displays the highest proportion of theoretical capacity, indicating that the modulation of the cyanide electronic structure is more favorable to exert the activity of both $M^{HS}$ and $Fe^{LS}$. The galvanostatic charge/discharge curves and dQ/dV results further verify the influence of cyanide electron structure on the reversible capacity. Compared to M4-PBA, the previous analysis reveals that the cyanide ligand electrons in M2-PBA and M5-PBA are biased to $M^{HS}$ and $Fe^{LS}$, respectively, thereby inhibiting the redox activity of the corresponding transition metals[43,44]. Therefore, M2-PBA demonstrates a greater capacity contribution from $Fe^{LS}$ in the high voltage region (over 3.25 V), while M5-PBA shows a higher redox activity from $M^{HS}$ in the low voltage region (below 3.25 V)[45]. As for M4-PBA, the dQ/dV profiles show symmetrical redox doublets after the first cycle because of the even distribution of cyanide electrons between $M^{HS}$ and $Fe^{LS}$. The first charge/discharge and dQ/dV curves greatly differ from the subsequent curves, which is caused by the irreversible decomposition of crystal water during the charge and discharge process[15,46].

The modified cyanide electronic structure also boosts the cycling stability by providing unobstructed ion transport channels. During the electrochemical reaction, the interaction between guest ions and the coordination framework structure not only affects ion migration but also has an adverse impact on the stability of the frame structure, resulting in capacity decline[47]. Benefitting from the uniform distribution of cyanide electrons, M4-PBA shows better cyclic stability (Fig. 3b). The capacity retention of M2-PBA, M4-PBA, and M5-PBA is 21.4%, 91.7% and 69.1% after 1000 cycles at the current density of 5 C, respectively. Similarly, M4-PBA also exhibits a better capacity retention compared to M2-PBA and M5-PBA in the cyclic tests at 1 C (Fig. S17). In order to further compare the interaction between Na$^+$ and coordination frameworks for the three PBAs samples, in situ electrochemical impedance spectroscopy testing was employed to investigate the Na$^+$ transport characteristic during the electrochemical process (Fig. S18a–c). The individual electrochemical processes of Na$^+$ in PBA's positive electrode were revealed by distribution of relaxation times (DRT), considered as a broad range of analysis methods without the relatively rigid constraints of equivalent circuits[48,49]. Based on different relaxation times, the electrochemical reaction process of PBAs is divided into four regions, representing the internal electrical resistance of the battery components ($\tau_A = 10^{-5}$–$10^{-3}$ s), the cathode-electrolyte interface ($\tau_B = 10^{-3}$–0.5 s), charge transfer in the electrodes ($\tau_C = 0.5$–10 s), and solid-state diffusion in the electrodes ($\tau_D = 10$–$10^3$ s)[50,51]. The DRT results of the three samples during the first charge-discharge cycle are shown in Fig. 3c–e and Fig. S19a–c. The differences in the internal resistance of the battery devices and cathode electrode interface (CEI) resistance are minimal for the three samples. However, there are significant differences in the charge transfer resistance in the positive electrode and the solid-state diffusion resistance. Compared to M2-PBA and M5-PBA, M4-PBA especially exhibits smaller values of solid-state diffusion resistance in the high voltage region (Fig. 3c–e), indicating a lower Na$^+$ migration electrochemical impedance in the M4-PBA lattice. It suggests that Na$^+$ migrates more rapidly within the M4-PBA lattice, resulting in a lower polarization and a smaller coordination structural decay during the galvanostatic charge and discharge process.

The Na$^+$ diffusion ability in the lattice of the three samples, with the evolution of Na$^+$ diffusion coefficient during charging and discharging process, was revealed by galvanostatic intermittent titration technique (GITT) in Fig. S20, which also reflects the advantageous effect of the modified cyanide electronic distribution in M4-PBA. During the charging process, the diffusion coefficients of the three samples show a similar trend with voltage change (Fig. S20b). The diffusion coefficients significantly decrease in the voltage range of 3.2 to 3.6 V, due to the Jahn–Teller effect of Mn$^{3+}$ and phase transition in these PBAs[12]. Compared with M4-PBA and M5-PBA, the Na$^+$ diffusion coefficient of M2-PBA decreases significantly, suggesting the adverse influence of the Jahn–Teller effect and serious phase transition during the charge process. During the discharge process, the Na$^+$ diffusion coefficients of M2-PBA and M5-PBA greatly vary with a sharp drop in the range from 3.0 to 2.8 V, while the Na$^+$ diffusion coefficients of M4-PBA are well maintained at about $10^{-10}$ cm$^2$·s$^{-1}$ (Fig. S20c). The uniform distribution of cyanide electrons in M4-PBA can ensure the maintenance of a steady Na$^+$ diffusion rate during the discharge process. Moreover, CV tests at different sweep speeds were also conducted for the three samples with the fitting line in Fig. S21, which was obtained according to linear fitting results and supporting equations (Equations S1 and S2)[22,52]. When the slope value is close to 1, the diffusion of Na$^+$ in the crystal is mainly controlled by the capacitive behavior, but when the slope value is closer to 0.5, the migration rate of Na$^+$ in the lattice is mainly affected by the diffusion property[11,53]. That is, the higher the slope, the faster the Na$^+$ migration through the lattice. The slope values (b) of M2-PBA, M4-PBA, and M5-PBA during the oxidation process are 0.92, 0.96, and 0.84, respectively, with the

corresponding slope values of 0.68, 0.89, and 0.75 during the reduction process. Therefore, M4-PBA with the modified cyanide electronic structure exhibits the highest capacitance contribution in the charging and discharging process, which helps to ensure fast Na$^+$ mobility channels. As a result, the rate performance of M4-PBA is significantly improved in comparison with that of M2-PBA and M5-PBA (Fig. S22). The capacity of 85.1 mAh·g$^{-1}$ in M4-PBA is emitted at high rate of 20 C, which is higher than M2-PBA (44.36 mAh·g$^{-1}$) and M5-PBA (25.81 mAh·g$^{-1}$). The full cell performances using these PBAs as positive electrode materials with presodiation of hard carbon as negative electrode material are presented in Fig. S23a. Following the same trend observed in the half-cell electrochemical performance, M4-PBA demonstrates a higher capacity and better cycling stability in comparison with M2-PBA and M5-PBA (Fig. 3f, g). Notably, M4-PBA achieves a discharge energy density of 458 Wh·kg$^{-1}$ (based on the calculation of positive electrode material mass) within the voltage range of 4.0–1.8 V (Fig. S23b). The electrochemical performance comparison of PBAs in this work is presented as a radar chart in Fig. S24. Additionally, the comparative performance of M4-PBA with previously reported results is summarized in Table S7. The enhanced electrochemical performance of M4-PBA indicates that the uniform electron distribution of the Fe$^{LS}$–C≡N–M$^{HS}$ framework not only enables M4-PBA to fully exert its theoretical capacity, but also maintains the localized coordination structural stability in the charge and discharge process.

## Structural evolution of PBAs during the desodiation/sodiation process

In situ FT-IR and ex situ XAS testing were employed to investigate the evolution of the Fe$^{LS}$–C≡N–M$^{HS}$ coordination structure during the charge and discharge processes[54]. The in situ FT-IR results of the cyanide in M4-PBA are shown in Fig. 4a and Fig. S25. During the charging process, the absorption peaks of cyanide near 2075 cm$^{-1}$ (denoted as peak 1) gradually show a red shift to approximately 2050 cm$^{-1}$, which indicates the electron loss process of M$^{HS}$. With further charging, the single absorption peak of cyanide is split into a doublet containing the newly emerged infrared absorption peak near 2150 cm$^{-1}$, which represents a change of cyanide coordination environment caused by the oxidation of Fe$^{LS}$ (denoted as peak 2). Although some nonlinear changes are observed in peak 1 for M4-PBA during the first charging process, better reversibility is maintained in subsequent cycling (Fig. 4a). In contrast, significant mutants are observed in peak 1 for both M2-PBA and M5-PBA (Fig. S26a, b). The change of absorption peak 1 is related to the electron interaction of M$^{HS}$ and N. M4-PBA shows higher structural reversibility than M2-PBA and M5-PBA due to the regulation of cyanide electronic structure, thus maintaining higher cycling stability. The area ratios of peaks 1 and 2 were calculated to reflect the capacity utilization of various active sites (Fig. S27). Since the peak areas of peak 1 and peak 2 at the fully charged state partially reflect the redox activity of M$^{HS}$ and Fe$^{LS}$, respectively, quantitative analysis of the peak area ratio (peak 1/peak 2) provides critical insights into the influence of the uniform electronic distribution of Fe$^{LS}$–C≡N–M$^{HS}$ on redox activity utilization of Fe$^{LS}$ and M$^{HS}$. The increased peak area ratio indicates inefficient utilization of Fe$^{LS}$ redox activity, whereas the decreased ratio demonstrates inferior activation of M$^{HS}$. For M2-PBA, the relatively small peak area ratio value suggests that cyanide electron delocalization toward M$^{HS}$ partially hinders its activity. M5-PBA exhibits a larger peak area ratio at the fully charged state, signifying restricted Fe$^{LS}$ activity. Moreover, a linearly varying area ratio throughout the second cycling process is shown in M4-PBA, indicating a stable Na$^+$ insertion and extraction process due to the uniform electronic distribution of Fe$^{LS}$–C≡N–M$^{HS}$ coordination structure. The calculated results of M4-PBA in the area ratios of peaks 1 and 2 further highlight the relationship between the regulation of the cyanide coordination electronic structure and the full utilization of reversible capacity. By comparing the changes in area ratios of peaks 1

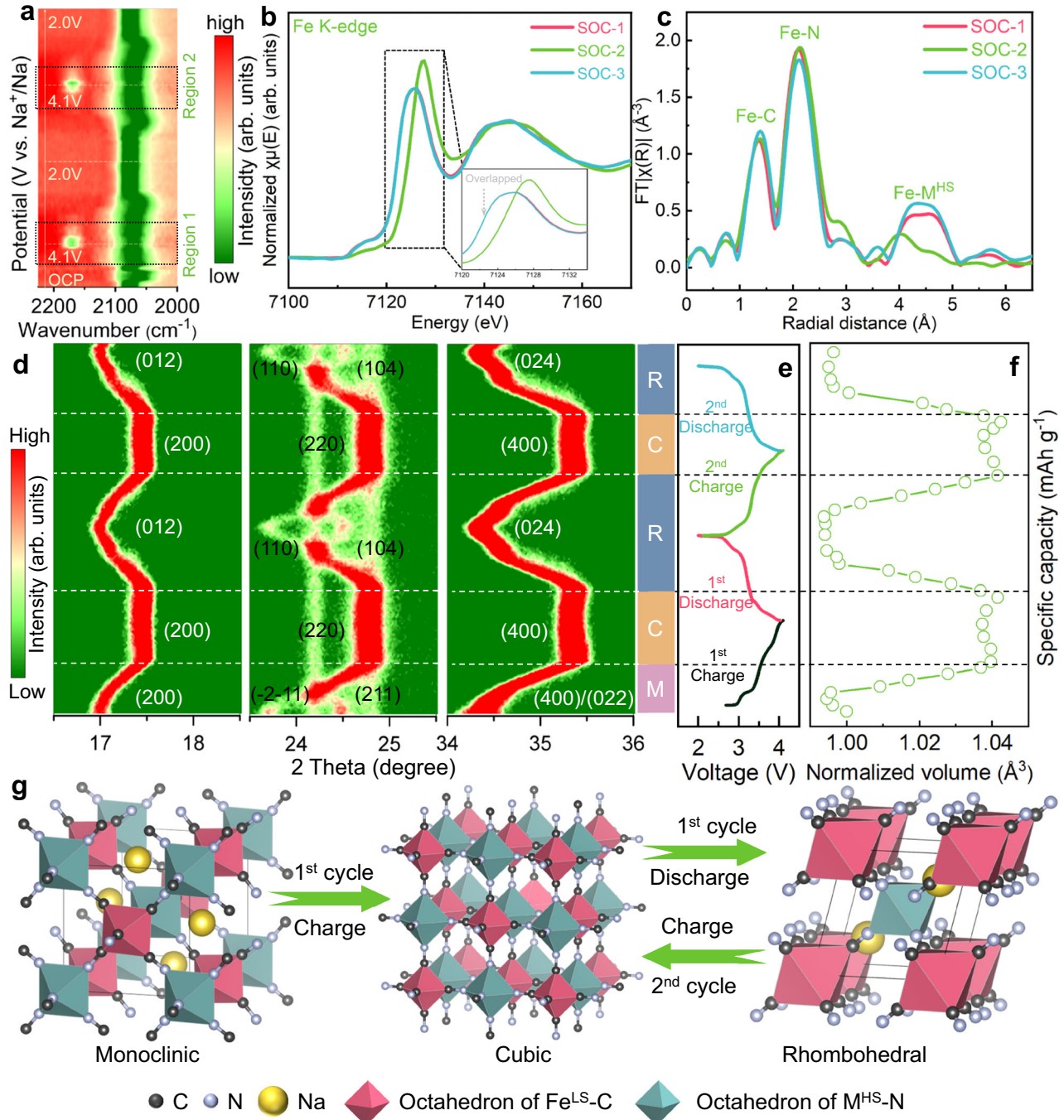

**Fig. 4 | Structural evolution of M4-PBA. a** In situ FT-IR for the first two cycles, ex situ XAS of different SOCs in the initial cycle: **b** XANES and **c** FT-EXAFS. (The final state of charge for the first discharge cycle was defined as SOC-1, while the charging and discharging final states of charge for the second cycle were defined as SOC-2 and SOC-3, respectively.) **d** In situ XRD patterns of M4-PBA and **e** the corresponding galvanostatic charge and discharge curves. **f** Normalized unit cell volume change of M4-PBA. **g** schematic illustration of the phase transition mechanism of M4-PBA.

and 2 between the first two cycling processes, M4-PBA shows a more consistent trend, suggesting a better stability of the $Fe^{LS}$–C≡N–$M^{HS}$ coordination structure.

The dissolution of transition metal elements is one of the primary factors leading to the capacity decay of PBAs[23], and maintaining the stability of the cyanide coordination electronic structure is beneficial to reduce the capacity loss. The ICP-OES results of the three samples at different cycles show the dissolution of transition metals (Mn and Fe ions) in the electrolyte (Fig. S28a–d). According to the previous research on Mn-based PBAs, $Mn^{3+}$ is considered the primary cause of electrode material failure due to the Jahn–Teller effect[32]. However, the

ICP-OES analysis of the three samples reveals the Fe dissolution concentration exceeding Mn by an order of magnitude during cycling. This phenomenon mainly stems from the destabilization of the inner-orbital $Fe^{LS}$–C coordination structure induced by the disruption of the outer-orbital $M^{HS}$–N coordination structure within the PBA coordination framework[26,45]. Furthermore, as the predominant transition metal across all three PBAs, Fe exhibits the highest dissolution tendency. The lower amount of transition metal dissolution not only signifies reduced active site loss but also indicates enhanced preservation of the $Fe^{LS}$–C≡N–$M^{HS}$ coordination framework. Combined with the decline in the reversible capacity of the electrode materials (Fig. 3b), it can be

deduced that the dissolution of Fe ions is one of the main factors contributing to the degradation of reversible specific capacity. Ex situ Fe K-edge XANES and EXAFS tests were further conducted to probe the evolution in the $Fe^{LS}–C≡N–M^{HS}$ coordination structure during the Na$^+$ insertion and extraction process (Fig. 4b, c). The XANES results suggest that all three samples maintain high reversibility in the redox behavior of Fe ions. The XANES spectra of the Fe element shift to a higher energy area during the Na$^+$ extraction process, indicating the oxidation of Fe to a higher valence state. During the Na$^+$ insertion process, the profiles of Fe finally reverse back to their original positions, manifesting that the valence states of Fe in the PBAs recover to their original states. The shift of XANES profiles after the first cycle for M2-PBA is more pronounced than that of M4-PBA (Fig. 4b) and M5-PBA (Fig. S29b), indicating relatively poorer $Fe^{LS}–C≡N–M^{HS}$ structural reversibility and redox activity of M2-PBA (Fig. S29a). As to the FT-EXAFS result of M2-PBA, the profiles of Fe–C and Fe–N coordination shells change obviously from SOC-1 to SOC-3 (Fig. S30a). In detail, both the profile intensity and position undergo significant changes in the FT-EXAFS curves and corresponding spectra of EXAFS oscillations (Fig. S31), demonstrating the poor stability of the $Fe^{LS}–C≡N–M^{HS}$ coordination structure. In the case of M5-PBA, although the FT-EXAFS profile of the Fe–N coordination shell is highly reversible, the shift of the cyanide electron clouds toward C leads to the stability reduction of $Fe^{LS}–C$ coordination environment (Fig. S30b). In contrast, the stable cyanide coordination electronic structure in M4-PBA is verified by the highly consistent FT-EXAFS curves of the Fe–C and Fe–N coordination shells at different states of charge, indicating the good coordination structural reversibility during cycling (Fig. 4c). Similarly, Mn K-edge XANES and corresponding EXAFS curves were collected and analyzed (Figs. S32 and S33). Benefiting from the uniform electronic distribution of $Fe^{LS}–C≡N–M^{HS}$, the profiles of the Mn–N and Mn–C coordination shells for M4-PBA exhibit smaller variations. This demonstrates that uniform electron distribution of $Fe^{LS}–C≡N–M^{HS}$ framework simultaneously enhances the stability of both $M^{HS}$ and $Fe^{LS}$ local coordination environments, which aligns with enhanced electrochemical cycling stability of M4-PBA.

The in situ XRD patterns of M4-PBA for the first two cycles were measured at a current density of 20 mAh·g$^{-1}$ (Fig. S34). The evolution of the characteristic peaks at about 17, 24, and 34° and the corresponding galvanostatic charge/discharge curves of M4-PBA are exhibited in Fig. 4d, e. The phase transition from monoclinic to cubic occurs for M4-PBA during the first Na$^+$ extraction process, and the material undergoes a phase transition from cubic to rhombohedral during the first Na$^+$ insertion process. In the second charging process, a reversible phase transition occurs between the rhombohedral and the cubic phase. A schematic diagram of phase transition processed for M4-PBA during the first two cycles is presented in Fig. 4g. The reversible structure transition confirmed by in situ XRD results ensures the better electrochemical performance for M4-PBA. A similar phase transition process of M2-PBA occurs in the ex situ XRD results for the first two Na$^+$ extraction/insertion processes (Fig. S35b–d). However, there is a significant fade in the intensity of the M2-PBA diffraction peaks in comparison with M4-PBA, indicating that M2-PBA undergoes a structural degradation in the second cycle. The irreversible change of coordination structure is consistent with previous ex situ FT-EXAFS results (Fig. S30). Different from the PXRD result of monoclinic M5-PBA powder, the pristine electrode of M5-PBA is determined as a rhombohedral phase, which results from the loss of some crystal water during the electrode preparation process[55]. With low content of crystal water, M5-PBA maintains the rhombohedral-cubic-rhombohedral phase transition throughout the initial two charge/discharge cycles (Fig. S36b–d). Intensity fade of Bragg diffraction peaks in M5-PBA suggests unstable structural evolution during the electrochemical process. Bragg's law allows quantitative determination of unit cell volume evolution in PBAs during the

Na$^+$ insertion and extraction process, providing critical insights into the effect of uniform cyanide electron distribution on structural stability. As shown in Fig. 4f, M4-PBA exhibits slight volume evolution caused by irreversible monoclinic-cubic phase transition during the initial charging process, and highly reversible unit cell volume variations are observed starting from the first discharge process. In contrast, M2-PBA shows minimal volume variation but suffers from poor reversibility (Fig. S35f). Although M5-PBA exhibits reversible volume changes (Fig. S36f), the substantial amplitude of these fluctuations compromises its structural stability during the repeated Na$^+$ insertion and extraction process.

The differences in the irreversible structural evolution among the three samples are attributed to the loss of crystallization water during the first Na$^+$ extraction/insertion process, coinciding with the variation trend of initial galvanostatic charge/discharge curves (Fig. 3a and Fig. S15a, b). Meanwhile, the $Fe^{LS}–C≡N–M^{HS}$ coordination electronic structure in PBAs plays an important role in the reversibility of the crystal structure changes during cycling. The well-distributed cyanide coordination electronic structure not only ensures stable and reversible structural phase transitions but also contributes to the enhanced stability of the crystal structure of M4-PBA during the electrochemical reaction process.

## DFT calculations

To investigate the localized electronic structure of the three PBA samples, DFT calculations were employed. According to the partial density of state (PDOS) of PBAs, $M^{HS}$ (such as Fe$^{2+}$, Mn$^{2+}$, and Co$^{2+}$) tend to be oxidized at low potentials, while $Fe^{LS}$ is oxidized at high potentials (Fig. 5a-c). The PDOS of $Fe^{LS}$ and $M^{HS}$ in M4-PBA is closer to the Fermi level compared to M2-PBA and M5-PBA, and thus both $Fe^{LS}$ and $M^{HS}$ coordinated with cyanide can be easily activated in M4-PBA, corresponding to higher practical discharge specific capacity. Furthermore, the PDOS of N in M2-PBA shows the highest overlap with the PDOS of $M^{HS}$, while the PDOS of C in M5-PBA exhibits the highest overlap with the PDOS of $Fe^{LS}$[56]. This indicates that M2-PBA possessed stronger coordination electronic interaction of N and $M^{HS}$, while stronger coordination electronic interaction between C and $Fe^{LS}$ is also exhibited in M5-PBA. In contrast, M4-PBA has moderate coordination electronic interaction, which not only enhances the stability of $Fe^{LS}–C≡N–M^{HS}$ coordination structure, but also provides a more uniform distribution of coordination electrons. The computational models of the electron clouds for the three samples and their corresponding cross-sectional charge density distribution diagrams are shown in Fig. 5d–f. According to the cross-sectional charge density distribution diagrams, M4-PBA exhibits lower charge density at both $M^{HS}$ and $Fe^{LS}$ compared to M2-PBA and M5-PBA, which is consistent with the XRD refinement results (Fig. 1k–m). Thus, the regulation of cyanide coordination electronic structure effectively reduces the charge density surrounding the transition metal elements of M4-PBA, thereby fully activating the transition metal ions and improving the practical reversible capacity. The evenly distributed cyanide coordination electronic structure is achieved by regulating the composition of $M^{HS}$ in PBAs (Fig. 5g). Moreover, the corresponding coordination electron interaction of transition metals and cyanide can be optimized to enhance the stability of $Fe^{LS}–C≡N–M^{HS}$ frameworks. Consequently, the optimized cyanide coordination electronic structure ultimately balances the reversible capacity and cycling stability of M4-PBAs.

## Discussion

According to the valence bond theory and the DFT calculations results of single N-coordinated metal PBA (Fig. 1a–j), the coordination electronic structure of PBAs is affected by the valence electron distribution of transition metals and the electronegativity of ligand elements. Although it has been proven that the stability of $Fe^{LS}–C≡N–M^{HS}$

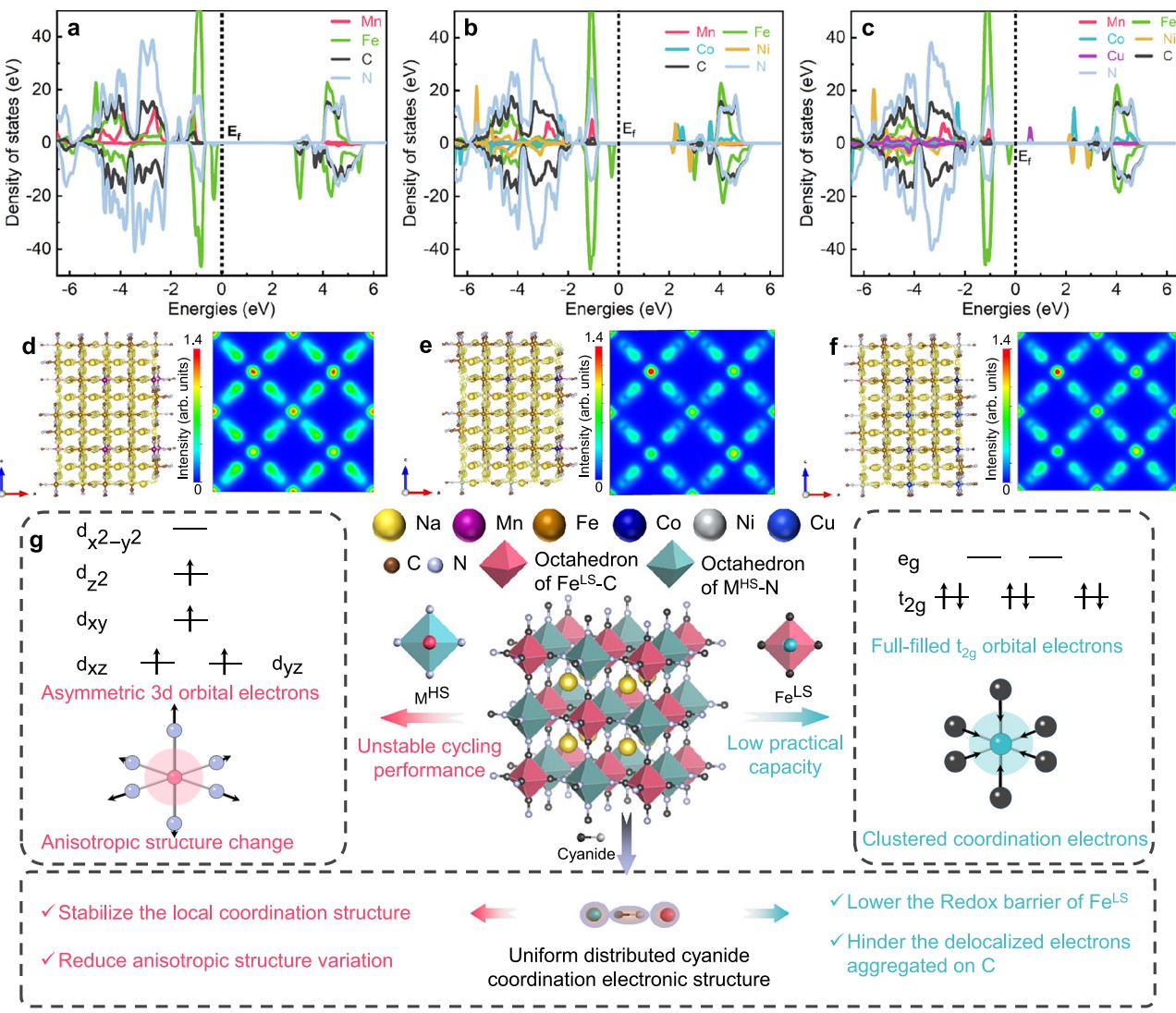

**Fig. 5 | DFT calculations.** PDOS profiles of **a** M2-PBA, **b** M4-PBA, and **c** M5-PBA. Structure model of electronic distribution: **d** M2-PBA, **e** M4-PBA, and **f** M5-PBA. **g** Schematic diagram of the enhanced electrochemical performance by cyanide electronic structure modification.

coordination frameworks can be improved by selecting transition metal cations with stronger π electron interaction to cyanide anions, the electrochemical performance enhancement mechanism of cyanide coordination electronic structure on cycling stability and redox activity of PBAs remains unclear. The uniform-distributed coordination electronic structure was accomplished for PBAs to achieve both high specific capacity and long cycling life. The optimized $Fe^{LS}-C\equiv N-M^{HS}$ electronic structure of M4-PBA is confirmed by the results of Rietveld refinement, PXRD patterns, and Mössbauer spectroscopy test. Further, linear variation of FT-IR absorption peak area ratios is beneficial to sufficiently activate the theoretical capacity and reduce the $Na^+$ migration energy barrier for M4-PBA. As to the cycling performance, the highly reversible structure change is reflected by ex situ XAS and in situ XRD, suggesting that the uniform distribution of $Fe^{LS}-C\equiv N-M^{HS}$ coordination electrons alleviates the irreversible structural change. The M4-PBA with the optimized coordination electronic structure can release a high capacity of 142.4 mAh·g$^{-1}$ at 0.1 C and retains 91.7% of its reversible capacity after 1000 cycles at 5 C. In addition, the higher rate capability of M4-PBA is also demonstrated with a reversible discharge capacity of 85.1 mAh·g$^{-1}$ at a high current density of 20 C. We believe that the homogenization of coordination electronic structure may open up potential avenues for future studies aimed at understanding and mitigating the degradation of PBAs.

## Methods

### Materials

The main chemical reagents included $Na_4Fe(CN)_6\cdot10H_2O$ (AR, ≥99%), PVP (K30, ~40,000), NaCl (AR, 99.5%), sodium citrate ($C_6H_5Na_3O_7\cdot2H_2O$, AR, > 99%), ascorbic acid (ACS, ≥99%), $MnSO_4\cdot H_2O$ (AR, 99.0%), $FeSO_4\cdot7H_2O$ (AR, ≥99%), $CoSO_4\cdot7H_2O$ (AR, ≥99%), $NiSO_4\cdot6H_2O$ (AR, 99.0%) and $CuSO_4\cdot5H_2O$ (AR, 99.0%). All the solid reagents were purchased from Shanghai Aladdin without further purification. Sodium tablet electrode (Na, purity > 99.7%, thickness 450 μm, diameter 15.6 mm) was purchased from Changgao New Materials Co., Ltd (CGM). Hard carbon (HC) was purchased from the Kuraray Co., Ltd., Japan. All the electrode materials were stored in the glovebox with water and oxygen concentrations below 0.01 ppm.

### PBAs preparation

5 mmol equimolar amount of $MnSO_4\cdot H_2O$, $FeSO_4\cdot7H_2O$, $CoSO_4\cdot7H_2O$, and $NiSO_4\cdot6H_2O$ were dissolved in 100 ml deoxygenated deionized water with 80 mmol sodium citrate to obtain solution A. Solution B was prepared by dissolving $Na_4Fe(CN)_6\cdot10H_2O$ with a molar concentration of 0.2 mol·L$^{-1}$ in 100 ml deionized water after removal of oxygen. 5 g PVP (K30), 23.4 g NaCl, and 0.2 g ascorbic acid were dissolved in deoxygenated deionized water to form a 200 mL transparent solution C. A white emulsion was evolved by dropwise pumping solution A and

B into solution C with a rate of 0.1 ml·min⁻¹. After the solutions were fully added, the mixed emulsion was left to stand and aged for an additional 12 h. The light green precipitate was centrifuged and transferred into a vacuum oven at 120 °C for 10 h, drying to obtain the prepared $Na_2Mn_{0.25}Fe_{0.25}Co_{0.25}Ni_{0.25}[Fe(CN)_6]$ with four $M^{HS}$ (Mn, Fe, Co, and Ni), defined as M4-PBA. The designed $Na_2Mn_{0.5}Fe_{0.5}[Fe(CN)_6]$ with two $M^{HS}$ (Mn and Fe) was defined as M2-PBA, and $Na_2Mn_{0.2}Fe_{0.2}Co_{0.2}Ni_{0.2}Cu_{0.2}[Fe(CN)_6]$ with five $M^{HS}$ (Mn, Fe, Co, Ni, and Cu) was marked as M5-PBA. A similar synthesis method was used to replace the transition metal salt in the synthesis of M4-PBA with the corresponding equimolecular amount of hydrated transition metal sulfate, with a total molar amount of 20 mmol, according to the chemical formulas of M2-PBA and M5-PBA.

## Electrochemical measurements

The positive electrode slurry was mixed by PBAs, Ketjen black, and polyvinylidene fluoride (PVDF) binder (5 wt% in N-methyl pyrrolidone) in a mass ratio of 7:2:1. It was evenly coated on aluminum foil with a thickness of 130 μm and then transferred to a vacuum drying oven at 120 °C for overnight. After being cut into a circle with a diameter of 14 mm, it was pressed under 10 MPa for 5 min. The active material loading of each slice was between 1.5 and 2.0 mg. A sodium metal tablet with a diameter of 15.6 mm was used as the counter electrode and reference electrode. The electrolyte, consisting of 1 mol·L⁻¹ NaClO₄ dissolved in a mixture of ethylene carbonate (EC), propylene carbonate (PC), and fluoroethylene carbonate (FEC) (volume ratio EC:PC:FEC = 47.5:47.5:5), was purchased from DodoChem. It was stored in an argon-filled glovebox with water and oxygen content below 0.01 ppm at a temperature of 30 °C. The glass fiber separators (Whatman, GF/D) were cut into circular disks with a diameter of 16 mm. These disks were dried in a vacuum oven at 120 °C for 12 h and subsequently transferred to an argon-filled glovebox (with water and oxygen content below 0.01 ppm) for storage and later use. Subsequently, a 2025 button battery was assembled in a glovebox with water and oxygen concentrations below 0.01 ppm. In full cell assembly, battery operations were performed in a glovebox (<0.01 ppm $H_2O/O_2$). The N/P ratio was set at about 1.05. The negative electrode slurry (hard carbon:Super P:PVDF = 8:1:1) was coated on Cu foil and dried at 120 °C for 12 h. In a half-cell against sodium, the negative electrode with a diameter of 14 mm was first cycled twice within the voltage range of 0.01–2.5 V, followed by discharging to 0.05 V for presodiation. The electrochemical performance data of galvanostatic charging and discharging curves were all measured by the Neware battery test system in the voltage range of 2.0–4.1 V at 30 °C. For the cycling performance test, the three samples were first activated five times at 0.2 C, and subsequently subjected to subsequent cycles at different rates. In the rate performance test, the three samples were cycled for 5 times each at the rate of 0.1, 0.2, 0.5, 1, 2, 5, 10, and 20 C in sequence, followed by 60 cycles at 1 C. GITT testing was conducted after five cycles of the fresh cell at a current density of 0.1 C between 2.0 and 4.1 V, in which the cell was alternately charged for 10 min followed by 60 min resting, then discharged in the same way. The GITT diffusion coefficient was calculated using the built-in function of the Neware test system. The cyclic voltammetry (CV) curves and in situ transient electrochemical impedance spectroscopy (in situ EIS) were measured by Shanghai Chenhua CHI 760e electrochemical workstation at 30 °C. The scanning speed of CV curves (from 2.0 V to 4.1 V) was 0.2, 0.4, 0.6, 0.8, and 1.0 mV·s⁻¹, respectively. The in situ EIS was tested from the open circuit voltages of each sample with the following voltage interval of 0.3 V from 2.9 V until the end of the first galvanostatic charging and discharging cycle. The voltage range for the in situ EIS measurement was from 2.0 V to 4.1 V, with a test frequency range of 10⁵–10⁻² Hz and an amplitude of 5 mV. The current density of the galvanostatic charging and discharging process of in situ EIS testing was 0.1 C for each sample. In electrochemical testing, the current density, specific capacity, and

specific energy of the battery are all calculated based on the mass of the active material in the electrodes.

## Materials characterization

The crystalline structures of the prepared samples were analyzed using laboratory powder X-ray diffraction (PXRD) with a Cu Kα source on an X'Pert3 MRD diffractometer, and in situ XRD tests were conducted using a Bruker AXS D8 Focus instrument. The microtopography of PBAs was examined via field emission scanning electron microscopy (FE-SEM, VEGA3 TESCAN), while scanning transmission electron microscopy (STEM, JEM-ARM 200F), equipped with energy dispersive spectroscopy (EDS), provided insights into the elemental distribution of the as-prepared samples. High-resolution transmission electron microscopy (HR-TEM) measurements were conducted with an FEI Tecnai G2 F30. Surface composition analysis was performed using X-ray photoelectron spectroscopy (XPS, EscaLab 250xi), and electron paramagnetic resonance (EPR) spectra were obtained using a Bruker A300-10/12 spectrometer. An elemental analyzer (EA) was performed using an Elementar Vario EL III elemental analyzer (Germany). Thermogravimetric analysis (TGA) measurements were taken from 25 to 400 °C at a heating rate of 5 °C min⁻¹ in nitrogen. FT-IR spectra, both powder and in situ, were recorded using Bruker FT-IR spectrometers (UK) and Nicolet 6700 FT-IR spectrometers (Thermo Scientific, USA). Raman spectroscopy was performed using a Renishaw inVia system with a 532 nm laser. The ⁵⁷Fe Mössbauer spectra were collected using a constant acceleration Halder-type spectrometer in transmission geometry, with a room-temperature ⁵⁷Co source embedded in a Rh matrix. The concentrations of sodium and transition metals in the samples and cycled electrolyte were determined using an inductively coupled plasma optical emission spectrometer (ICP-OES) analysis (OPTIMA 8000DV Optical Emission). Hard X-ray absorption spectroscopy (XAS) measurements were carried out in a transmission mode at BL17B beamline, Shanghai Synchrotron Radiation Facility (SSRF). For each spectrum, the scanning energy range was −200 eV to 700 eV with a step size of 0.5 eV near the K-edge (−30 eV to 100 eV) and a step size of 3 eV for the other energy ranges. X-ray photon energy was monochromatized by a Si (111) channel-cut crystal monochromator.

## Theoretical calculations

Theoretical calculations were performed using density functional theory (DFT) as implemented in the Vienna Ab-initio Simulation Package (VASP). The projector augmented wave (PAW) method was employed alongside the Kohn-Sham equations, utilizing the spin-polarized generalized gradient approximation (GGA) with the Perdew−Burke−Ernzerhof (PBE) functional to analyze the electronic interactions between valence electrons and core ions, as well as the exchange-correlation effects. To account for electronic correlation effects of transition metal 3$d$ electrons, Hubbard U corrections were applied for Mn (5.0 eV), Fe (4.3 eV), Co (5.7 eV), Ni (6.0 eV), and Cu (4.0 eV). The van der Waals dispersion interactions were incorporated into the calculations by employing the DFT-D3 dispersion correction method. In the DFT calculations, the kinetic energy cutoff for the electron wave functions was set to 520 eV. Geometry optimization was conducted using the conjugate gradient method, with convergence criteria of 10⁻⁵ eV for energy and 0.02 eV Å⁻¹ for forces. For the single $M^{HS}$-based PBAs, due to their high symmetry, the unit cells for computational modeling were employed directly without supercell construction (Fig. 1a-j). Depending on the phase structure differences, the number of atoms per unit cell varies. For the cubic phase (desodiated state), there are 58 atoms in the unit cell, while 48 atoms are in the unit cell of the rhombohedral phase (sodiated state). As to the density functional theory (DFT) calculations of the actual PBA structures, we constructed a 2 × 2 × 1 supercell containing 192 atoms to closely match the elemental ratios observed in the experimentally synthesized PBA crystals. The Brillouin zone was sampled using a 3 × 3 × 3 Monkhorst-

Pack k-point grid. Visualization of the electrolyte structures was performed using VESTA. The atomic coordinates from DFT calculations have been provided in the "Supplementary Data" file.

## Data availability

The data generated in this study are provided in the Source Data file. Source data are provided with this paper.

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

## Acknowledgements

The authors would like to express their appreciation to the National Natural Science Foundation of China (No. 52272241, L.W.), the Natural Science Foundation of Heilongjiang Province for Distinguished Young Scholars (JQ2024B001, P.Z.), the National Key R&D Program of China (2024YFA1211900, L.W.), the Key Research and Development Program of Heilongjiang Province (No. GA21A102, P.Z.), Guangxi Key Technologies R&D Program (Grant No. AB25069462, P.Z.), Zhejiang Provincial Natural Science Foundation of China under Grant No. LR24E020001 (L.W.), the Leading Innovative and Entrepreneur Team Introduction Program of Zhejiang (No. 2023R01007, L.W.), Jiangsu Provincial Double-Innovation Doctor Program (No. JSSCBS20220308, B.X.), the Harbin Science and Technology Innovation Talent Project (2023CXRCGD036, P.Z.), and the Fundamental Research Funds for the Central Universities (No. HIT.DZJJ.2023052, Y.W.). We thank the staff of the BL17B beamline (https://cstr.cn/31129.02.NFPS.BL17B) at the National Facility for Protein Science in Shanghai (NFPS, https://cstr.cn/31129.02.NFPS), Shanghai Advanced Research Institute, Chinese Academy of Sciences, for their technical support in X-ray absorption spectroscopy measurements.

## Author contributions

Y.W., B.X., and L.W. proposed the design concept and completed the PBA material preparation. Y.W. and J.Y. conducted DFT computations. Y.W., L.W., and X.W. performed the X-ray absorption fine spectroscopy test. Y.W. and Y.M. carried out the assembly of the full cells and completed the electrochemical performance test. C.F. and F.K. provided advice on material structural characterization. Q.Z. and X.C. provided suggestions on electrolyte selection and electrochemical testing. J.L. and C.D. assisted in the improvement of material processing techniques. B.X., L.W., and P.Z. supervised this work and revised the paper. All authors made contribution to the experimental testing, data analysis, and discussion of the results in this thesis.

## Competing interests

The authors declare no competing interests.
