## [Transparent Peer Review file · Nature Communications]

Tuning Cyanide Coordination Electronic Structure Enables Stable Prussian Blue Analogues for Sodium-ion Batteries

Corresponding Author: Professor Pengjian Zuo

Version 0:

Reviewer comments:

Reviewer #1

(Remarks to the Author)

Manuscript background information

This paper investigates the challenges associated with utilizing Prussian blue analogues (PBAs) as cathode materials for sodium-ion batteries, particularly addressing issues such as structural instability and capacity degradation due to transition metal dissolution and Jahn-Teller distortion. While the topic is of interest, the cause-and-effect relationships are not clearly established in the manuscript, and a more complete data analysis is required. Although the focus is on the coordination electronic structure—a concept that is not well defined within the paper—it remains unclear how the three proposed structures effectively support the hypothesis presented. Furthermore, the discussion of the localized electronic structure is based solely on DFT calculations, and a more robust experimental validation would be highly beneficial. Additionally, the experimental section lacks sufficient detail to adequately support the findings.

Specific major questions

Line 116 – 117, table S1: stoichiometry calculation – how from only ICP-OES and TGA was it possible to determine the ratio between high and low spin Fe? And therefore, define the coefficient of $[\text{Fe}(\text{CN})_6]$ unit?

Line 130, table S2-S4: the occupancy inside the refinement parameters does not match with stoichiometry, considers equimolar distribution of all high spin metals (and 2Na), and doesn't consider vacancies that were included in the formulae (maybe it could be mentioned, that within the quality of the data, further accuracy couldn't be reached). Also, the rationale for the selected three materials is not clear. M in the general formula is one metal. Why the authors synthesized different materials by substituting the same (potential) crystalline site? This is not suggested by the DFT calculated structure reported previously.

Line 137. The sentence "Since Fe^{2+} is introduced in both high-spin sites and low-spin sites for all the three samples" is not clear. Please specify.

Line 142-143, fig. 2a: it's written "M4-PBA exhibits a relatively lower near edge absorption energy, indicating a less oxidation of Fe^{2+} ", however in the figure 2a inset, M5-PBA looks more shifted towards the lower energy than M4-PBA;

Line 143-145: it is written: "An increase in the pre-edge peak intensity of M4-PBA suggests a slight deviation from octahedral symmetry due to a higher sodium content", however later the authors are always speaking about M4-PBA having highest symmetry among three, which sounds contradictory. Also, better to add extra figure or inset of pre-edge, because in fig 2a, it is really hard to see any difference;

Line 146, fig 2b: better to mention that it is Fourier transform and not directly EXAFS spectra, also better to additionally show the actual spectra (oscillations), do give the reliability to the authors claims about the "different coordination structures of central Fe atom for the three samples";

Line 149. The sentence "The higher profile intensity for both Fe-C and Fe-N coordination bonds reflects the fine structural symmetry for M4-PBA" is lack of clarity and meaning. Peak intensity in the FT EXAFS are not due to symmetry, nor to the Fe-N is a direct bond.

Line 164, fig 2c: for the XPS spectra fitting for M5-PBA, the Fe^{+3} contribution for Fe 2p_{3/2} is visible, however, Fe 2p_{1/2} it is not; also, in M4-PBA, there seems to be a satellite around 715 eV, which is not assigned or fitted; Please explain;

Line 220. Not clear why M4 sample displays the highest proportion of theoretical capacity

Line 226-228, fig. S15: the voltage is divided in two regions below and above 3.5 V; but why exactly 3.5 V? it is not the plateau position, neither the position in-between the peaks on dQ/dV curve, also doesn't help visually the statement about

M4_PBA: "more symmetrically redox doublet peaks";

Line 313: A mitigation of the JT effect is stated here, but it is not substantially proven.

Line 316-319: it is a little unclear.

Line 333-336. It is suggested that Fe dissolution might occur based on IPC-OES, but an explanation is missing;

Line 337. What "cyanide coordination" stands for?

Line 343-345: why the authors think that the larger shift in XANES of M2-PBA means the "FeLS-CN-MHS coordination structure is taken apart"? Also, why only Fe K-edge data are presented? What's about the other M K-edge?

Line 346. EXAFS. The authors made some chemical/structural deduction on spectra only based on experimental FTs, this leads to misinterpretation of the structural variations

Experimental. Details missing in the various adopted experimental techniques, such as - for example in XAS data at synchrotron, starting E and final E as well as step of each K-edges, detector used and characteristics, use of mirrors?, monochromator and energy of the sources.

Minor points

Line 102, please specify which metals are involved in the notation M2-PBA, M4-PBA, and M5-PBA, upon mentioning them the first time"

Line 105: "transmission electronic microscopy (TEM)" change to "electron";

Line 125, fig. 1j: the figure is oriented at angle;

Line 130, table S2-S4: in monoclinic cells, it's better to also indicate the numeric value of the angle which differs from 90°;

Line 164: figure caption "a" and "b" are mixed;

Line 266: the indication "(Fig. S19b)" needs to be written one sentence up, when there is explanation about charging (not about discharging);

Reviewer #2

(Remarks to the Author)

Reviewer #3

(Remarks to the Author)

The work reported that modulating the local electronic structure surrounding high-spin metals in PBAs can be employed to optimize the cyanide coordination environment, enabling a uniform electron distribution within the crystal structure. Thanks to the uniform electron distribution, the reactivity of the transition metals can be enhanced to get a high sodium storage capacity. In addition, the regulation of electronic displacement within the cyanide coordination environment significantly improves the crystal structural stability. The work is significant and can be considered in Nc Journal after a further improvement.

1)How to choose the transition metal type and the combination of the transition metals? And how many kinds are reasonable? Is there a principle of universality? In the work, the samples of $\text{Na}_{1.84}\text{Mn}_{0.503}\text{Fe}_{0.497}[\text{Fe}(\text{CN})_6]_{0.892}\cdot 1.633\text{H}_2\text{O}$, $\text{Na}_{1.927}\text{Mn}_{0.266}\text{Fe}_{0.247}\text{Co}_{0.248}\text{Ni}_{0.239}[\text{Fe}(\text{CN})_6]_{0.943}\cdot 0.862\text{H}_2\text{O}$ and $\text{Na}_{1.898}\text{Mn}_{0.211}\text{Fe}_{0.202}\text{Co}_{0.199}\text{Ni}_{0.198}\text{Cu}_{0.190}[\text{Fe}(\text{CN})_6]_{0.927}\cdot 0.675\text{H}_2\text{O}$ are selected. Why use the combination of MnFe, MnFeCoNi, and MnFeCoNiCu with different types and numbers? How to decide? We just observe an uniform and symmetric distribution of the cyanide electron for the M4 sample based on the calculation and experiments, while we don't know the optimized procedure of designing the 4 transition metals together.

2)Whether the uniform electron distribution for the crystal structure is suitable for the design of other kind of cathode, for example, polyanion types?

Reviewer #4

(Remarks to the Author)

Prussian blue analogues (PBAs) with open 3D framework structure are highly promising for practical applications as cathode of sodium-ion batteries (SIBs) due to their high theoretical energy density, low cost, and ease of large-scale preparation. However, PBAs often suffer from capacity degradation, resulting in continuous energy loss, impeding commercialization for practical SIBs. Herein, Wang et al. addressed the challenges by modulating the local electronic structure surrounding high-spin metals to optimize the cyanide coordination environment, enabling a uniform electron distribution within the crystal structure. The authors found that uniform electronic structure enhances the reactivity of the transition metals, resulting in 95.7 % of the theoretical capacity. The reported results demonstrate the potential of PBA cathode for large-scale commercial SIBs. The manuscript is of wide interest, of good significance and of high impact. Considering its valuable insights into enhancing the practical application of SIBs, I recommend this manuscript for publication in Nature Communications after some proper revision are made.

The following points are my concerns to the manuscript that require further clarification.

1. In the "PBAs Preparation" section, the authors wrote that 5 mmol equimolar amount of $\text{MnSO}_4\cdot\text{H}_2\text{O}$, $\text{FeSO}_4\cdot 7\text{H}_2\text{O}$, $\text{CoSO}_4\cdot 7\text{H}_2\text{O}$, and $\text{NiSO}_4\cdot 6\text{H}_2\text{O}$ in 0.8 mol·L⁻¹ sodium citrate solution were dissolved to obtain 100 ml solution A. From this description, one cannot figure out how many grams of mols of sodium citrate, or salts in solution A.

2. In the "Theoretical calculations" section, the authors used the spin-polarized generalized gradient approximation (GGA) with the Perdew-Burke-Ernzerhof (PBE) functional with Hubbard U corrections. It is known that Hubbard U corrections can

improve the results for electronic properties such as band gap. However, the van der Waals dispersive interactions is underestimated. Why not to use dispersion-correction method in the energy calculations? How large the supercell was used in the calculations?

3. The manuscript identifies that a uniform cyanide electron distribution within the crystal structure, maintain the stability of FeLS-CN-MHS coordination structure during the desodiation/sodiation process. Could the authors provide more quantitative data linking specific structural changes (e.g., lattice distortions, volume changes) during cycles with a descriptor of cyanide electron distribution? Or the quantitative relationship between the electron distributions and the observed capacity loss?

4. The manuscript shows Co and Ni doping improve the cycling stability performance of cathode materials (M4-PBA). Could the authors clarify whether the introduction of Co and Ni affects the electronic structure of Fe and Mn, and how does this impact the overall redox activity?

5. The authors should introduce in Section 1 the general cathode materials of SIBs. It is known that several cathode materials have been extensively explored for SIBs in recent years, including layered oxides, polyanionic compounds, and Prussian blue analogues (PBAs). Among them, as I known, P2 phases of Na₂Ni₂TeO₆ [see: ACS Appl. Energy Mater. 7, 8715–8725 (2024); Mater. Today Energy 37, 101414 (2023)] has also been extensively explored for SIBs recently with good electrochemical performances. A simple outline of cathode materials for SIBs would be essential to broaden the interest of readers and enhance the significance of the manuscript.

6. To obtain a full comment on the materials studied, a clear comparison in terms of electrochemical performance is needed for M2-PBA, M4-PBA, and M5-PBA. Could the authors provide a radar chart or a similarity to visually represent the trade-offs between energy density, rate performance, and cycle stability among M2-PBA, M4-PBA, and M5-PBA?

7. Minor issues: *The temperature at which the data shown in Fig. 3, S14-S16, S21 and S22 were measured should be given in the figure caption. <ii> For clarity and easy understanding, I suggest the authors use the left and right arrows to indicate, respectively, the curves for the capacity and Coulombic efficiency in Fig. 3b and 3g. <iii> The atoms for which the balls in different colors should be explained in Fig. 4e, 5g for clarity. <iv> The unit of density of states in Fig. 5a-c should be given in the title of the vertical axis. <v> In the "PBAs Preparation" section, the authors wrote: "5 g PVP (K30), 0.4 mol NaCl, and 0.2 g ascorbic acid were dissolved in deoxygenated deionized water". Why not to use the grams of NaCl here?*

Version 1:

Reviewer comments:

Reviewer #1

(Remarks to the Author)

We appreciate the modifications that has been made in this manuscript, all the additional and/or repeated experiments that has been conducted. We think that the overall quality of the manuscript has significantly increased, and a lot of doubtful points have been clarified. However, here are a few aspects regarding the new modification, that triggered our curiosity:

Response 1 : stoichiometry calculations

a. It was mentioned that EA was used, however the details about the instrumentation, or the experimental conditions are missing from the experimental part of the manuscript;

b. TGA and EA results about H₂O quantification does not match. However, they do not offer any explanations for this difference? Ultimately, in the obtained stoichiometric formula, the water content calculated from the TGA experiment was considered, and not from EA. Again, the explanation about this choice is not given?

c. Formula stoichiometry needs an additional check. With the backwards calculation from the offered stoichiometry, the errors for each element are substantial, sometimes over 10%

Response 2: XRD Rietveld refinement: the improvement in this portion is substantial, however it is not mentioned, after the defining initial occupancies of the elements from the stoichiometric data, if these values are fixed or varied during the refinement process (and why).

Response 5: Pre-edge spectra (Fig. S11a) show a very small difference among three. The conclusion that: "deviation from octahedral symmetry, which could be attributed to its higher sodium content" is a strong statement for such a tiny modification.

Response 6 (also 15 & 16): on our request to provide the spectra of EXAFS oscillations was answered by providing the plots of imaginary part of Fourier Transform signals. It's not clear why. We again would encourage the authors to provide the spectra of EXAFS oscillations.

Respond 8: the revised XPS spectrum of M5-PBA (fig. 2c) was a shoulder at Fe 2p_{3/2} peak (in the same area as for M2-PBA the Fe³⁺ contribution is offered), however, it is not fitted.

Response 9: With the reported stoichiometry, the values of theoretical capacity calculated by the authors has been rechecked. With M2-PBA, it was evident that the number of charge carrier was considered to be 2 (Na); and entire molecular mass of the material was taken (recalculation gave close enough match, with the negligible difference of ≈0.08 mAh g⁻¹). However, with M4-PBA and M5-PBA, results are not that clear. Their explanation states: "In PBAs cathodes, Ni and Cu coordinated to N are electrochemically inactive", and because of that they have decided to modify the theoretical capacity. But, if they have decided: a) to modify 2Na, and reduced its value by the "Ni and Cu portion", the values do not match, b) to subtract the portion of theoretical capacity, assigned to Ni and Cu, while still considering 2Na – the values again do not match. It would be helpful if they would provide the explanation and detailed calculation of the theoretical capacity obtained by the authors.

Reviewer #2

(Remarks to the Author)

Reviewer #3

(Remarks to the Author)

The work has been polished carefully and can be accepted in the current version.

Reviewer #4

(Remarks to the Author)

The authors have well-addressed all my concerns in the revised version of the manuscript. The authors have revised the related description in the "PBAs Preparation" section in the revised manuscript to supplement details and eliminate ambiguities. The authors have explained in the revised manuscript that they employed the PBE+U method primarily to address the strong electron correlation effects in the transition metal 3d orbitals, which is critical for accurately describing the electronic structure (e.g., bandgap and oxidation states) of their system and incorporated the van der Waals dispersion interactions into our calculations by employing the DFT-D3 dispersion correction method. The authors have reprocessed the in-situ and ex-situ XRD datasets to quantitatively evaluate the impact of uniform cyanide electron distribution on crystal stability through cyclic variations in unit cell volume during charge/discharge in the revised version of the manuscript. Some discussions on the effects of Co and Ni on electronic structure of Fe and Mn, and how does this impact the overall redox activity have been supplemented in the revised manuscript. Now the revised manuscript has been improved adequately and enough detail has been provided in the methods for the work to be reproduced. Therefore, I am very willing to recommend the current form of the revised manuscript for publication in Nature Communications.

Version 2:

Reviewer comments:

Reviewer #1

(Remarks to the Author)

We appreciate the continuous effort of authors are making in order to improve the manuscript, and thank them for the comprehensive response letter, with detailed explanations.

Regarding the calculation of the stoichiometry, the of extra technical information of EA, and the definition of the origin of water content calculation are positive additions. Unfortunately, one thing keeps being doubtful: the values about Fe HS/LS distribution, namely:

The EA results do not agree with the stoichiometric formula for C and N.

It is understandable, why the water content calculated from the EA is not reliable, but what about C and N? If the offered stoichiometry is correct, then the %wt of C&N should be different:

Sample: M2-PBA M4-PBA M5-PBA

Offered stoichiometry $\text{Na}_{1.84}\text{Mn}_{0.50}\text{Fe}_{0.50}[\text{Fe}(\text{CN})_6]_{0.89}\square_{0.11}\cdot 2.74\text{H}_2\text{O}$

$\text{Na}_{1.89}\text{Mn}_{0.27}\text{Fe}_{0.27}\text{Co}_{0.25}\text{Ni}_{0.21}[\text{Fe}(\text{CN})_6]_{0.91}\square_{0.09}\cdot 1.42\text{H}_2\text{O}$

$\text{Na}_{1.90}\text{Mn}_{0.23}\text{Fe}_{0.22}\text{Co}_{0.20}\text{Ni}_{0.18}\text{Cu}_{0.17}[\text{Fe}(\text{CN})_6]_{0.93}\square_{0.07}\cdot 1.14\text{H}_2\text{O}$

Mw (g mol⁻¹) according to offered stoichiometry: 335.71 318.90 319.43

C (%wt.) calculated from offered stoichiometry $(12.01 \times 0.89 \times 6) / 335.71 \times 100 = 19.10$ $(12.01 \times 0.91 \times 6) / 318.90 \times 100 = 20.56$
 $(12.01 \times 0.93 \times 6) / 319.43 \times 100 = 20.98$

N (%wt.) calculated from offered stoichiometry $(14.01 \times 0.89 \times 6) / 335.71 \times 100 = 22.29$ $(14.01 \times 0.91 \times 6) / 318.90 \times 100 = 23.99$
 $(14.01 \times 0.93 \times 6) / 319.43 \times 100 = 24.47$

C (%wt.) according to EA 17.48 18.25 19.07

N (%wt.) according to EA 20.39 21.29 22.24

Error (measur. Vs stoich.) 8.5% 11.2% 9.1%

Instead, if the EA measurement about the C & N are reliable (and H₂O is taken from TGA), then the Fe HS/LS ratio (and consequently entire formula, as it is normalised on HS metal sum to be 1), will be changed.

For M2-PBA:

Element Relative atomic (molecular) mass (g mol⁻¹) ICP-OES results (%wt)

(metal mass sum = 100%) EA (C & N) / TGA (H₂O) (%wt)

Recalculated (%wt) of metals

(entire mass = 100%) Normalised on Mw (mol) Normalised on HS metal sum to be 1

Na 22.99 28.7 $(28.7 \times (100 - (17.48 + 20.39 + 14.70))) / 100 = 13.61$ 0.59 0.59 / (0.16 + 0.21) = 1.62

Mn 54.94 18.64 $(18.64 \times (100 - (17.48 + 20.39 + 14.70))) / 100 = 8.84$ 0.16 0.16 / (0.16 + 0.21) = 0.44

Fe 55.85 52.67 $(52.67 \times (100 - (17.48 + 20.39 + 14.70))) / 100 = 24.98$ 0.45 0.45 - 0.16 = 0.21 (HS) 0.21 / (0.16 + 0.21) = 0.56
1.46/6 = 0.24 (LS) 0.24 / (0.16 + 0.21) = 0.66

C 12.01 - 17.48 1.46 1.46 / (0.16 + 0.21) = 3.98

N 14.01 - 20.39 1.46 1.46 / (0.16 + 0.21) = 3.98

H₂O 18.016 - 14.70 0.82 0.82 / (0.16 + 0.21) = 2.23

The obtained stoichiometry from this calculation is:
[Na] _1.62 [Mn] _0.44 [Fe] _0.56 [Fe(CN)_6] _0.66·2.23H_2O

With the molecular mass of: 273.50

Reverse calculation (for checking):

Water content:

Mw of H₂O H₂O (%wt)

273.50 (2×1.008+16) ×2.23=40.20 40.20/273.50×100=14.7

Carbon and Nitrogen:

Mw of C Mass of N %wt C %wt N

273.50 6×0.66×12.01=47.80 6×0.66×14.01=55.76 47.80/273.50×100=17.48 55.76/273.50×100=20.39

Note: the mass of C and N are calculated before rounding up the stoichiometric values (0.66338), that's why they slightly mismatch if the calculations are done on already rounded up stoichiometry.

Metals:

Mw of only metals %wt Na %wt Mn %wt Fe HS %wt Fe LS %wt Fe

273.50 273.50-40.20-47.80-55.76=129.73 (22.99×1.62)/129.73×100=28.70 (54.94×0.44)/129.73×100=18.64

(55.85×0.56)/129.73×100=24.11 (55.85×0.56)/129.73×100=28.56 52.67

Note: the masses are calculated before rounding up the stoichiometric values, that's why they slightly mismatch if the calculations are done on already rounded up stoichiometry.

If the calculations are done is this way, for M4-PBA and M5-PBA the stoichiometric formulas will be:

[Na] _1.60 [Mn] _0.23 [Fe] _0.38 [Co] _0.21 [Ni] _0.18 [Fe(CN)_6] _0.62·1.09H_2O

[Na] _1.65 [Mn] _0.20 [Fe] _0.32 [Co] _0.17 [Ni] _0.16 [Cu] _0.15 [Fe(CN)_6] _0.67·0.91H_2O

If the vacancy calculations are coming from EPR, then the information is scarce and needs more details. The authors only say: "The g values of ~2.03 for three samples from the electron paramagnetic resonance (EPR) tests indicates the presence of [Fe(CN)₆]⁴⁻ defect (Fig. S9). The amplitude difference between M4-PBA and M5-PBA in the curves is small. The higher amplitude for M2-PBA indicates more crystal defects of [Fe(CN)₆]⁴⁻ in comparison with M4-PBA and M5-PBA". It seems, the described vacancy assessment from EPR is approximate (only qualitative, and not quantitative), so this should be acknowledged in the text.

If the stoichiometric calculations are done according to the different assumptions/thought process, it has to be explained, as the difference is substantial.

Regarding the theoretical capacity calculations, we thank the authors for the detailed explanation. The calculation of the theoretical capacity is almost always conducted via making assumptions about the system, so it was beneficial to understand how authors reached these values.

In the response letter, authors stated: "However, not all MHS enable reversible +2/+3 redox. Ni and Cu ions preferentially maintain +2 oxidation states during cycling (Adv. Mater. 2024, 2405458), limiting Na⁺ storage to 1 mol via FeLS redox alone, with theoretical capacities of 84.64 and 83.36 mAh g⁻¹ for Ni-based and Cu-based PBAs, respectively".

The transition metal redox activity in PBAs depend on many aspects, and while there are quite a few examples of Ni and Cu not being redox active, the opposite can also be found. Such as:

<https://doi.org/10.1002/adma.202419446> (Ni),

<https://doi.org/10.1021/acs.jpcc.8b03429> (Cu),

<https://doi.org/10.1016/j.ensm.2020.08.008> (Cu),...

We are not stating that Ni and/or Cu are necessarily electrochemically active in the compounds described in current manuscript, but to state the opposite, the proof must be demonstrated (such as XAS, for example).

Reviewer #2

(Remarks to the Author)

Response to reviewers' comments

We sincerely thank the reviewers for their constructive comments and valuable suggestions, which have greatly contributed to improving the quality of the manuscript. In response to these comments and suggestions, we have conducted additional experiments and made comprehensive revisions to the original manuscript. Below are our responses (in blue font) to the comments. To be clear, all changes in the revised manuscript and supporting information are highlighted by red font.

Reviewer #1 (Remarks to the Author):

Manuscript background information

This paper investigates the challenges associated with utilizing Prussian blue analogues (PBAs) as cathode materials for sodium-ion batteries, particularly addressing issues such as structural instability and capacity degradation due to transition metal dissolution and Jahn-Teller distortion. While the topic is of interest, the cause-and-effect relationships are not clearly established in the manuscript, and a more complete data analysis is required. Although the focus is on the coordination electronic structure—a concept that is not well defined within the paper—it remains unclear how the three proposed structures effectively support the hypothesis presented. Furthermore, the discussion of the localized electronic structure is based solely on DFT calculations, and a more robust experimental validation would be highly beneficial. Additionally, the experimental section lacks sufficient detail to adequately support the findings.

We sincerely appreciate the reviewer's thought-provoking comments. Regarding the concerns raised—such as insufficient clarity in causal relationships, improvements for enhanced data analysis, and the need for expanded experimental details—we agree that supplemental experiments to substantiate local coordination structure evolution and deeper analysis of existing data are essential to strengthen the scientific narrative. To ensure the manuscript presents well-supported arguments with compelling evidence, we have addressed each point raised in the reviewer's feedback with detailed responses below.

Specific major questions

1. Line 116-117, table S1: stoichiometry calculation – how from only ICP-OES and TGA was it possible to determine the ratio between high and low spin Fe? And therefore, define the coefficient of [Fe(CN)₆] unit?

Response 1: We appreciate the reviewer for highlighting the calculations in determining the stoichiometry of Prussian blue analogues (PBAs). In fact, precise determination of their chemical formulas—particularly quantifying the contents of high-spin Fe ions (Fe^{HS}) and low-spin Fe ions (Fe^{LS})—remains a persistent difficulty in the field.

Typically, the general formula of PBAs is $\text{Na}_x\text{M}^{\text{HS}}[\text{Fe}^{\text{LS}}(\text{CN})_6]_y \cdot \square_{1-y} \cdot z\text{H}_2\text{O}$, where M^{HS} and Fe^{LS} respectively represent high-spin transition metal ions and low-spin Fe ions, \square means vacancy defects of $[\text{Fe}^{\text{LS}}(\text{CN})_6]$, and x, y, z are stoichiometric coefficients for Na^+ , $[\text{Fe}^{\text{LS}}(\text{CN})_6]$ and H_2O , respectively. During synthesis, intrinsic crystal defects—including $[\text{Fe}^{\text{LS}}(\text{CN})_6]$ vacancies and crystalline water—are unavoidably incorporated. Although inductively coupled plasma optical emission spectrometry (ICP-OES) accurately quantifies total Fe ion content as well as other metal ions, it cannot distinguish between Fe^{HS} and Fe^{LS} . Fortunately, four critical stoichiometry calculation relationships established by structure characterizations enable precise formula determination of PBAs, as outline below: (1)

The content of each metal ion in PBAs can be determined by ICP-OES, enabling the determination of the molar ratio of each metal element. (2) Based on the mass loss and mass retention rate from thermogravimetric analysis (TGA), the mass fractions of H₂O and Na_xM^{HS}[Fe^{LS}(CN)₆]_y can be distinguished. (3) The elemental analyzer (EA) can be used to measure the content of non-metallic elements, determining the molar ratio of C in Na_xM^{HS}[Fe^{LS}(CN)₆]_y and O in H₂O. (4) In PBAs, the coordination relationship between Fe and C ensures that the molar ratios of C and Fe^{LS} are exactly 6:1. Thus, this methodology—combining ICP-OES for metal ion content, TGA for H₂O content and EA for C/O ratios—is widely adopted by researchers (*J. Electrochem. Soc.* 2016, 163, A2117-A2123.).

As raised by Reviewer #1, the accuracy of PBA formulas directly impacts conclusions about cyanide-bridged electronic structures. We have now supplemented EA measurements and retested ICP-OES data in order to recalculate precise formulas of PBAs as well as quantifying the contents of Fe^{HS} and Fe^{LS}. This revision ensures accurate correlation between stoichiometry and electronic properties in our analysis. The EA (Tables S1) and ICP-OES (Table S2) results are updated as followed:

Revised Table S1 The result of EA for PBAs (Supporting Information)

Sample	C (%)	N (%)	H (%)	O (%)
M2-PBA	17.48	20.39	0.90	7.15
M4-PBA	18.25	21.29	0.80	6.32
M5-PBA	19.07	22.24	0.65	5.19

Revised Table S2 The result of ICP-OES for PBAs (Supporting Information)

Sample	Na (%)	Mn (%)	Fe (%)	Co (%)	Ni (%)	Cu (%)
M2-PBA	28.70	18.64	52.67	/	/	/
M4-PBA	28.73	9.81	43.57	9.74	8.15	/
M5-PBA	28.42	8.22	41.79	7.67	6.87	7.03

Consequently, by using the comprehensive ICP-OES + TGA + EA approach, the chemical formulas of the three PBAs were recalculated. The corresponding discussions of PBA chemical formulas in the manuscript were revised to:

“Thermogravimetric analysis (TGA) in Fig. S10, EA (Table S1) and ICP-OES tests (Table S2) confirm that the chemical formulas of M2-PBA, M4-PBA, and M5-PBA are Na_{1.84}Mn_{0.50}Fe_{0.50}[Fe(CN)₆]_{0.89□0.11}·2.74H₂O, Na_{1.89}Mn_{0.27}Fe_{0.27}Co_{0.25}Ni_{0.21}[Fe(CN)₆]_{0.91□0.09}·1.42H₂O and Na_{1.90}Mn_{0.23}Fe_{0.22}Co_{0.20}Ni_{0.18}Cu_{0.17}[Fe(CN)₆]_{0.93□0.07}·1.14H₂O, respectively, where □ denotes [Fe(CN)₆] vacancies in the crystal lattice³⁰.” (Page 6 Line 123-128)

30 Yang, Y. et al. Influence of structural imperfection on electrochemical behavior of Prussian blue cathode materials for sodium ion batteries. *J. Electrochem. Soc.* **163**, A2117-A2123 (2016).

2. Line 130, table S2-S4: the occupancy inside the refinement parameters does not match with stoichiometry, considers equimolar distribution of all high spin metals (and 2Na), and doesn't consider vacancies that were included in the formulae (maybe it could be mentioned, that within the quality of the data, further accuracy couldn't be reached). Also, the rationale for the selected three materials is not clear. M in the general formula is one metal. Why the authors synthesized different materials by substituting the same (potential) crystalline site? This is not suggested by the DFT calculated structure reported previously.

Response 2: We sincerely appreciate the reviewer's critical suggestions and apologize for not performing Rietveld refinement using the actual stoichiometry of the PBAs. As rightly noted by the reviewer, discrepancies between the elemental occupancy rates and the true PBA stoichiometry not only introduce significant errors in XRD refinement results but also compromise the analysis of cyanide-bridged electronic structures. Below, we provide point-by-point responses to the specific questions raised by the reviewer.

(1) Revision of elemental occupancy and structural Rietveld refinement

Based on the revised chemical formulas of PBAs, the atomic occupancies of corresponding elements have been adjusted. Structural refinements for the three PBA materials were re-performed with explicit consideration of vacancy defects, aiming to obtain more accurate atomic coordinates and electron density distributions. These updates ensure a more accurate representation of the structural and electronic properties of the materials, aligning with the refined experimental and computational data. The labels of tables containing atomic coordinates from XRD refinement have been modified in the manuscript:

“The crystallographic information for the three samples is detailed in Table S3-S5.” (Page 6 Line 134-135)

The refined PXRD patterns and corresponding cross-sectional electronic structure analyses have been updated in Fig. 1k-m and their insets.

Revised Fig. 1 Rietveld refinement PXRD patterns with the schematic of the cross-sectional

electronic structure shown in the inset: k M2-PBA; l M4-PBA; m M5-PBA. (Page 7 Line 146 and Line 150-151)

The revised atomic coordinates for the three PBAs are now presented in Tables S3-S5 of the supporting information.

Revised Table S3 Rietveld refined result of M2-PBA (Supporting Information)

M2-PBA Monoclinic, space group $P21/n$,
 $a = 10.54505 \text{ \AA}$, $b = 7.50288 \text{ \AA}$, $c = 7.27566 \text{ \AA}$, $V = 575.091 \text{ \AA}^3$, $\alpha = \gamma = 90^\circ$, $\beta = 92.498^\circ$

Atom	x	y	z	Occ.	Site
N1	0.0048	0.1959	0.1950	0.89	4e
N2	0.3005	0.5014	0.5101	0.89	4e
N3	0.0000	0.8081	0.2617	0.89	4e
C1	0.0050	0.3026	0.3130	0.89	4e
C2	0.1837	0.4998	0.5003	0.89	4e
C3	0.0037	0.6829	0.2998	0.89	4e
Na	0.2488	0.5154	0.0263	0.92	4e
Fe1	0.0000	0.5000	0.5000	0.50	2d
Mn	0.0000	0.0000	0.0000	0.50	2a
Fe2	0.0000	0.0000	0.0000	0.89	2a

Revised Table S4 Rietveld refined result of M4-PBA (Supporting Information)

M4-PBA Monoclinic, space group $P21/n$,
 $a = 10.49118 \text{ \AA}$, $b = 7.48657 \text{ \AA}$, $c = 7.24433 \text{ \AA}$, $V = 568.430 \text{ \AA}^3$, $\alpha = \gamma = 90^\circ$, $\beta = 92.545^\circ$

Atom	x	y	z	Occ.	Site
N1	0.0048	0.1988	0.1967	0.91	4e
N2	0.3019	0.5042	0.5015	0.91	4e
N3	0.0033	0.8060	0.2094	0.91	4e
C1	0.0054	0.3036	0.3231	0.91	4e
C2	0.1833	0.5179	0.5169	0.91	4e
C3	0.0032	0.6839	0.3007	0.91	4e
Na	0.2289	0.4991	0.0236	0.945	4e
Fe1	0.0000	0.5000	0.5000	0.91	2d
Mn	0.0000	0.0000	0.0000	0.27	2a
Fe2	0.0000	0.0000	0.0000	0.27	2a
Co	0.0000	0.0000	0.0000	0.25	2a

Ni	0.0000	0.0000	0.0000	0.21	2a
----	--------	--------	--------	------	----

Revised Table S5 Rietveld refined result of M5-PBA (Supporting Information)

M5-PBA Monoclinic, space group $P21/n$,					
$a = 10.3043 \text{ \AA}$, $b = 7.4024 \text{ \AA}$, $c = 7.1707 \text{ \AA}$, $V = 546.435 \text{ \AA}^3$, $\alpha = \gamma = 90^\circ$, $\beta = 92.501^\circ$					
Atom	x	y	z	Occ.	Site
N1	0.0053	0.1997	0.1958	0.93	4e
N2	0.3011	0.5043	0.5026	0.93	4e
N3	0.0029	0.8054	0.2068	0.93	4e
C1	0.0062	0.3035	0.3244	0.93	4e
C2	0.1865	0.5271	0.5296	0.93	4e
C3	0.0034	0.6840	0.3001	0.93	4e
Na	0.2621	0.5389	0.0053	0.95	4e
Fe1	0.0000	0.5000	0.5000	0.93	2d
Mn	0.0000	0.0000	0.0000	0.23	2a
Fe2	0.0000	0.0000	0.0000	0.22	2a
Co	0.0000	0.0000	0.0000	0.20	2a
Ni	0.0000	0.0000	0.0000	0.18	2a
Cu	0.0000	0.0000	0.0000	0.17	2a

(2) Electronic structure analysis of PBAs and conceptual validation of electronic modulation

As discussed in the Introduction, the electronic structure of the $\text{Fe}^{\text{LS}}\text{-C}\equiv\text{N-M}^{\text{HS}}$ coordination framework is influenced by the selection of N-coordinated transition metal ions (M^{HS}), due to differences in electronic configurations of M^{HS} . To systematically investigate this, DFT calculations were performed to visualize the electronic structures of single M^{HS} -based PBAs (Fig. 1a–j). Mn-PBA: Coordinated electrons of $\text{Fe}^{\text{LS}}\text{-C}\equiv\text{N-M}^{\text{HS}}$ preferentially localize near high-spin Mn^{2+} ; Fe-PBA: Electron distribution of cyanide distributes evenly between high-spin Fe^{2+} and low-spin Fe^{2+} ; Co-PBA/Ni-PBA: Electrons shift toward low-spin Fe^{2+} ; Cu-PBA: Electron distribution of cyanide transitions from biasing to high-spin Cu^{2+} into biasing to low-spin Fe^{2+} during Na^+ insertion process. Based on these findings, a simple additive design concept was adopted for selecting and synthesizing equimolar multi-component PBAs (Mn/Fe binary, Mn/Fe/Co/Ni quaternary, and Mn/Fe/Co/Ni/Cu quinary systems), which were expected to conceptually validated how variations in coordinated electron distributions within the $\text{Fe}^{\text{LS}}\text{-C}\equiv\text{N-M}^{\text{HS}}$ framework affect the electrochemical performance of PBAs. To reflect the design rationale of the three PBA materials, we have provided supplementary elaboration in the manuscript:

“To conceptually validate the electronic modulation of the $\text{Fe}^{\text{LS}}\text{-C}\equiv\text{N-M}^{\text{HS}}$ framework, three PBAs samples with varying M^{HS} compositions were synthesized via an optimized co-precipitation

method. To systematically probe the influence of different M^{HS} ions on the $Fe^{LS}-C\equiv N-M^{HS}$ electronic structure, the M^{HS} components in each PBA were introduced at equimolar ratios during synthesis.” (Page 5 Line 105-109)

(3) Commonly simplified treatment in XRD Rietveld refinement

In PBA crystals, M^{HS} universally denotes the high-spin transition metals. The coordination structures of $Fe^{LS}-C\equiv N-M^{HS}$ collectively constitute the PBA coordination framework. As previously discussed, distinct M^{HS} elements induce subtle variations in bond lengths and angles within this framework. During XRD Rietveld refinement, the crystallographic coordinates for all M^{HS} atoms were assigned to a single site to accommodate crystallographic symmetry constraints. This simplification avoids potential refinement inaccuracies arising from minute differences in crystallographic positions among different M^{HS} elements. In contrast, for computational modeling in Material Studio, each atomic site can only host one M^{HS} species. To resolve this, we constructed a $2\times 2\times 1$ supercell to assign independent crystallographic sites for each M^{HS} type. However, during density functional theory (DFT) calculations for the three PBAs, the large number of atoms in realistic PBA structures made computations prohibitively expensive. Consequently, simplified models were employed to ensure computational tractability. Detailed DFT parameters are supplemented in the Methods section:

“For the single M^{HS} -based PBAs, due to their high symmetry, the unit cells for computational modeling were employed directly without supercell construction (Fig. 1a-j). Depending on the phase structure differences, the number of atoms per unit cell varies. For cubic phase (desodiated state), there are 58 atoms in the unit cell, while 48 atoms in the unit cell of rhombohedral phase (sodiated state). As to the density functional theory (DFT) calculations of the actual Prussian blue analog (PBA) structures, we constructed a $2\times 2\times 1$ supercell containing 192 atoms to closely match the elemental ratios observed in the experimentally synthesized PBA crystals.” (Page 23-24 Line 565-572)

3. Line 137. The sentence “Since Fe^{2+} is introduced in both high-spin sites and low-spin sites for all the three samples” is not clear. Please specify.

Response 3: We appreciate the reviewer for pointing out the lack of clarity in the manuscript’s discussion. According to the suggestions, we have revised the relevant passages accordingly. The modifications are as follows:

“Since both $Fe-N$ and $Fe-C$ coordination bonds exist in all three samples, various structural characterization techniques focused on the Fe element were employed to investigate the effect of the electronic structural modulation on cyanide coordination structure in PBAs.” (Page 7 Line 153-155)

4. Line 142-143, fig. 2a: it's written "M4-PBA exhibits a relatively lower near edge absorption energy, indicating a less oxidation of Fe²⁺", however in the figure 2a inset, M5-PBA looks more shifted towards the lower energy than M4-PBA;

Response 4: Thank you for the reviewer's attentive suggestion. The term "M4-PBA" in Line 142 was a typographical error and should be revised to "M5-PBA". We apologize for this oversight and have carefully reviewed the entire manuscript for related typographical errors of "M4-PBA". The relative revisions are itemized below:

"The X-ray absorption near edge structure (XANES) spectra of the three samples show minimal differences, in which M5-PBA exhibits a relatively lower near edge absorption energy, indicating a less oxidation of Fe²⁺ (Fig. 2a)." (Page 7 Line 157-159)

"An increase in the pre-edge peak intensity of M5-PBA suggests a slight deviation from octahedral symmetry due to a higher sodium content (Fig. S11a)³⁵." (Page 7 Line 159-161)

"Table S5 Rietveld refined result of M5-PBA" (Supporting Information)

35 Li, M. et al. Influence of vacancies in manganese hexacyanoferrate cathode for organic Na-ion batteries: a structural perspective. *ChemSusChem* **16**, e202300201 (2023).

5. Line 143-145: it is written: "An increase in the pre-edge peak intensity of M4-PBA suggests a slight deviation from octahedral symmetry due to a higher sodium content", however later the authors are always speaking about M4-PBA having highest symmetry among three, which sounds contradictory. Also, better to add extra figure or inset of pre-edge, because in fig 2a, it is really hard to see any difference;

Response 5: Thank reviewer for raising the important questions. The term "M4-PBA" in Line 144 was a typographical error and the corresponding revision related to this point can be found in **Respon**s 4. As rightly suggested by the reviewer, providing detailed pre-edge comparison plots is essential. This significantly contributes to understanding how electronic structure modulation enhances structural symmetry in PBAs. In the supplemented pre-edge spectra (Revised Fig. S11a), M5-PBA exhibits the most pronounced pre-edge feature while M4-PBA shows the weakest, indicating that M5-PBA may experience subtle deviation from octahedral symmetry, which could be attributed to its higher sodium content. The supplemented pre-edge spectra are as followed:

Supplemented Fig. S11a pre-edge of K-edge Fe for M2-PBA, M4-PBA and M5-PBA. (Supporting Information)

6. Line 146, fig 2b: better to mention that it is Fourier transform and not directly EXAFS spectra, also better to additionally show the actual spectra (oscillations), do give the reliability to the authors claims about the “different coordination structures of central Fe atom for the three samples”;

Response 6: We appreciate the reviewer’s valuable suggestion. According to the suggestion on description of terminological precision, we have revised the relevant expression to “Fourier-transformed EXAFS (FT-EXAFS)”. The related discussions and figures have been revised:

“The distinct profiles of Fourier-transformed EXAFS (FT-EXAFS) in Fig. 2b with their imaginary parts (Fig. S11b) exhibit different coordination structures of central Fe atom for the three samples, correspondingly revealing the coordination environment of Fe–C and Fe–N shells in PBAs.” (Page 7-8 Line 161-164)

Revised “Fig. 2b FT-EXAFS” (Page 9 Line 203)

“As to the FT-EXAFS result of M2-PBA,” (Page 15 Line 371-372)

“In detail, both the profile intensity and position undergo significant changes in the FT-EXAFS curves and corresponding imaginary parts (Fig. S31),” (Page 16 Line 373-374)

“In the case of M5-PBA, although the FT-EXAFS profile of the Fe–N coordination shell is highly reversible,” (Page 16 Line 375-376)

“In contrast, the stable cyanide coordination electronic structure in M4-PBA is verified by the highly consistent FT-EXAFS curves of the Fe–C and Fe–N coordination shells at different sates of charge,” (Page 16 Line 378-380)

“Similarly, Mn K-edge XANES and corresponding FT-EXAFS curves were collected and analyzed (Figs. S32-S33).” (Page 16 Line 381-382)

“The irreversible change of coordination structure is consistent with previous ex-situ FT-EXAFS results (Fig. S30).” (Page 16 Line 399-400)

Revised “Fig. 4c FT-EXAFS” (Page 18 Line 417)

Revised Fig. 2b FT-EXAFS (Page 9 Line 202)

Revised Fig. 4c FT-EXAFS (Page 17 Line 415)

Additionally, to give the reliability to “different coordination structures of central Fe atom for the three samples”, the corresponding imaginary parts of EXAFS signals are now presented in Figure S11b. The augmented discussion and supplementary experimental spectral data are revised below:

“The distinct profiles of Fourier-transformed EXAFS (FT-EXAFS) in Fig. 2b with their imaginary parts (Fig. S11b) exhibit different coordination structures of central Fe atom for the three samples, correspondingly revealing the coordination environment of Fe–C and Fe–N shells in PBAs.” (Page 7-8 Line 161-164)

Supplemented Fig. S11b imaginary parts of EXAFS for Fe K-edge. (Supporting Information)

7. Line 149. The sentence “The higher profile intensity for both Fe-C and Fe-N coordination bonds reflects the fine structural symmetry for M4-PBA” is lack of clarity and meaning. Peak intensity in the FT EXAFS are not due to symmetry, nor to the Fe-N is a direct bond.

Response 7: Thank you for the reviewer’s correction regarding the analysis of FT-EXAFS results. We apologize for the ambiguity in the conceptual description of coordination bonds. It should be mentioned that the profile intensities of Fe-C and Fe-N coordination shells do not directly represent coordination structural symmetry, they reflect the change of the local coordination structure (*Nat. Commun.* 2024, 15, 7371; *ACS Appl. Energy Mater.* 2023, 6, 8607-8615), which correlates with the change in structural symmetry of coordination shells to some extent (*Adv. Energy Mater.* 2024, 14, 2400875). Thus, the changes of profile intensity and coordination shell radial distance reflect the evolution of coordination frameworks in PBAs (*J. Am. Chem. Soc.* 2000, 122, 28, 6653-6658; *Small* 2024, 20, 2404584). Accordingly, in order to make the FT-EXAFS analysis results more rigorous,

the related statements have been revised:

“The splitting of Fe–N profiles in all samples indicates the influence of M^{HS} on the Fe–N coordination shell. The higher profile intensity for both Fe–C and Fe–N coordination shells reflect an ideal coordination environment for M4-PBA³⁶.” (Page 8 Line 164-166)

“As to the FT-EXAFS result of M2-PBA, the profiles of Fe–C and Fe–N coordination shells change obviously from SOC-1 to SOC-3 (Fig. S30a).” (Page 15-16 Line 371-373)

“In the case of M5-PBA, although the FT-EXAFS profile of the Fe–N coordination shell is highly reversible, the shift of the cyanide electron clouds towards to C leads to the stability reduction of Fe^{LS}–C coordination environment (Fig. S30b).” (Page 16 Line 375-378)

“In contrast, the stable cyanide coordination electronic structure in M4-PBA is verified by the highly consistent FT-EXAFS curves of the Fe–C and Fe–N coordination shells at different sates of charge, indicating the good coordination structural reversibility during cycling (Fig. 4c).” (Page 16 Line 378-381)

“Benefiting from the uniform electronic distribution of Fe^{LS}–C≡N–M^{HS}, the profiles of the Mn–N and Mn–C coordination shells for M4-PBA exhibit smaller variations.” (Page 16 Line 382-384)

36 Xie, B. X. et al. Achieving long-life Prussian blue analogue cathode for Na-ion batteries via triple-cation lattice substitution and coordinated water capture. *Nano Energy* **61**, 201-210 (2019).

8. Line 164, fig 2c: for the XPS spectra fitting for M5-PBA, the Fe+3 contribution for Fe 2p_{3/2} is visible, however, Fe 2p_{1/2} it is not; also, in M4-PBA, there seems to be a satellite around 715 eV, which is not assigned or fitted; Please explain;

Response 8: Thank you for the reviewer’s suggestion regarding the XPS analysis. The XPS spectra of the three PBAs have been re-measured and deconvoluted, and revised Fig. 2c has been updated accordingly. In the revised figure, the spectra are arranged from bottom to top as follows: M2-PBA, M4-PBA, and M5-PBA. The colors for spectral components are reassigned. Light green and light blue represent Fe²⁺ 2p_{3/2} and 2p_{1/2} peaks, respectively. Yellow and purple represent Fe³⁺ 2p_{3/2} and 2p_{1/2} peaks, respectively. Dark green and dark blue are satellite peaks corresponding to 2p_{3/2} and 2p_{1/2}, respectively. The revised Fig. 2c is as followed:

9. Line 220. Not clear why M4 sample displays the highest proportion of theoretical capacity

Response 9: Thank you for the reviewer's concern regarding on the calculation of theoretical capacity proportion. In PBAs cathodes, Ni and Cu coordinated to N are electrochemically inactive, while Mn, Fe, Co coordinated to N and Fe coordinated to C is electrochemically active (*Adv. Mater.* 2024, 2405458). According to the revised chemical formula of PBAs, the theoretical capacities of M2-PBA ($\text{Na}_{1.84}\text{Mn}_{0.5}\text{Fe}_{0.5}[\text{Fe}(\text{CN})_6]_{0.89} \cdot 0.11 \cdot 2.74\text{H}_2\text{O}$), M4-PBA ($\text{Na}_{1.89}\text{Mn}_{0.27}\text{Fe}_{0.27}\text{Co}_{0.25}\text{Ni}_{0.21}[\text{Fe}(\text{CN})_6]_{0.91} \cdot 0.09 \cdot 1.42\text{H}_2\text{O}$) and M5-PBA ($\text{Na}_{1.90}\text{Mn}_{0.23}\text{Fe}_{0.22}\text{Co}_{0.20}\text{Ni}_{0.18}\text{Cu}_{0.17}[\text{Fe}(\text{CN})_6]_{0.93} \cdot 0.07 \cdot 1.14\text{H}_2\text{O}$) are 159.76 mAh g⁻¹, 144.85 mAh g⁻¹ and 135.69 mAh g⁻¹, respectively. The practical specific capacity of the three PBAs is 142.7 mAh g⁻¹, 142.4 mAh g⁻¹ and 109.5 mAh g⁻¹, respectively. Consequently, the proportion of theoretical capacity can be calculated as 89.32% for M2-PBA, 98.31% for M4-PBA, and 80.70% for M5-PBA. Since the chemical formulas for the three PBAs have been updated, the proportions of theoretical capacity have been correspondingly revised:

“The initial discharge capacities of M2-PBA, M4-PBA and M5-PBA are 142.7, 142.4 and 109.5 mAh·g⁻¹, corresponding to 89.32%, 98.31% and 80.70% of their theoretical capacities, respectively.” (Page 10-11 Line 235-237)

10. Line 226-228, fig. S15: the voltage is divided in two regions below and above 3.5 V; but why exactly 3.5 V? it is not the plateau position, neither the position in-between the peaks on dQ/dV curve, also doesn't help visually the statement about M4_PBA: “more symmetrically redox doublet peaks”;

Response 10: We sincerely appreciate the reviewer's insightful suggestions. As rightly noted by the reviewer, we have redefined the redox contributions of M^{HS} and Fe^{LS} based on the voltage gap between the two peaks in the dQ/dV curve of M4-PBA. Therefore, we have revised the threshold voltage from “3.5 V” to “3.25 V”. This modification enables clearer distinction of the capacity contributions from M^{HS} and Fe^{LS}, with specific revisions detailed below.

“Therefore, M2-PBA demonstrates a greater capacity contribution from Fe^{LS} in the high voltage region (over 3.25V), while M5-PBA shows a higher redox activity from M^{HS} in the low voltage region (below 3.25V)⁴⁴.” (Page 11 Line 243-245)

44 Huang, Y. et al. Boosting the sodium storage performance of Prussian blue analogs by single-crystal and high-entropy approach. *Energy Stor. Mater.* **58**, 1-8, (2023).

Additionally, to prevent potential ambiguity in the statement “more symmetrically redox doublet peaks”, we have modified the relevant discussion in the manuscript.

“As for M4-PBA, the dQ/dV profiles show symmetrical redox doublets after the first cycle

because of even distribution of cyanide electrons between M^{HS} and Fe^{LS} .” (Page 11 Line 245-247)

11. Line 313: A mitigation of the JT effect is stated here, but it is not substantially proven.

Response 11: We appreciate the reviewer’s valuable suggestion. In this section, we employed in-situ FT-IR to investigate the changes in the absorption peaks of the cyanide functional group for PBAs during the charge/discharge processes, thereby elucidating the influence of electronic structure modulation in $Fe^{LS}-C\equiv N-M^{HS}$ on the evolution of PBA crystal structure. Although the Jahn-Teller (J-T) effect is a predominant factor affecting the structure stability of PBA crystal structure (*Angew. Chem. Int. Ed.* 2021, 60, 18519-18526; *Angew. Chem. Int. Ed.* 2023, 62, e202217761), as you rightly pointed out, we indeed lack direct evidence to substantiate the mitigation of the J-T effect. Therefore, to ensure accuracy, we delete the description on the Jahn-Teller effect. To address this issue, we have made corresponding modifications in the manuscript:

“M4-PBA shows higher structural reversibility than M2-PBA and M5-PBA due to the regulation of cyanide electronic structure, thus maintaining higher cycling stability.” (Page 14 Line 330-331)

12. Line 316-319: it is a little unclear.

Response 12: We appreciate the reviewer for highlighting the lack of clarity in our in-situ FT-IR discussion. This section primarily focuses on the evolution of FT-IR peak area ratios during charge/discharge cycles. Since the emergence of peak 2 corresponds to the oxidation of Fe^{LS} , the ratio of peak 1 to peak 2 quantitatively reflects the respective capacity contributions from M^{HS} and Fe^{LS} . A higher area ratio indicates that the activity of M^{HS} is more easily activated compared to Fe^{LS} , and vice versa. This is related to the distribution of $Fe^{LS}-C\equiv N-M^{HS}$ electronic structure. Furthermore, the reversible variations in absorption peak positions and normalized areas between sodiated and desodiated states signify the reversible structural evolution in PBAs (*ACS Appl. Mater. Interfaces* 2019, 11, 46705-46713; *Nano Energy* 2019, 61, 201-210).

Thus, following the reviewer’s suggestion, we have substantially strengthened this discussion, which now more clearly articulates how uniform electron distribution enhances both capacity and cycling stability in PBAs. Key revisions in the manuscript include:

“Since the peak areas of peak 1 and peak 2 at the fully charged state partially reflect the redox activity of M^{HS} and Fe^{LS} , respectively, quantitative analysis of the peak area ratio (peak 1/peak 2) provides critical insights into the influence of uniform electronic distribution of $Fe^{LS}-C\equiv N-M^{HS}$ on redox activity utilization of Fe^{LS} and M^{HS} . The increased peak area ratio indicates inefficient utilization of Fe^{LS} redox activity, whereas the decreased ratio demonstrates inferior activation of M^{HS} . For M2-PBA, the relatively small peak area ratio value suggests that cyanide electron delocalization toward M^{HS} partially hinders its activity. M5-PBA exhibits a larger peak area ratio at full charged state, signifying restricted Fe^{LS} activity. Moreover, a linearly varying area ratio

throughout the second cycling process is shown in M4-PBA, indicating a stable Na^+ insertion and extraction process due to the uniform electronic distribution of $\text{Fe}^{\text{LS}}\text{-C}\equiv\text{N-M}^{\text{HS}}$ coordination structure.” (Page 14-15 Line 333-343)

13. Line 333-336. It is suggested that Fe dissolution might occur based on IPC-OES, but an explanation is missing;

Response 13: We sincerely appreciate the reviewer’s suggestion to strengthen the analysis of iron dissolution mechanisms. We agree that further clarification on the causes of Fe dissolution is essential, as it directly supports the argument that electronic modulation of the $\text{Fe}^{\text{LS}}\text{-C}\equiv\text{N-M}^{\text{HS}}$ coordination framework enhances the structural stability of PBAs. Accordingly, the following explanation has been added to the manuscript:

“However, the ICP-OES analysis of the three samples reveals the Fe dissolution concentration exceeding Mn by the order of magnitude during cycling. This phenomenon mainly stems from the destabilization of the inner-orbital $\text{Fe}^{\text{LS}}\text{-C}$ coordination structure induced by disruption of the outer-orbital $\text{M}^{\text{HS}}\text{-N}$ coordination structure within the PBA coordination framework^{26,44}. Furthermore, as the predominant transition metal across all three PBAs, Fe exhibits the highest dissolution tendency. The lower amount of transition metal dissolution not only signifies reduced active site loss but also indicates superior preservation of the $\text{Fe}^{\text{LS}}\text{-C}\equiv\text{N-M}^{\text{HS}}$ coordination framework.” (Page 15 Line 353-360)

26 Lee, J. H. et al. Effect of the interaction between transition metal redox center and cyanide ligand on structural evolution in Prussian white cathodes. *ACS Nano* **18**, 1995-2005 (2024).

44 Huang, Y. et al. Boosting the sodium storage performance of Prussian blue analogs by single-crystal and high-entropy approach. *Energy Stor. Mater.* **58**, 1-8, (2023).

14. Line 337. What “cyanide coordination” stands for?

Response 14: Thank you for the reviewer’s comments. In PBAs, the coordination framework is formed through cyanide ligands bridging between M^{HS} and Fe^{LS} . Thus, the term “cyanide coordination structure” specifically refers to the “ $\text{Fe}^{\text{LS}}\text{-C}\equiv\text{N-M}^{\text{HS}}$ coordination structure”. While both expressions have been used in prior sections to describe the local coordination environment of PBAs, we have revised this term to “ $\text{Fe}^{\text{LS}}\text{-C}\equiv\text{N-M}^{\text{HS}}$ coordination structure”, which is more appropriate to the discussion of EXAFS analysis. Additionally, we have provided an explanation at the first occurrence of “cyanide coordination structure”. The relevant discussion has been revised in the manuscript:

“By selecting transition metals preferring to form stronger $\text{M}^{\text{HS}}\text{-N}$ bonds, the stability of cyanide coordination structure ($\text{Fe}^{\text{LS}}\text{-C}\equiv\text{N-M}^{\text{HS}}$) will be effectively improved²².” (Page 4 Line 62-63)

“Ex-situ Fe K-edge XANES and EXAFS tests were further conducted to probe the evolution in

the $\text{Fe}^{\text{LS}}\text{-C}\equiv\text{N-M}^{\text{HS}}$ coordination structure during the Na^+ insertion and extraction process (Fig. 4b-c).” (Page 15 Line 362-364)

22 Luo, Y. et al. Inhibiting the Jahn-Teller effect of manganese hexacyanoferrate via Ni and Cu codoping for advanced sodium-ion batteries. *Adv. Mater.* **36**, 2405458 (2024).

15. Line 343-345: why the authors think that the larger shift in XANES of M2-PBA means the “ $\text{Fe}^{\text{LS}}\text{-CN-M}^{\text{HS}}$ coordination structure is taken apart”? Also, why only Fe K-edge data are presented? What’s about the other M K-edge?

Response 15: We appreciate the reviewer’s comments and apologize for the lack of precision in our original conclusion. While significant deviations in XANES spectra generally indicate changes in redox activity and the local coordination environment—potentially related to degree of order for the central ion’s coordination environment—they cannot directly prove that the “ $\text{Fe}^{\text{LS}}\text{-C}\equiv\text{N-M}^{\text{HS}}$ coordination structure is taken apart.” Therefore, we have revised the statement to:

“The shift of XANES profiles after the first cycle for M2-PBA is more pronounced than that of M4-PBA (Fig. 4b) and M5-PBA (Fig. S29b), indicating relatively poorer $\text{Fe}^{\text{LS}}\text{-C}\equiv\text{N-M}^{\text{HS}}$ structural reversibility and redox activity of M2-PBA (Fig. S29a).” (Page 15 Line 369-371)

In response to the reviewer’s inquiry about other M^{HS} K-edge data, we strongly agree and supplement the Mn K-edge measurements which present in all three PBAs. And it is essential to evaluate how cyanide electronic structure modulation affects the local changes for Mn coordination environment. The supplementary Mn K-edge XANES and EXAFS spectra are now provided in Figures S31 and S32, respectively. The following analysis has been added to the manuscript:

“Similarly, Mn K-edge XANES and corresponding FT-EXAFS curves were collected and analyzed (Figs. S32-S33). Benefiting from the uniform electronic distribution of $\text{Fe}^{\text{LS}}\text{-C}\equiv\text{N-M}^{\text{HS}}$, the profiles of the Mn–N and Mn–C coordination shells for M4-PBA exhibit smaller variations. This demonstrates that uniform electron distribution of $\text{Fe}^{\text{LS}}\text{-C}\equiv\text{N-M}^{\text{HS}}$ framework simultaneously enhances the stability of both M^{HS} and Fe^{LS} local coordination environments, which aligns with superior electrochemical cycling stability of M4-PBA.” (Page 16 Line 381-386)

Revised Fig.S32 Ex-situ XANES for Mn K-edge of a M2-PBA, b M4-PBA and c M5-PBA at different SOC with pre-edge d M2-PBA, e M4-PBA and f M5-PBA. (The final state of charge for the first discharge cycle was defined as SOC-1, while the charging and discharging final states of charge for the second cycle were defined as SOC-2 and SOC-3, respectively.) (Supporting Information)

Revised Fig.S33 Ex-situ FT-EXAFS for Mn K-edge of a M2-PBA, b M4-PBA and c M5-PBA with their imaginary parts d M2-PBA, e M4-PBA and f M5-PBA at different SOC. (The final state of charge for the first discharge cycle was defined as SOC-1, while the charging and discharging final states of charge for the second cycle were defined as SOC-2 and SOC-3, respectively.) (Supporting Information)

16. Line 346. EXAFS. The authors made some chemical/structural deduction on spectra only based on experimental FTs, this leads to misinterpretation of the structural variations

Response 16: We sincerely appreciate the reviewer's guidance on refining our discussion of chemical/structural evolution derived from FT-EXAFS data. As detailed in **Response 7**, while the coordination shells in FT-EXAFS do not directly map cyanide coordination changes, variations in intensity and radial distance of coordination shell enable inferences about reversible $\text{Fe}^{\text{LS}}\text{-C}\equiv\text{N-M}^{\text{HS}}$ structural evolution during desodiation/sodiation processes (*Adv. Energy Mater.* 2024, 14, 2400875; *J. Am. Chem. Soc.* 2000, 122, 28, 6653-6658; *Small* 2024, 20, 2404584). We fully acknowledge the reviewer's valid critique regarding the limitations of relying solely on FT-EXAFS for structural analysis. Consequently, we have implemented a comprehensive multi-technique correlation strategy: in-situ FT-IR spectroscopy (Fig. 4a and Fig. S25-S27) specifically monitors cyanide group evolution, while in-situ/ex-situ XRD (Fig. 4d-f and Fig. S34-S36) tracks phase transformation and unit cell volume change, collectively elucidating structural changes across three PBAs at varying states of charge. In accordance with the reviewer's earlier suggestion, we have supplemented the corresponding imaginary parts of FT-EXAFS curves as additional evidence to

validate the chemical/structural evolution (Fig. S31):

“As to the FT-EXAFS result of M2-PBA, the profiles of Fe–C and Fe–N coordination shells change obviously from SOC-1 to SOC-3 (Fig. S30a). In detail, both the profile intensity and position undergo significant changes in the FT-EXAFS curves and corresponding imaginary parts (Fig. S31), demonstrating the poor stability of the Fe^{LS}–C≡N–M^{HS} coordination structure.” (Page 15-16 Line 371-375)

Revised Fig.S31 Imaginary parts of ex-situ Fe K-edge FT-EXAFS signals for a M2-PBA, b M4-PBA and c M5-PBA at different SOC. (The final state of charge for the first discharge cycle was defined as SOC-1, while the charging and discharging final states of charge for the second cycle were defined as SOC-2 and SOC-3, respectively.) (Supporting Information)

17. Experimental. Details missing in the various adopted experimental techniques, such as - for example in XAFS data at synchrotron, starting E and final E as well as step of each K-edges, detector used and characteristics, use of mirrors?, monochromator and energy of the sources.

Response 17: We appreciate the reviewer’s comments and supplement the XAFS experimental details. The methodology has been supplemented as follows:

“Hard X-ray absorption spectroscopy (XAS) measurements were carried out in a transmission mode at BL17B beamline, Shanghai Synchrotron Radiation Facility (SSRF). For each spectrum, the scanning energy range was -200 eV to 700 eV with a step size of 0.5 eV near the K-edge (-30 eV to 100 eV) and a step size of 3 eV for the other energy ranges. X-ray photon energy was monochromatized by a Si (111) channel-cut crystal monochromator.” (Page 23 Line 548-553)

Minor points

We are very grateful to the reviewers for pointing out the minor but important issues regarding the details of the paper. We carefully revised these issues and reviewed the entire text. The following is a point-to-point reply.

18. Line 102, please specify which metals are involved in the notation M2-PBA, M4-PBA, and M5-PBA, upon mentioning them the first time”

Response 18: Thank you for the reviewer’s meticulous suggestion. We agree that the metal constituents should be specified upon first mention of M2-PBA, M4-PBA and M5-PBA. The

text was revised as:

“In consideration of the selection number of M^{HS} , the samples are denoted as M2-PBA (in which the pre-set M^{HS} are equimolar Mn and Fe), M4-PBA (in which the pre-set M^{HS} are equimolar Mn, Fe, Co and Ni), and M5-PBA (in which the pre-set M^{HS} are equimolar Mn, Fe, Co, Ni and Cu), respectively.” (Page 5 Line 109-112)

19. Line 105: “transmission electronic microscopy (TEM)” change to “electron”;

Response 19: Thank you for the reviewer’s scrupulous suggestion. The “transmission electronic microscopy (TEM)” has been revised to “electron”. And all suggested revisions have been incorporated, and the manuscript has undergone thorough proofreading to ensure consistency and accuracy throughout. The relative discussion has been revised:

“The comparison of high-resolution transmission electron microscopy (TEM) images accompanied by energy dispersive spectroscopy (EDS) mapping are displayed in Fig. S2-S4.” (Page 6 Line 113-115)

20. Line 125, fig. 1j: the figure is oriented at angle;

Response 20: Thank you for the reviewer’s meticulous comments. The requested DFT calculations have been performed, and the relevant Fig. 1j have been revised accordingly.

Revised Fig. 1j Cross-sectional charge density distribution diagrams of rhombohedral phase Cu-PBA. (Page 7 Line 146)

21. Line 130, table S2-S4: in monoclinic cells, it’s better to also indicate the numeric value of the angle which differs from 90° ;

Response 21: Thank you for the reviewer’s comments, which rightly emphasize that identifying bond angles deviating from 90° is essential for understanding the crystallographic data of monoclinic unit cells. Based on the re-refined XRD results, we have identified a series of such angles (β) and updated them in Tables S3-S5 for clarity. For more detailed XRD refinement data, please refer to **Response 2**. Here, as specifically requested by the reviewer, we present the β angle deviating from 90° in the monoclinic cell:

Revised Table S3 Rietveld refined result of M2-PBA: “ $\beta = 92.498^\circ$ ” (Supporting Information)

Revised Table S4 Rietveld refined result of M4-PBA: " $\beta = 92.545^\circ$ " (Supporting Information)

Revised Table S5 Rietveld refined result of M5-PBA: " $\beta = 92.501^\circ$ " (Supporting Information)

22. Line 164: figure caption "a" and "b" are mixed;

Response 22: Thank you for the reviewer's comments, and we apologize for our carelessness. The figure caption of "2a" and "2b" has been revised as suggested, and all figure captions throughout the manuscript have been thoroughly verified for consistency and accuracy. The following are the revised contents:

Revised "Fig. 2 | Characterization of cyanide coordination structures. a XANES. b EXAFS."
(Page 9 Line 203)

23. Line 266: the indication "(Fig. S19b)" needs to be written one sentence up, when there is explanation about charging (not about discharging);

Response 23: Thank you for the reviewer's comments. The suggested supplementary explanations will make the discussion more accurate. The relative description has been revised:

"Compared with M4-PBA and M5-PBA, the Na^+ diffusion coefficient of M2-PBA decreases obviously, suggesting the adverse influence of Jahn-Teller effect and serious phase transition during the charge process." (Page 12 Line 282-284)

---Reviewer #2 (Remarks to the Author):

We appreciate Reviewer #2 for the participation in the review process as part of Nature Communications' initiative to support early career researchers (ECRs). We greatly appreciate the time and effort you and your co-reviewer invested in evaluating our manuscript. Initiatives like this that foster mentorship and recognition for ECRs are invaluable to the scientific community, and we are pleased to have contributed to this endeavor through the review of our work.

---Reviewer #3 (Remarks to the Author):

The work reported that modulating the local electronic structure surrounding high-spin metals in PBAs can be employed to optimize the cyanide coordination environment, enabling a uniform electron distribution within the crystal structure. Thanks to the uniform electron distribution, the reactivity of the transition metals can be enhanced to get a high sodium storage capacity. In addition, the regulation of electronic displacement within the cyanide coordination environment significantly improves the crystal structural stability. The work is significant and can be considered in Nc Journal after a further improvement.

We sincerely appreciate the reviewer's recognition for the innovative concept of "Uniform electron distribution in the $\text{Fe}^{\text{LS}}\text{-C}\equiv\text{N-M}^{\text{HS}}$ framework enhances the electrochemical activity of transition metal ions and stabilizes the coordination structure" in this work. Regarding the reviewer's inquiries about the optimization process for local coordination electronic structure modulation and the potential application of this modification strategy to polyanion-type cathode materials, we provide the following detailed responses.

1. How to choose the transition metal type and the combination of the transition metals? And how many kinds are reasonable? Is there a principle of universality? In the work, the samples of $\text{Na}_{1.84}\text{Mn}_{0.503}\text{Fe}_{0.497}[\text{Fe}(\text{CN})_6]_{0.892}\cdot 1.633\text{H}_2\text{O}$, $\text{Na}_{1.927}\text{Mn}_{0.266}\text{Fe}_{0.247}\text{Co}_{0.248}\text{Ni}_{0.239}[\text{Fe}(\text{CN})_6]_{0.943}\cdot 0.862\text{H}_2\text{O}$ and $\text{Na}_{1.898}\text{Mn}_{0.211}\text{Fe}_{0.202}\text{Co}_{0.199}\text{Ni}_{0.198}\text{Cu}_{0.190}[\text{Fe}(\text{CN})_6]_{0.927}\cdot 0.675\text{H}_2\text{O}$ are selected. Why use the combination of MnFe, MnFeCoNi, and MnFeCoNiCu with different types and numbers? How to decide? We just observe an uniform and symmetric distribution of the cyanide electron for the M4 sample based on the calculation and experiments, while we don't know the optimized procedure of designing the 4 transition metals together.

Response 1: We appreciate the reviewer's comments regarding the principle behind the selection of transition metals and their potential contributions to the performance of PBAs. The selection and compositional tuning of N-coordinated transition metals (M^{HS}) represent a significant challenge in the structural design and controllable synthesis of Prussian blue analogues (PBAs). This study aims to address this challenge by proposing a $\text{Fe}^{\text{LS}}\text{-C}\equiv\text{N-M}^{\text{HS}}$ electronic structure design strategy. In PBA crystals, the choice of M^{HS} is constrained by lattice compatibility requirements. Considering battery energy density demands, fourth-period transition metals (e.g., Mn, Fe, Co, Ni, Cu) remain the primary focus in PBA research (*Adv. Mater.* 2021, 33, 2101342; *Adv. Funct. Mater.* 2022, 32, 2202372). However, the impact of varying M^{HS} compositions on the electronic structure of PBAs has not been comprehensively investigated.

Thus, as a proof-of-concept investigation, we performed density functional theory (DFT) calculations on single- M^{HS} -based PBAs to systematically evaluate their electronic structures.

Comparative analysis of differential charge density distributions revealed distinct M^{HS} -dependent modulations of the $Fe^{LS}-C\equiv N-M^{HS}$ coordination environment. Leveraging the additive effects of M^{HS} on this framework, we synthesized equimolar multi-metal PBAs to engineer targeted electronic configurations: M2-PBA (Mn/Fe) was designed to bias electron density toward M^{HS} , M4-PBA (Mn/Fe/Co/Ni) to achieve uniform electron distribution, and M5-PBA (Mn/Fe/Co/Ni/Cu) to shift electron density toward Fe^{LS} . Subsequent DFT simulations and structural characterizations confirmed that tailoring M^{HS} composition effectively modulates the $Fe^{LS}-C\equiv N-M^{HS}$ electronic structure.

It should be noted that formula deviations from pre-set stoichiometries inevitably occur due to ligand interactions and intrinsic defects during synthesis. This work primarily establishes the fundamental relationship between $Fe^{LS}-C\equiv N-M^{HS}$ electronic modulation and electrochemical performance, validated through comprehensive in-situ/ex-situ structural analyses. We acknowledge that the optimization process for achieving ideal uniform electron distribution requires further computational screening and experimental refinement, which will be systematically addressed in future studies. In response to the reviewer's queries regarding the design rationale of PBAs, we have provided supplementary elaboration in the manuscript, thereby enhancing the logical coherence of the paper:

“To conceptually validate the electronic modulation of the $Fe^{LS}-C\equiv N-M^{HS}$ framework, three PBAs samples with varying M^{HS} compositions were synthesized via an optimized co-precipitation method. To systematically probe the influence of different M^{HS} ions on the $Fe^{LS}-C\equiv N-M^{HS}$ electronic structure, the M^{HS} components in each PBA were introduced at equimolar ratios during synthesis.” (Page 5 Line 105-109)

2. Whether the uniform electron distribution for the crystal structure is suitable for the design of other kind of cathode, for example, polyanion types?

Response 2: Thank you for the reviewer's thought-provoking comment regarding the universality of the study on uniform electron distribution on other sodium-ion battery cathodes. To our knowledge, localized electronic structure modulation strategies have been implemented in other sodium-ion battery cathode materials, particularly polyanionic-type cathodes (*Adv. Mater.* 2023, 35, 2304428; *Adv. Funct. Mater.* 2024, 34, 2309701; *Adv. Funct. Mater.* 2024, 34, 2310248; *Adv. Mater.* 2025, 2505093, etc.). However, distinct from the design principles for polyanionic materials, PBAs uniquely benefit from uniform-distributed electron clouds across $Fe^{LS}-C\equiv N-M^{HS}$ due to their characteristic coordination framework.

In PBAs, cyanide ligands act as “bridges” forming $M^{HS}-N$ and $Fe^{LS}-C$ coordination networks, creating a repeating structural unit. Within the $Fe^{LS}-C\equiv N-M^{HS}$ chains, the choice of M^{HS} directly modulates electron density distribution at $M^{HS}-N$, consequently influencing the electron distribution

of Fe^{LS}-C bonds. While electron accumulation around active sites (M^{HS} and Fe^{LS}) hinders electron transfer and lowers redox activity, and excessive electron localization may reduce structural integrity of PBAs frameworks (ACS Nano 2024, 18, 1995-2005). Therefore, uniform cyanide coordination electronic distribution in PBAs simultaneously optimizes transition metal reactivity and structural stability. Contrastingly, the electronic structural design of polyanion-type cathode follows divergent principles owing to distinct structural configurations. Research in this domain primarily focuses on: (1) Enhancing structural stability through element doping to increase electron localization function (ELF) values; (2) Reducing bandgaps to improve electronic conductivity and minimize polarization for higher reversible capacity.

Although uniform electron distribution strategies may not directly apply to polyanion-type cathodes, given their significance as promising SIB cathodes, we have incorporated two seminal references on polyanionic electronic structure engineering in the Introduction (Adv. Mater. 2025, 2505093; Adv. Funct. Mater. 2020, 30, 1908680). These additions broaden the scientific context and enhance the manuscript's impact within the energy materials community. The related supplement is as followed:

“Sodium-ion batteries (SIBs) have emerged as promising candidates for large-scale energy storage applications. Among the key components determining their performance, cathode materials play a pivotal role in governing the energy density, cost-effectiveness, and cycling stability of SIBs. Currently, three main categories of cathode materials are being extensively investigated: polyanion compounds^{1,2}, layered oxides^{3,4}, and Prussian blue analogues (PBAs). Notably, PBAs stand out as particularly attractive cathode materials due to their unique open three-dimensional framework structure, which offers distinct advantages for sodium ion storage⁵.” (Page 3 Line 29-35)

- 1 Wang, L. et al. Fast screening suitable doping transition metals to Na₃V₂(PO₄)₂F₃ for sodium-ion batteries with high energy density in wide-temperature range. *Adv. Mater.* 2505093 (2025).
- 2 Zhao, Y. et al. Three electron reversible redox reaction in sodium vanadium chromium phosphate as a high-energy-density cathode for sodium-ion batteries. *Adv. Funct. Mater.* **30**, 1908680 (2020).
- 3 Wu, Z.-H. et al. Regulating the electrochemical performance of A₂Ni₂TeO₆ (A = Na, K) as a cathode of alkali metal ion battery by 3d transition metal substitution from a theoretical perspective. *ACS Appl. Energy Mater.* **7**, 8715-8725 (2024).
- 4 Wu, Z.-H. et al. Density functional theory investigation on fast storage of sodium and potassium in Ni₂TeO₆ as a novel promising cathode material. *Mater. Today Energy* **37**, 101414 (2023).
- 5 Wang, L. et al. Rhombohedral prussian white as cathode for rechargeable sodium-ion batteries. *J. Am. Chem. Soc.* **137**, 2548-2554 (2015).

---Reviewer #4 (Remarks to the Author):

Prussian blue analogues (PBAs) with open 3D framework structure are highly promising for practical applications as cathode of sodium-ion batteries (SIBs) due to their high theoretical energy density, low cost, and ease of large-scale preparation. However, PBAs often suffer from capacity degradation, resulting in continuous energy loss, impeding commercialization for practical SIBs. Herein, Wang et al. addressed the challenges by modulating the local electronic structure surrounding high-spin metals to optimize the cyanide coordination environment, enabling a uniform electron distribution within the crystal structure. The authors found that uniform electronic structure enhances the reactivity of the transition metals, resulting in 95.7 % of the theoretical capacity. The reported results demonstrate the potential of PBA cathode for large-scale commercial SIBs. The manuscript is of wide interest, of good significance and of high impact. Considering its valuable insights into enhancing the practical application of SIBs, I recommend this manuscript for publication in Nature Communications after some proper revision are made.

We sincerely appreciate the reviewer's constructive feedback regarding the research context, experimental details, and theoretical computational methodologies. In response to the specific concerns raised, we have provided detailed clarifications and revisions as outlined below, aiming to enhance the rigor and completeness of the manuscript.

The following points are my concerns to the manuscript that require further clarification.

1. In the "PBAs Preparation" section, the authors wrote that 5 mmol equimolar amount of $\text{MnSO}_4 \cdot \text{H}_2\text{O}$, $\text{FeSO}_4 \cdot 7\text{H}_2\text{O}$, $\text{CoSO}_4 \cdot 7\text{H}_2\text{O}$, and $\text{NiSO}_4 \cdot 6\text{H}_2\text{O}$ in 0.8 mol·L⁻¹ sodium citrate solution were dissolved to obtain 100 ml solution A. From this description, one cannot figure out how many grams of mols of sodium citrate, or salts in solution A.

Response 1: We appreciate for the reviewer's suggestion regarding for the preparation details of PBAs. We have revised the related description in the "PBAs Preparation" section in the manuscript to supplement details and eliminate ambiguities.

"5 mmol equimolar amount of $\text{MnSO}_4 \cdot \text{H}_2\text{O}$, $\text{FeSO}_4 \cdot 7\text{H}_2\text{O}$, $\text{CoSO}_4 \cdot 7\text{H}_2\text{O}$, and $\text{NiSO}_4 \cdot 6\text{H}_2\text{O}$ were dissolved in 100 ml deoxygenated deionized water with 80 mmol sodium citrate to obtain solution A." (Page 21 Line 489-491)

2. In the "Theoretical calculations" section, the authors used the spin-polarized generalized gradient approximation (GGA) with the Perdew-Burke-Ernzerhof (PBE) functional with Hubbard U corrections. It is known that Hubbard U corrections can improve the results for electronic properties such as band gap. However, the van der Waals dispersive interactions is underestimated. Why not to use dispersion-correction method in the energy calculations? How large the supercell was used in the calculations?

Response 2: We sincerely appreciate the reviewer's insightful comment regarding the inclusion of van der Waals (vdW) interactions in our calculations. In this study, we employed the PBE+U method primarily to address the strong electron correlation effects in the transition metal 3d orbitals, which is critical for accurately describing the electronic structure (e.g., bandgap and oxidation states) of our system. We incorporated the van der Waals dispersion interactions into our calculations by employing the DFT-D3 dispersion correction method. However, this detail was inadvertently omitted in the manuscript.

In addition, we apologize for not explicitly stating the supercell size in the original manuscript. For the single M^{HS} -based PBAs, due to their high symmetry, we directly employed the unit cell for computational modeling without supercell construction (Fig. 1a-j). Depending on the phase structure differences, the number of atoms per unit cell varies. For cubic phase (desodiated state), there are 58 atoms in the unit cell, while 48 atoms in the unit cell of rhombohedral phase (sodiated state). As to the density functional theory (DFT) calculations of the actual Prussian blue analog (PBA) structures, we constructed a $2 \times 2 \times 1$ supercell containing 192 atoms to closely match the elemental ratios observed in the experimentally synthesized PBA crystals. This supercell configuration ensures that the computational model accurately reflects the compositional and structural complexity of the real materials, including the precise stoichiometry of M^{HS} , Fe^{LS} , and cyanide ligands.

According to the suggestion of reviewer, we have supplemented the theoretical calculation methods details:

“The van der Waals dispersion interactions were incorporated into the calculations by employing the DFT-D3 dispersion correction method.” (Page 23 Line 561-563)

“For the single M^{HS} -based PBAs, due to their high symmetry, the unit cells for computational modeling were employed directly without supercell construction (Fig. 1a-j). Depending on the phase structure differences, the number of atoms per unit cell varies. For cubic phase (desodiated state), there are 58 atoms in the unit cell, while 48 atoms in the unit cell of rhombohedral phase (sodiated state). As to the density functional theory (DFT) calculations of the actual Prussian blue analog (PBA) structures, we constructed a $2 \times 2 \times 1$ supercell containing 192 atoms to closely match the elemental ratios observed in the experimentally synthesized PBA crystals.” (Page 23-24 Line 565-572)

3. The manuscript identifies that a uniform cyanide electron distribution within the crystal structure, maintain the stability of Fe^{LS} -CN- M^{HS} coordination structure during the desodiation/sodiation process. Could the authors provide more quantitative data linking specific structural changes (e.g., lattice distortions, volume changes) during cycles with a descriptor of cyanide electron distribution? Or the quantitative relationship between the electron distributions and the observed capacity loss?

Response 3: We sincerely appreciate the reviewer’s suggestion to incorporate quantitative comparisons of how uniform cyanide electron distribution influences PBAs structural evolution. We fully agree that applying Bragg’s law to X-ray diffraction data enables precise calculation of unit cell parameter changes, thereby providing deeper insights into how uniform electron distribution enhances structural stability. Accordingly, we have reprocessed the in-situ and ex-situ XRD datasets to quantitatively evaluate the impact of uniform cyanide electron distribution on crystal stability through cyclic variations in unit cell volume during charge/discharge (Revised Fig. 4f, Fig. S35f and Fig. S36f).

Additionally, in-situ FT-IR spectroscopy (Figure S24) quantitatively compares the CN absorption peak area ratios associated with M^{HS} and Fe^{LS} sites. The results demonstrate that uneven electron distribution asymmetrically suppresses the activity of either M^{HS} or Fe^{LS} , directly correlating with capacity decay in PBAs.

According to the reviewer’s suggestion, the following is the supplementary content and corresponding figures:

“Bragg’s law allows quantitative determination of unit cell volume evolution in PBAs during Na^+ insertion and extraction process, providing critical insights into the effect of uniform cyanide electron distribution on structural stability. As shown in Fig. 4f, M4-PBA exhibits slight volume evolution caused by irreversible monoclinic-cubic phase transition during initial charging process, and highly reversible unit cell volume variations are observed starting from the first discharge process. In contrast, M2-PBA shows minimal volume variation but suffers from poor reversibility (Fig. S35f). Although M5-PBA exhibits reversible volume changes (Fig. S36f), the substantial amplitude of these fluctuations compromises its structural stability during the repeated Na^+ insertion and extraction process.” (Page 17 Line 406-414)

Revised Fig. 4f normalized unit cell volume change of M4-PBA (Page 17 Line 415 and Page 18

Line 420)

Revised Fig. S35f normalized unit cell volume change of M2-PBA (Supporting Information)

Revised Fig. S36f normalized unit cell volume change of M5-PBA (Supporting Information)

4. The manuscript shows Co and Ni doping improve the cycling stability performance of cathode materials (M4-PBA). Could the authors clarify whether the introduction of Co and Ni affects the electronic structure of Fe and Mn, and how does this impact the overall redox activity?

Response 4: Thank you for the reviewer's comments regarding for the influence of Co and Ni on the electronic structure in PBAs. The doping of Co and Ni modulates the electronic structure of the $\text{Fe}^{\text{LS}}\text{-C}\equiv\text{N-M}^{\text{HS}}$ coordination framework, thereby influencing the local coordination electronic environments of Fe^{2+} and Mn^{2+} .

By comparing the electronic structures of M4-PBA (containing Co/Ni doping) and M2-PBA (Fe/Mn-based), we elucidate the mechanistic impact of $\text{Co}^{2+}/\text{Ni}^{2+}$ incorporation on $\text{Fe}^{2+}/\text{Mn}^{2+}$ -based PBAs. For instance, cross-sectional electronic structure comparisons in Fig. 1k and 1l reveal that the synergistic effects of Co^{2+} and Ni^{2+} promote homogenized electron distribution within the $\text{Fe}^{\text{LS}}\text{-}$

$C\equiv N-M^{HS}$ framework, effectively mitigating localized charge polarization. Additionally, the differential charge density maps in Fig. 5d and 5e illustrate charge redistribution between the Co/Ni dopants and the Fe^{2+}/Mn^{2+} host lattice, demonstrating enhanced electronic uniformity in M4-PBA due to strong dopant-host electronic interactions. Furthermore, the Mössbauer spectroscopy analysis in Fig. 2d-e and S13 provides atomic-scale insights into how Co^{2+}/Ni^{2+} doping modulates the electronic structure of PBAs. The results show that Co^{2+}/Ni^{2+} substitution alters the hyperfine interactions of Fe nuclei, directly reflecting changes in the coordination electronic structure symmetry of Fe^{LS} and M^{HS} . This electronic restructuring not only stabilizes the lattice but also optimizes the electrochemical activity of the material. Specifically, the synergistic role of Co^{2+} and Ni^{2+} balances electron density distribution across the coordination framework, alleviating structural distortions caused by localized strain during cycling, thereby significantly improving the capacity retention rates and structural reversibility of multi-metal PBAs. The series of analyses systematically clarifies the mechanism by which Co^{2+}/Ni^{2+} doping enhances PBA performance from the perspectives of electronic structure and atomic coordination environments.

According to the reviewer's kind suggestion, we highly agree that it is crucial to further clarify the influence of Co and Ni on the cyanide coordination electronic structure. The discussion is supplemented as followed:

“In M2-PBA, where the M^{HS} sites are exclusively occupied by Mn and Fe, the electron density of the cyanide ligands shifts toward M^{HS} ions. In M4-PBA, the partial substitution from Mn and Fe to Co and Ni at the M^{HS} sites results in symmetrical distribution of cyanide electron clouds between M^{HS} and Fe^{LS} . The incorporation of Cu at M^{HS} sites (M5-PBA) induces an electron density redistribution of cyanide coordination structure toward the Fe^{LS} centers. Additionally, the cyanide electron clouds with a pear shape in M2-PBA and M5-PBA can be observed in the insets of Fig. 1k and Fig. 1m, while M4-PBA displays a uniform cyanide electronic distribution with an ellipsoidal shape, aligning with the intended distribution of cyanide electrons according to the results of first-principles calculations in Fig. 1a-j.” (Page 6-7 Line 137-145)

As to the improvement of redox activity by Ni^{2+} and Co^{2+} doping, it is related to the distribution of cyanide electronic coordination structure. The corresponding discussion has been revised as:

“Comparing to M4-PBA, the previous analysis reveals that the cyanide ligand electrons in M2-PBA and M5-PBA are biased to M^{HS} and Fe^{LS} , respectively, thereby inhibiting the redox activity of the corresponding transition metals^{42,43}.” (Page 11 Line 241-243)

42 Zhang, H. et al. Prussian blue analogues with optimized crystal plane orientation and low crystal defects toward 450 Wh kg^{-1} alkali-ion batteries. *Angew. Chem. Int. Ed. Engl.* **62**, e202303953 (2023).

43 He, Y. et al. Entropy-mediated stable structural evolution of Prussian white cathodes for long-

life Na-ion batteries. *Angew. Chem. Int. Ed. Engl.* **63**, e202315371 (2024).

5. The authors should introduce in Section 1 the general cathode materials of SIBs. It is known that several cathode materials have been extensively explored for SIBs in recent years, including layered oxides, polyanionic compounds, and Prussian blue analogues (PBAs). Among them, as I known, P2 phases of $\text{Na}_2\text{Ni}_2\text{TeO}_6$ [see: *ACS Appl. Energy Mater.* **7**, 8715–8725 (2024); *Mater. Today Energy* **37**, 101414 (2023)] has also been extensively explored for SIBs recently with good electrochemical performances. A simple outline of cathode materials for SIBs would be essential to broaden the interest of readers and enhance the significance of the manuscript.

Response 5: Thank you for the reviewer's suggestion to provide an overview of other sodium-ion battery cathode materials. We have included a brief overview of other cathode materials, including layered oxides, polyanion compounds and PBAs, to broaden the readers' interest and enhance the promotional significance of the manuscript. The relevant discussion has been added to the first paragraph of the Introduction section:

“Sodium-ion batteries (SIBs) have emerged as promising candidates for large-scale energy storage applications. Among the key components determining their performance, cathode materials play a pivotal role in governing the energy density, cost-effectiveness, and cycling stability of SIBs. Currently, three main categories of cathode materials are being extensively investigated: polyanion compounds^{1,2}, layered oxides^{3,4}, and Prussian blue analogues (PBAs). Notably, PBAs stand out as particularly attractive cathode materials due to their unique open three-dimensional framework structure, which offers distinct advantages for sodium ion storage⁵.” (Page 3 Line 29-35)

- 1 Wang, L. et al. Fast screening suitable doping transition metals to $\text{Na}_3\text{V}_2(\text{PO}_4)_2\text{F}_3$ for sodium-ion batteries with high energy density in wide-temperature range. *Adv. Mater.* 2505093 (2025).
- 2 Zhao, Y. et al. Three electron reversible redox reaction in sodium vanadium chromium phosphate as a high-energy-density cathode for sodium-ion batteries. *Adv. Funct. Mater.* **30**, 1908680 (2020).
- 3 Wu, Z.-H. et al. Regulating the electrochemical performance of $\text{A}_2\text{Ni}_2\text{TeO}_6$ (A = Na, K) as a cathode of alkali metal ion battery by 3d transition metal substitution from a theoretical perspective. *ACS Appl. Energy Mater.* **7**, 8715-8725 (2024).
- 4 Wu, Z.-H. et al. Density functional theory investigation on fast storage of sodium and potassium in Ni_2TeO_6 as a novel promising cathode material. *Mater. Today Energy* **37**, 101414 (2023).
- 5 Wang, L. et al. Rhombohedral prussian white as cathode for rechargeable sodium-ion batteries. *J. Am. Chem. Soc.* **137**, 2548-2554 (2015).

6. To obtain a full comment on the materials studied, a clear comparison in terms of electrochemical

performance is needed for M2-PBA, M4-PBA, and M5-PBA. Could the authors provide a radar chart or a similarity to visually represent the trade-offs between energy density, rate performance, and cycle stability among M2-PBA, M4-PBA, and M5-PBA?

Response 6: Thank you for the reviewer’s comments. The radar chart is a clear way to compare the electrochemical performance of the three materials. We have added the radar chart in Fig. S24 to more intuitively present the electrochemical performance of M2-PBA, M4-PBA, and M5-PBA, including the first discharge specific capacity at 0.1C, the first discharge specific energy at 0.1C, the average discharge voltage at 0.1C, the capacity retention after 1000 cycles at 5C, and the discharge specific capacity at 20C. This visualization highlights the multi-dimensional advantages of the optimized M4-PBA system, particularly its superior rate capability and cycling stability. According to the reviewer’s suggestion, we have supplemented the radar chart and the relative discussion as followed:

“The electrochemical performance comparison of PBAs in this work is presented as a radar chart in Fig. S24. Additionally, the comparative performance of M4-PBA with previously reported results is summarized in Table S7.” (Page 13 Line 307-309)

Revised Fig. S24 Radar chart of electrochemical performance of PBAs. (Supporting Information)

7. Minor issues: The temperature at which the data shown in Fig. 3, S14-S16, S21 and S22 were measured should be given in the figure caption. For clarity and easy understanding, I suggest the authors use the left and right arrows to indicate, respectively, the curves for the capacity and Coulombic efficiency in Fig. 3b and 3g. The atoms for which the balls in different colors should be explained in Fig. 4e, 5g for clarity. The unit of density of states in Fig. 5a-c should be given in the title of the vertical axis. <v> In the “PBAs Preparation” section, the authors wrote: “5 g PVP (K30),

0.4 mol NaCl, and 0.2 g ascorbic acid were dissolved in deoxygenated deionized water". Why not to use the grams of NaCl here?

Response 7: Thank you for the reviewer's minor yet very important suggestions. These recommendations are crucial for enhancing the rigor of the paper and reducing potential misinterpretations of the images. We have made corrections to the images and the corresponding experiment details accordingly:

Revised Fig. 3 | Electrochemical characterization of PBAs with modified cyanide electronic structures at 30°C. a the first five times galvanostatic charge and discharge curves of M4-PBA, b comparison of cycling performance. DRT results of c M2-PBA, d M4-PBA and e M5-PBA. Full cell electrochemical performance comparison of f galvanostatic charge and discharge curves and g cycling performance of 1 C. (Page 13-14 Line 313-317)

Revised Fig. S15 The first five times galvanostatic charge and discharge curves of a M2-PBA and b M5-PBA at 30°C. (Supporting Information)

Revised Fig.S16 Comparison of dQ/dV curves at 30°C. a M2-PBA, b M4-PBA, and c M5-PBA.

(Supporting Information)

Revised Fig.S17 Comparison of cycling performance at 1 C for M2-PBA, M4-PBA and M5-PBA at 30°C. (Supporting Information)

Revised Fig.S20 a GITT testing curves, Na⁺ diffusion coefficients altered with electrode voltage for b charging and c discharging procedure in M2-PBA, M4-PBA and M5-PBA at 30°C. (Supporting Information)

Revised Fig.S22 Comparison of rate performance at 30°C. (Supporting Information)

Revised Fig.S23 Comparison of full cell electrochemical performance at 30°C. a specific capacity of PBA cathodes and hard carbon anode, b specific energy. (Supporting Information)

Revised Fig. 4g schematic illustration of the phase transition mechanism of M4-PBA. (Page 17-18 Line 415-421)

Revised Fig. 5g Schematic diagram of the enhanced electrochemical performance by cyanide electronic structure modification (Page 19 Line 455-458)

Revised Fig. 5 PDOS profiles of a M2-PBA, b M4-PBA and c M5-PBA. (Page 19 Line 455)

“The electrochemical performance data of galvanostatic charging and discharging curves were all measured by the Neware battery test system in the voltage range of 2.0-4.1 V at 30°C.” (Page 22 Line 513-515)

“The cyclic voltammetry (CV) curves and in-situ transient electrochemical impedance spectroscopy (in-situ EIS) were measured by Shanghai Chenhua CHI 760e electrochemical workstation at 30°C.” (Page 22 Line 523-524)

“5 g PVP (K30), 23.4 g NaCl, and 0.2 g ascorbic acid were dissolved in deoxygenated deionized water to form 200 ml transparent solution C.” (Page 21 Line 492-494)

Response to reviewers' comments

We sincerely thank the reviewers for their constructive comments and valuable suggestions, which have greatly contributed to improving the quality of the manuscript. In response to these comments and suggestions, we have conducted additional experiments and made comprehensive revisions to the original manuscript. Below are our responses (in blue font) to the comments. To be clear, all changes in the revised manuscript and supporting information are highlighted by red font.

REVIEWER COMMENTS

Reviewer #1 (Remarks to the Author):

We appreciate the modifications that has been made in this manuscript, all the additional and/or repeated experiments that has been conducted. We think that the overall quality of the manuscript has significantly increased, and a lot of doubtful points have been clarified. However, here are a few aspects regarding the new modification, that triggered our curiosity:

We highly appreciate the reviewer's careful review of the revised manuscript and their positive recognition. Addressing the remaining deficiencies in the manuscript and concerns raised by the reviewers, we have implemented corresponding modifications in the original text. Our point-by-point responses to the reviewer's comments are provided below.

Response 1 : stoichiometry calculations

- a. It was mentioned that EA was used, however the details about the instrumentation, or the experimental conditions are missing from the experimental part of the manuscript;
- b. TGA and EA results about H₂O quantification does not match. However, they do not offer any explanations for this difference? Ultimately, in the obtained stoichiometric formula, the water content calculated from the TGA experiment was considered, and not from EA. Again, the explanation about this choice is not given?
- c. Formula stoichiometry needs an additional check. With the backwards calculation from the offered stoichiometry, the errors for each element are substantial, sometimes over 10%

Authors' response 1:

We sincerely appreciate the reviewer's series of inquiries regarding the stoichiometric calculations, which are critical for precise determination of the PBA chemical formulas. Our point-by-point responses to the reviewer's comments are detailed below:

a. We are grateful for the reviewer's valuable suggestion to supplement experimental instrumentation details. Although we have added elemental analyzer (EA) testing, we acknowledge having omitted to provide specific instrument parameters in the original manuscript. Accordingly, we have incorporated the EA instrument parameters into the "Materials Characterization" section as follows:

"Elemental analyzer (EA) was performed using an Elementar Vario EL III elemental analyzer (Germany)." (Page 23 Line 541-542)

b. We sincerely appreciate the reviewer's insightful critique on this critical issue. As previously addressed, precise determination of PBA stoichiometry presents inherent challenges, which prompted our initial proposal of four stoichiometric equations under ideal conditions. However, the multiplicity of metallic elements in PBAs may significantly interfere with EA measurements. Thus,

our supplementary EA testing specifically aimed to verify the presence of C, N, H, and O as cyanide groups and H₂O, evidenced by the near-equimolar ratio of C to N elements and the approximately 2:1 molar ratio of H to O. Exclusive presence of [Fe(CN)₆] vacancies in all three PBAs was confirmed by electron paramagnetic resonance (EPR) results (Fig. S9), so that the total molar quantity of M^{HS} species was set as 1 during the determining process of the PBAs chemical formula. In fact, this method of determining PBAs chemical formulas has been widely adopted (*Nat. Commun.* 2025, 16, 4707, *Angew. Chem. Int. Ed.* 2023, e202315371). Based on the above analysis, PBA stoichiometry was determined through integrated analysis of EA, EPR, ICP-OES, and TGA data.

Regarding the quantification of crystal water, we advocate for TGA over EA results due to structural consideration of PBAs. EA involves instantaneous combustion in pure oxygen, wherein coordinated water molecules bound to M^{HS} sites may form transition metal oxides during rapid heating (*Adv. Funct. Mater.* 2022, 32, 2111727)—fundamental limitations affecting EA accuracy. Moreover, sample handling during EA (re-weighing/portioning) inevitably introduces moisture absorption errors. However, the TGA test employed a nitrogen atmosphere with 5 °C min⁻¹ heating, enabling sequential release of adsorbed, interstitial, and coordinated water (collectively termed crystal water) while preserving the PBA framework. This controlled dehydration minimizes the formation of metal oxidation, allowing accurate water quantification via mass loss. Consequently, based on comparative methodological analysis, TGA-derived crystal water content serves as the definitive basis for stoichiometric calculations (*Nat. Commun.* 2020, 11, 980).

In response to the reviewer's valid suggestion to eliminate ambiguity and enhance analytical rigor, we have explicitly detailed the methodology for determining crystalline water content through the following manuscript revisions:

“... and the content of crystal water in PBAs was determined by TGA³⁰.” (Page 6 Line 128-129)

30 Yang, Y. et al. Influence of structural imperfection on electrochemical behavior of Prussian blue cathode materials for sodium ion batteries. *J. Electrochem. Soc.* **163**, A2117-A2123 (2016).

c. We thank the reviewer for their suggestion and have meticulously re-verified the chemical formulas of the three PBA samples as requested. The formulas were confirmed to be accurate through comprehensive validation, with back-calculation verification tables presented below. To reiterate, given the substantial interference of high metallic content on EA results (as detailed in the preceding section), we employed TGA measurements as the definitive standard for crystalline water quantification.

① Back-calculation based on ICP-OES results

For M2-PBA, with the chemical formula $\text{Na}_{1.84}\text{Mn}_{0.50}\text{Fe}_{0.50}[\text{Fe}(\text{CN})_6]_{0.89}\square_{0.11}\cdot 2.74\text{H}_2\text{O}$,

Element	Relative atomic mass (g/mol)	Calculated atomic mass (g)	Calculated mass fraction (%)	ICP-OES results (%)
Na	22.99	42.30	28.70	28.70
Mn	54.94	27.47	18.64	18.64
Fe	55.85	77.63	52.67	52.67

For M4-PBA, with the chemical formula $\text{Na}_{1.89}\text{Mn}_{0.27}\text{Fe}_{0.27}\text{Co}_{0.25}\text{Ni}_{0.21}[\text{Fe}(\text{CN})_6]_{0.91}\square_{0.09}\cdot 1.42\text{H}_2\text{O}$,

Element	Relative atomic mass (g/mol)	Calculated atomic mass (g)	Calculated mass fraction (%)	ICP-OES results (%)
Na	22.99	43.45	28.73	28.73
Mn	54.94	14.83	9.81	9.81
Fe	55.85	65.90	43.58	43.57
Co	58.93	14.73	9.74	9.74
Ni	58.69	12.32	8.15	8.15

For M5-PBA, with the chemical formula $\text{Na}_{1.90}\text{Mn}_{0.23}\text{Fe}_{0.22}\text{Co}_{0.20}\text{Ni}_{0.18}\text{Cu}_{0.17}[\text{Fe}(\text{CN})_6]_{0.93}\square_{0.07}\cdot 1.14\text{H}_2\text{O}$,

Element	Relative atomic mass (g/mol)	Calculated atomic mass (g)	Calculated mass fraction (%)	ICP-OES test results (%)
Na	22.99	43.68	28.42	28.42
Mn	54.94	12.64	8.22	8.22
Fe	55.85	64.23	41.79	41.79
Co	58.93	11.79	7.67	7.67
Ni	58.69	10.56	6.87	6.87
Cu	63.55	10.80	7.03	7.03

These values correspond to metal ions content and demonstrate excellent agreement with the back-calculated results (deviations $\leq 0.05\%$).

② Back-calculation verification based on thermogravimetric analysis (TGA)

For M2-PBA with the chemical formula $\text{Na}_{1.84}\text{Mn}_{0.50}\text{Fe}_{0.50}[\text{Fe}(\text{CN})_6]_{0.89}\square_{0.11}\cdot 2.74\text{H}_2\text{O}$ (molecular weight: $335.71 \text{ g mol}^{-1}$), the back-calculated crystalline water mass fraction is 14.70%. For M4-PBA ($\text{Na}_{1.89}\text{Mn}_{0.27}\text{Fe}_{0.27}\text{Co}_{0.25}\text{Ni}_{0.21}[\text{Fe}(\text{CN})_6]_{0.91}\square_{0.09}\cdot 1.42\text{H}_2\text{O}$, MW: $318.90 \text{ g mol}^{-1}$), the derived water content is 8.02%. For M5-PBA with the chemical formula $\text{Na}_{1.90}\text{Mn}_{0.23}\text{Fe}_{0.22}\text{Co}_{0.20}\text{Ni}_{0.18}\text{Cu}_{0.17}[\text{Fe}(\text{CN})_6]_{0.93}\square_{0.07}\cdot 1.14\text{H}_2\text{O}$, MW: $319.43 \text{ g mol}^{-1}$, the calculated

mass fraction is 6.43%. As shown in TGA (Fig. S10a), the measured weight losses at 250°C are 14.7%, 8.0% and 6.4% for M2-PBA, M4-PBA and M5-PBA, respectively. These values correspond to crystal water content and demonstrate excellent agreement with the back-calculated results (deviations $\leq 0.05\%$).

③ The atomic ratios of non-metallic elements are determined using the mass percentages from elemental analyzer (EA) test results. The EA test data are presented in the following table:

Sample	C (%)	N (%)	H (%)	O (%)
M2-PBA	17.48	20.39	0.90	7.15
M4-PBA	18.25	21.29	0.80	6.32
M5-PBA	19.07	22.24	0.65	5.19

The relative atomic masses of C, N, H, and O are 12.01, 14.01, 1.008, and 16.00 respectively, enabling calculation of the molar ratios between corresponding atoms. For M2-PBA, the atomic ratio of C to N calculated from relative atomic masses is $1.46:1.46 = 1$. The H to O atomic ratio is $0.89:0.45 = 1.98$, approximating 2. For M4-PBA, the C:N atomic ratio is $1.52:1.52 = 1$, while the H:O ratio is $0.79:0.40 = 1.98$, approaching 2. For M5-PBA, the C:N atomic ratio is $1.59:1.59 = 1$, and the H:O ratio is $0.64:0.32 = 2$. However, as previously analyzed, EA results are significantly impacted by the high metallic content in PBAs, rendering them unsuitable for direct quantification of crystalline water content.

In summary, through back-calibration using ICP-OES, TGA and EA data, and combining with EPR results, the chemical formulas of the three PBAs remain consistent with our prior revision. In detail, $\text{Na}_{1.84}\text{Mn}_{0.50}\text{Fe}_{0.50}[\text{Fe}(\text{CN})_6]_{0.89}\square_{0.11} \cdot 2.74\text{H}_2\text{O}$ for M2-PBA, $\text{Na}_{1.89}\text{Mn}_{0.27}\text{Fe}_{0.27}\text{Co}_{0.25}\text{Ni}_{0.21}[\text{Fe}(\text{CN})_6]_{0.91}\square_{0.09} \cdot 1.42\text{H}_2\text{O}$ for M4-PBA and $\text{Na}_{1.90}\text{Mn}_{0.23}\text{Fe}_{0.22}\text{Co}_{0.20}\text{Ni}_{0.18}\text{Cu}_{0.17}[\text{Fe}(\text{CN})_6]_{0.93}\square_{0.07} \cdot 1.14\text{H}_2\text{O}$ for M5-PBA.

Response 2: XRD Rietveld refinement: the improvement in this portion is substantial, however it is not mentioned, after the defining initial occupancies of the elements from the stoichiometric data, if these values are fixed or varied during the refinement process (and why).

Authors' response 2:

We sincerely appreciate the reviewer's positive assessment of our re-refined XRD results. During the refinement process, we constrained the atomic site occupancies to prevent potential misinterpretations of the crystal structure, consistent with established practices in PBA research (*Angew. Chem. Int. Ed.* 2024, 63, e202315371, *Adv. Funct. Mater.* 2024, 2314167, *Angew. Chem. Int. Ed.* 2021, 60, 13050-13056). Acknowledging the reviewer's valid concern regarding precision enhancement, we have implemented supplementary elaboration in the manuscript to eliminate ambiguity, with revisions detailed as follows:

“The powder X-ray diffraction (PXRD) patterns of the three samples were analyzed through Rietveld refinement, with atomic site occupancies in agreement with the calculated stoichiometry (Fig. 1k-m).” (Page 6 Line 132-134)

Response 5: Pre-edge spectra (Fig. S11a) show a very small difference among three. The conclusion that: “deviation from octahedral symmetry, which could be attributed to its higher sodium content” is a strong statement for such a tiny modification.

Authors' response 3:

We fully agree with the reviewer's point regarding this issue. Although subtle differences exist in the pre-edge peaks among the three samples, directly correlating these variations to sodium content appears overly definitive. Therefore, as suggested by the reviewer, we have revised the relevant statements as below:

“All three samples exhibit distinct pre-edge peaks in their K-edge spectra of Fe, which can likely be attributed to relatively high sodium content (Fig. S11a)³⁵.” (Page 7-8 Line 160-162)

35 Li, M. et al. Influence of vacancies in manganese hexacyanoferrate cathode for organic Na-ion batteries: a structural perspective. *ChemSusChem* **16**, e202300201 (2023).

Response 6 (also 15 & 16): on our request to provide the spectra of EXAFS oscillations was answered by providing the plots of imaginary part of Fourier Transform signals. It's not clear why. We again would encourage the authors to provide the spectra of EXAFS oscillations.

Authors' response 4:

We appreciate the reviewer's suggestion. As requested, we have supplemented the spectra of EXAFS oscillations and replaced the corresponding plots of the imaginary part of Fourier transform signals. The relevant modifications in the manuscript and Supporting Information are provided below:

“The distinct profiles of Fourier-transformed EXAFS (FT-EXAFS) in Fig. 2b with their spectra of EXAFS oscillations (Fig. S11b) exhibit different coordination structures of central Fe atom for the three samples, correspondingly revealing the coordination environment of Fe–C and Fe–N shells in PBAs.” (Page 8 Line 162-165)

Fig. S11b spectra of EXAFS oscillations for Fe K-edge. (Supporting Information)

In detail, both the profile intensity and position undergo significant changes in the FT-EXAFS curves and corresponding spectra of EXAFS oscillations (Fig. S31), demonstrating the poor stability of the $\text{Fe}^{\text{LS}}\text{-C}\equiv\text{N-M}^{\text{HS}}$ coordination structure. (Page 16 Line 374-376)

Fig.S31 Spectra of EXAFS oscillations for a M2-PBA, b M4-PBA and c M5-PBA at different SOC. (The final state of charge for the first discharge cycle was defined as SOC-1, while the charging and discharging final states of charge for the second cycle were defined as SOC-2 and SOC-3, respectively.) (Supporting Information)

Similarly, Mn K-edge XANES and corresponding EXAFS curves were collected and analyzed (Fig. S32-S33). (Page 16 Line 382-383)

Fig.S33 Spectra of EXAFS oscillations for d M2-PBA, e M4-PBA and f M5-PBA at different SOC. (The final state of charge for the first discharge cycle was defined as SOC-1, while the charging and discharging final states of charge for the second cycle were defined as SOC-2 and SOC-3, respectively.) (Supporting Information)

Respond 8: the revised XPS spectrum of M5-PBA (fig. 2c) was a shoulder at Fe 2p_{3/2} peak (in the same area as for M2-PBA the Fe³⁺ contribution is offered), however, it is not fitted.

Authors' response 5:

We appreciate the reviewer's suggestion regarding the XPS deconvolution. Following this recommendation, we have reperfomed the deconvolution for M5-PBA in Fig. 2c. Details of this revision are outlined below:

Revised Fig. 2c XPS of M2-PBA, M4-PBA and M5-PBA (Page 9 Line 203)

It should be noted that while XPS deconvolution results appear to indicate the presence of Fe^{3+} , these primarily reflect the near-surface valence characteristics of the material (*Adv. Mater.* 2021, 33, 2101342). The XAFS and Mössbauer spectroscopy analyses confirm that iron predominantly exists in the +2 oxidation state throughout the bulk of all three PBAs. These findings are not contradictory, but rather complementary, as they probe distinct material regions with different depth sensitivities.

Response 9: With the reported stoichiometry, the values of theoretical capacity calculated by the authors has been rechecked. With M2-PBA, it was evident that the number of charge carrier was considered to be 2 (Na); and entire molecular mass of the material was taken (recalculation gave close enough match, with the negligible difference of $\approx 0.08 \text{ mAh g}^{-1}$). However, with M4-PBA and M5-PBA, results are not that clear. Their explanation states: “In PBAs cathodes, Ni and Cu coordinated to N are electrochemically inactive”, and because of that they have decided to modify the theoretical capacity. But, if they have decided: a) to modify 2Na, and reduced its value by the “Ni and Cu portion”, the values do not match, b) to subtract the portion of theoretical capacity, assigned to Ni and Cu, while still considering 2Na – the values again do not match. It would be helpful if they would provide the explanation and detailed calculation of the theoretical capacity obtained by the authors.

Authors’ response 6:

We appreciate the reviewer’s attention to the theoretical capacity calculation of PBAs materials, which is crucial for better comparing the electrochemical performance of the three PBAs. It is worth clarifying that, in our previous response, the adjusted theoretical capacity was not due to the electrochemically inactive Ni and Cu—which had already been accounted for in the initial calculation—but rather stemmed from minor stoichiometric adjustments in the PBAs. As requested, we have refined the theoretical capacity calculation method based on established PBA research (*ACS Energy Lett.* 2024, 9, 2748-2757, *Adv. Mater.* 2025, 2417876), with detailed analysis and computations provided below.

The general formula for calculating the theoretical capacity of electrode materials is:

$$C_T = \frac{n \times F \times 1000}{3600 \times M}$$

where C_T denotes the theoretical capacity, n represents the number of charge carriers transferred per formula unit, F is the Faraday constant (typically 96,485 C mol⁻¹), and M stands for the relative molecular mass of the electrode material.

In PBAs, two distinct transition metal sites are existed as the N-coordinated M^{HS} and the C-coordinated Fe^{LS}. To achieve high sodium content (approaching 2 Na⁺ per formula unit), both sites typically incorporate divalent transition metal ions. Through reversible +2/+3 redox reactions at both M^{HS} (e.g., Mn, Fe, Co) and Fe^{LS} centers, 2 mol of Na⁺ can be reversibly (de)intercalated, delivering capacities approaching 170 mAh g⁻¹ (specifically 171.32, 170.82, and 169.16 mAh g⁻¹ for Mn/Fe/Co-based PBAs, respectively). However, not all M^{HS} enable reversible +2/+3 redox. Ni and Cu ions preferentially maintain +2 oxidation states during cycling (*Adv. Mater.* 2024, 2405458), limiting Na⁺ storage to 1 mol via Fe^{LS} redox alone, with theoretical capacities of 84.64 and 83.36 mAh g⁻¹ for Ni-based and Cu-based PBAs, respectively.

Notably, these capacity values assume ideal crystal structures. As noted by the reviewer, practical calculations must account for [Fe(CN)₆] vacancies, actual Na⁺ content, and crystal water content on both charge carrier transfer number (n) and relative molecular mass (M) (*ACS Energy Lett.* 2024, 9, 2748-2757, *Adv. Mater.* 2025, 2417876). As recommended by the reviewer, we have recalculated the theoretical capacities for all three PBAs using the moles of electrochemically active transition metal ions as the charge carrier transfer number n and the calculated relative molecular mass of the three PBAs samples as M .

For M2-PBA with the chemical formula Na_{1.84}Mn_{0.50}Fe_{0.50}[Fe(CN)₆]_{0.89}□_{0.11}·2.74H₂O and a relative molecular mass of 335.71 g mol⁻¹, the theoretical capacity (C_T) is calculated as:

$$C_T = \frac{n \times F \times 1000}{3600 \times M} = \frac{(0.50 + 0.50 + 0.89) \times 96485 \times 1000}{3600 \times 335.71} \approx 150.89 \text{ mAh g}^{-1}$$

The theoretical specific capacity of M2-PBA is 150.89 mAh g⁻¹, with a practical specific capacity of 142.7 mAh g⁻¹, achieving 94.57% utilization of its theoretical capacity.

For M4-PBA with the chemical formula Na_{1.89}Mn_{0.27}Fe_{0.27}Co_{0.25}Ni_{0.21}[Fe(CN)₆]_{0.91}□_{0.09}·1.42H₂O and a relative molecular mass of 318.90 g mol⁻¹, the theoretical capacity (C_T) is calculated as:

$$C_T = \frac{n \times F \times 1000}{3600 \times M} = \frac{(0.27 + 0.27 + 0.25 + 0.91) \times 96485 \times 1000}{3600 \times 318.90} \approx 142.87 \text{ mAh g}^{-1}$$

The theoretical specific capacity of M4-PBA is 142.87 mAh g⁻¹, with a practical specific capacity of 142.4 mAh g⁻¹, achieving 99.67% utilization of its theoretical capacity.

As to M5-PBA (Na_{1.90}Mn_{0.23}Fe_{0.22}Co_{0.20}Ni_{0.18}Cu_{0.17}[Fe(CN)₆]_{0.93}□_{0.07}·1.14H₂O) with a relative molecular mass of 319.43 g mol⁻¹, the theoretical capacity (C_T) is calculated as:

$$C_T = \frac{n \times F \times 1000}{3600 \times M} = \frac{(0.23 + 0.22 + 0.20 + 0.93) \times 96485 \times 1000}{3600 \times 319.43} \approx 132.57 \text{ mAh g}^{-1}$$

The theoretical specific capacity of M5-PBA is $132.57 \text{ mAh g}^{-1}$, with a practical specific capacity of 109.5 mAh g^{-1} , achieving 82.60% utilization of its theoretical capacity.

We acknowledge the oversight in standardizing the theoretical capacity calculations for PBAs. Based on the recalculated capacities using the standardized methodology, M4-PBA still demonstrates the highest capacity utilization rate (99.67%), surpassing both M2-PBA (94.57%) and M5-PBA (82.60%). In accordance with the reviewer's suggestions and revised capacity values, we have modified the relevant descriptions as detailed below:

“The resulting uniform electronic structure enhances the reactivity of the transition metals, which helps to achieve 99.67 % of the theoretical capacity.” (Page 2 Line 21-23)

“Consequently, the optimized PBA achieves the simultaneous improvement of capacity and cycling lifetime, delivering 99.67% of the theoretical capacity at 0.1 C and retaining 91.7% of the reversible capacity after 1000 cycles at 5 C.” (Page 4 Line 81-83)

“The initial discharge capacities of M2-PBA, M4-PBA and M5-PBA are 142.7, 142.4 and 109.5 mAh g^{-1} , corresponding to 94.57%, 99.67% and 82.60% of their theoretical capacities, respectively.” (Page 10-11 Line 236-238)

Fig. S24 Radar chart of electrochemical performance comparison for PBAs in this work. (Supporting Information)

Reviewer #2 (Remarks to the Author):

We sincerely thank Reviewer #2 and editors for their constructive feedback, which has significantly strengthened the rigor and clarity of our manuscript.

Reviewer #3 (Remarks to the Author):

The work has been polished carefully and can be accepted in the current version.

We are deeply grateful for your final endorsement and scholarly insights. Your constructive critique has fundamentally strengthened the methodological rigor and readability of this work.

Reviewer #4 (Remarks to the Author):

The authors have well-addressed all my concerns in the revised version of the manuscript. The authors have revised the related description in the “PBAs Preparation” section in the revised manuscript to supplement details and eliminate ambiguities. The authors have explained in the revised manuscript that they employed the PBE+U method primarily to address the strong electron correlation effects in the transition metal 3d orbitals, which is critical for accurately describing the electronic structure (e.g., bandgap and oxidation states) of their system and incorporated the van der Waals dispersion interactions into our calculations by employing the DFT-D3 dispersion correction method. The authors have reprocessed the in-situ and ex-situ XRD datasets to quantitatively evaluate the impact of uniform cyanide electron distribution on crystal stability through cyclic variations in unit cell volume during charge/discharge in the revised version of the manuscript. Some discussions on the effects of Co and Ni on electronic structure of Fe and Mn, and how does this impact the overall redox activity have been supplemented in the revised manuscript. Now the revised manuscript has been improved adequately and enough detail has been provided in the methods for the work to be reproduced. Therefore, I am very willing to recommend the current form of the revised manuscript for publication in Nature Communications.

We deeply appreciate Reviewer #4’s suggestions for enhancing the details in the “PBAs Preparation” section and the DFT theoretical calculations, as well as for adding the quantitative description of how uniformly distributed cyanide electron distribution enhances structural stability. These suggestions proved crucial in refining the logical flow of the manuscript and further highlighting its novelty.

Response to reviewers' comments

We sincerely thank the reviewers for their constructive comments and valuable suggestions, which have greatly contributed to improving the quality of the manuscript. In response to these comments and suggestions, we have conducted additional experiments and made comprehensive revisions to the original manuscript. Below are our responses (in blue font) to the comments. To be clear, all changes in the revised manuscript and supporting information are highlighted by red font.

REVIEWER COMMENTS

Reviewer #1 (Remarks to the Author):

We appreciate the continuous effort of authors are making in order to improve the manuscript, and thank them for the comprehensive response letter, with detailed explanations.

We thank the reviewers for their valuable suggestions. In response to the concerns raised, we have summarized the key points into the following aspects and addressed each individually.

1. Regarding the calculation of the stoichiometry, the of extra technical information of EA, and the definition of the origin of water content calculation are positive additions. Unfortunately, one thing keeps being doubtful: the values about Fe HS/LS distribution, namely:

The EA results do not agree with the stoichiometric formula for C and N.

It is understandable, why the water content calculated from the EA is not reliable, but what about C and N? If the offered stoichiometry is correct, then the %wt of C&N should be different:

Sample: M2-PBA M4-PBA M5-PBA

Offered stoichiometry $\text{Na}_{1.84}\text{Mn}_{0.50}\text{Fe}_{0.50}[\text{Fe}(\text{CN})_6]_{0.89} \cdot 0.11 \cdot 2.74\text{H}_2\text{O}$

$\text{Na}_{1.89}\text{Mn}_{0.27}\text{Fe}_{0.27}\text{Co}_{0.25}\text{Ni}_{0.21}[\text{Fe}(\text{CN})_6]_{0.91} \cdot 0.09 \cdot 1.42\text{H}_2\text{O}$

$\text{Na}_{1.90}\text{Mn}_{0.23}\text{Fe}_{0.22}\text{Co}_{0.20}\text{Ni}_{0.18}\text{Cu}_{0.17}[\text{Fe}(\text{CN})_6]_{0.93} \cdot 0.07 \cdot 1.14\text{H}_2\text{O}$

Mw (g mol⁻¹) according to offered stoichiometry: 335.71 318.90 319.43

C (%wt.) calculated from offered stoichiometry $(12.01 \times 0.89 \times 6) / 335.71 \times 100 = 19.10$

$(12.01 \times 0.91 \times 6) / 318.90 \times 100 = 20.56$ $(12.01 \times 0.93 \times 6) / 319.43 \times 100 = 20.98$

N (%wt.) calculated from offered stoichiometry $(14.01 \times 0.89 \times 6) / 335.71 \times 100 = 22.29$

$(14.01 \times 0.91 \times 6) / 318.90 \times 100 = 23.99$ $(14.01 \times 0.93 \times 6) / 319.43 \times 100 = 24.47$

C (%wt.) according to EA 17.48 18.25 19.07

N (%wt.) according to EA 20.39 21.29 22.24

Error (measur. Vs stoich.) 8.5% 11.2% 9.1%

Instead, if the EA measurement about the C & N are reliable (and H₂O is taken from TGA), then the Fe HS/LS ratio (and consequently entire formula, as it is normalised on HS metal sum to be 1), will be changed.

For M2-PBA:

Element Relative atomic (molecular) mass (g mol⁻¹) ICP-OES results (%wt)

(metal mass sum = 100%) EA (C & N) / TGA (H2O) (%wt)

Recalculated (%wt) of metals

(entire mass = 100%) Normalised on Mw (mol) Normalised on HS metal sum to be 1

Na 22.99 28.7 $(28.7 \times (100 - (17.48 + 20.39 + 14.70))) / 100 = 13.61$ 0.59 $0.59 / (0.16 + 0.21) = 1.62$

Mn 54.94 18.64 $(18.64 \times (100 - (17.48 + 20.39 + 14.70))) / 100 = 8.84$ 0.16 $0.16 / (0.16 + 0.21) = 0.44$

Fe 55.85 52.67 $(52.67 \times (100 - (17.48 + 20.39 + 14.70))) / 100 = 24.98$ 0.45 $0.45 - 0.16 = 0.21$ (HS)
 $0.21 / (0.16 + 0.21) = 0.56$

$1.46 / 6 = 0.24$ (LS) $0.24 / (0.16 + 0.21) = 0.66$

C 12.01 - 17.48 $1.46 / (0.16 + 0.21) = 3.98$

N 14.01 - 20.39 $1.46 / (0.16 + 0.21) = 3.98$

H2O 18.016 - 14.70 $0.82 / (0.16 + 0.21) = 2.23$

The obtained stoichiometry from this calculation is:

[Na] $_1.62$ [Mn] $_0.44$ [Fe] $_0.56$ [Fe(CN) $_6$] $_0.66 \cdot 2.23\text{H}_2\text{O}$

With the molecular mass of: 273.50

Reverse calculation (for checking):

Water content:

Mw Mw of H2O H2O (%wt)

$273.50 (2 \times 1.008 + 16) \times 2.23 = 40.20$ $40.20 / 273.50 \times 100 = 14.7$

Carbon and Nitrogen:

Mw Mass of C Mass of N %wt C %wt N

273.50 $6 \times 0.66 \times 12.01 = 47.80$ $6 \times 0.66 \times 14.01 = 55.76$ $47.80 / 273.50 \times 100 = 17.48$

$55.76 / 273.50 \times 100 = 20.39$

Note: the mass of C and N are calculated before rounding up the stoichiometric values (0.66338), that's why they slightly mismatch if the calculations are done on already rounded up stoichiometry.

Metals:

Mw Mw of only metals %wt Na %wt Mn %wt Fe HS %wt Fe LS %wt Fe

273.50 $273.50 - 40.20 - 47.80 - 55.76 = 129.73$ $(22.99 \times 1.62) / 129.73 \times 100 = 28.70$

$(54.94 \times 0.44) / 129.73 \times 100 = 18.64$ $(55.85 \times 0.56) / 129.73 \times 100 = 24.11$

$(55.85 \times 0.56) / 129.73 \times 100 = 28.56$ 52.67

Note: the masses are calculated before rounding up the stoichiometric values, that's why they slightly mismatch if the calculations are done on already rounded up stoichiometry.

If the calculations are done in this way, for M4-PBA and M5-PBA the stoichiometric formulas will be:

[Na] _1.60 [Mn] _0.23 [Fe] _0.38 [Co] _0.21 [Ni] _0.18 [Fe(CN)_6] _0.62·1.09H_2
O

[Na] _1.65 [Mn] _0.20 [Fe] _0.32 [[Co] _0.17 [Ni] _0.16 [Cu] _0.15 [Fe(CN)_6]]
_0.67·0.91H_2 O

Response 1: We sincerely appreciate the reviewer's kind suggestions on the precise determination of PBA stoichiometry. **Accurate formulation of PBAs remains a persistent challenge, where one primary obstacle stems from reliable mass quantification of light elements (C, N, O and H).**

Elemental analysis (EA) fundamentally operates by detecting gaseous combustion products and deriving elemental composition through stoichiometric calculations of these gases. Within PBAs, transition metal constituents demonstrably interfere with H and O quantification via EA—a phenomenon comprehensively discussed in our previous response. **Similarly, EA detection of C and N remains susceptible to analogous interference mechanisms.** (1) Thermal stability. Strong Fe^{LS}-C coordination interaction (*Adv. Funct. Mater.* 2020, 30, 1908754; *Angew. Chem. Int. Ed.* 2022, 61, e202205867) may cause incomplete decomposition of [Fe(CN)₆]⁴⁻ units even up to 800°C. (2) Interfering byproducts. Decomposition of PBAs primarily releases HCN and (CN)₂ with strictly 1:1 stoichiometric relationship of C/N (*Adv. Energy Mater.* 2021, 11, 2101764; *Dalton Trans.* 2020, 49, 3570), which can cross-react with CO₂/N₂ detection systems.

We acknowledge the reviewer's instruction in utilizing EA test results, yet the derivation of PBA stoichiometric formulas based on imprecise content of C and N data appears to introduce considerable discrepancies. **Notably, the calculated M^{HS} stoichiometries exhibit significant deviations from the precursor content of transition metal salts (exclusively occupy M^{HS} sites in PBAs).** To prevent further misunderstandings, we outline the definitive procedure for PBA stoichiometric calculation method which is widely accepted in the field (*Nat. Commun.* 2025, 16, 4707; *Angew. Chem. Int. Ed.* 2023, e202315371; *Energy Stor. Mater.* 2023, 58, 1-8, as well as the citation noted by reviewer <https://doi.org/10.1002/adma.202419446>):

Conventionally, determining PBA stoichiometry begins with consideration of precursor inputs—particularly transition metal salts (M^{HS}) and sodium ferrocyanide (Fe^{LS}). Adjust transition metal contents based on ICP-OES data to normalize the total M^{HS} molar quantity to 1. Calculate Fe^{LS} content by subtracting Fe^{HS} from total Fe (ICP-OES), thereby quantifying $[Fe(CN)_6]$ units and associated vacancies. Incorporate crystal water content from TGA data to finalize the stoichiometric formula.

We maintain that EA test remains valuable for determining PBA stoichiometry, as it provides distinct C/N and H/O molar ratios—data unobtainable through either ICP-OES or TGA. Accurate stoichiometric determination persists as a fundamental challenge in PBA research. We will continue investigating precise characterization methodologies for definitive formula resolution, and we express renewed appreciation for the reviewer’s insightful attention to this matter. In accordance with the reviewer’s suggestion, we have supplemented the reference to <https://doi.org/10.1002/adma.202419446> as a methodological reference for PBA stoichiometric calculations, with specific modifications detailed as follows:

“Thermogravimetric analysis (TGA) in Fig. S10, EA (Table S1) and ICP-OES tests (Table S2) confirm that the chemical formulas of M2-PBA, M4-PBA, and M5-PBA are $Na_{1.84}Mn_{0.50}Fe_{0.50}[Fe(CN)_6]_{0.89 \square 0.11} \cdot 2.74H_2O$, $Na_{1.89}Mn_{0.27}Fe_{0.27}Co_{0.25}Ni_{0.21}[Fe(CN)_6]_{0.91 \square 0.09} \cdot 1.42H_2O$ and $Na_{1.90}Mn_{0.23}Fe_{0.22}Co_{0.20}Ni_{0.18}Cu_{0.17}[Fe(CN)_6]_{0.93 \square 0.07} \cdot 1.14H_2O$, respectively, where \square denotes $[Fe(CN)_6]$ vacancies in the crystal lattice and the content of crystal water in PBAs was determined by TGA^{30,31}.”

30 Yang, Y. et al. Influence of structural imperfection on electrochemical behavior of Prussian blue cathode materials for sodium ion batteries. *J. Electrochem. Soc.* **163**, A2117-A2123 (2016).

31 Zhou, M. et al. Ligand field-induced dual active sites enhance redox potential of nickel hexacyanoferrate for ammonium ion storage. *Adv. Mater.* 2419446 (2025).

2. If the vacancy calculations are coming from EPR, then the information is scarce and needs more details. The authors only say: “The g values of ~2.03 for three samples from the electron paramagnetic resonance (EPR) tests indicates the presence of $[Fe(CN)_6]^{4-}$ defect (Fig. S9). The

amplitude difference between M4-PBA and M5-PBA in the curves is small. The higher amplitude for M2-PBA indicates more crystal defects of $[\text{Fe}(\text{CN})_6]^{4-}$ in comparison with M4-PBA and M5-PBA". It seems, the described vacancy assessment from EPR is approximate (only qualitative, and not quantitative), so this should be acknowledged in the text.

If the stoichiometric calculations are done according to the different assumptions/thought process, it has to be explained, as the difference is substantial.

Response 2: We sincerely appreciate the reviewer's kind suggestions. As noted by the reviewer, EPR testing enables qualitative identification of radical species through g value calculations. For PBAs, EPR primarily serves to qualitatively confirm the presence of $[\text{Fe}(\text{CN})_6]^{4-}$ vacancies. **While some studies have achieved semi-quantitative comparison of vacancy concentrations by analyzing EPR curve amplitude and peak width** (Chem 2020, 6, 1804-1818; Nat. Commun. 2022, 13, 7790), our original text has been revised to prevent misinterpretation. The detailed reversion is as follows:

"The g values of ~ 2.03 for three samples from the electron paramagnetic resonance (EPR) tests indicates the presence of $[\text{Fe}(\text{CN})_6]^{4-}$ defect²⁸ (Fig. S9). It has been demonstrated that comparing the amplitude and peak width of EPR curves can reflect the concentration of $[\text{Fe}(\text{CN})_6]^{4-}$ defects to some extent²⁹. The amplitude difference between M4-PBA and M5-PBA in the curves is small, and the higher amplitude for M2-PBA indicates more crystal defects of $[\text{Fe}(\text{CN})_6]^{4-}$ in comparison with M4-PBA and M5-PBA" (Page 6 Line 120-125)

28 Li, X. et al. Interior-confined vacancy in potassium manganese hexacyanoferrate for ultra-stable potassium-ion batteries. *Adv. Mater.* **36**, 2310428 (2024).

29 Shang, Y. et al. Unconventional Mn vacancies in Mn-Fe Prussian blue analogs: suppressing Jahn-Teller distortion for ultrastable sodium storage. *Chem* **6**, 1804-1818 (2020).

Regarding the calculations of PBA chemical formula, we have provided a comprehensive discussion in **Response 1**. It must be explicitly reiterated that in determining PBA stoichiometric calculations, EPR serves solely to qualitatively verify the existence of $[\text{Fe}(\text{CN})_6]^{4-}$ vacancies.

3. Regarding the theoretical capacity calculations, we thank the authors for the detailed explanation. The calculation of the theoretical capacity is almost always conducted via making

assumptions about the system, so it was beneficial to understand how authors reached these values.

In the response letter, authors stated: “However, not all MHS enable reversible +2/+3 redox. Ni and Cu ions preferentially maintain +2 oxidation states during cycling (Adv. Mater. 2024, 2405458), limiting Na⁺ storage to 1 mol via FeLS redox alone, with theoretical capacities of 84.64 and 83.36 mAh g⁻¹ for Ni-based and Cu-based PBAs, respectively”.

The transition metal redox activity in PBAs depend on many aspects, and while there are quite a few examples of Ni and Cu not being redox active, the opposite can also be found. Such as:

<https://doi.org/10.1002/adma.202419446> (Ni),

<https://doi.org/10.1021/acs.jpcc.8b03429> (Cu),

<https://doi.org/10.1016/j.ensm.2020.08.008> (Cu),...

We are not stating that Ni and/or Cu are necessarily electrochemically active in the compounds described in current manuscript, but to state the opposite, the proof must be demonstrated (such as XAS, for example).

Response 3: We fully acknowledge the reviewer’s concerns about the electrochemical activity of Ni and Cu in PBAs and appreciate the careful citation of studies (marked as Reference 1, Reference 2 and Reference 3, respectively) demonstrating partial redox activity of these metals. Indeed, as noted by the reviewer, the electrochemical activity of transition metals is influenced by multifaceted factors including: (1) Charge carrier species (such as Li⁺, Na⁺ and NH₄⁺), (2) Charge/discharge voltage windows, (3) Microstructural morphology, (4) Electronic structure of PBAs and etc. Consequently, characterizing valence evolution of Ni/Cu during cycling via XAS or XPS proves essential for determining electrochemical activity origins in PBAs. Following the reviewer’s guidance, we have examined relevant studies in similar systems with our work. Prof. Shulei Chou’s group employed in-situ XAS to probe Mn, Fe, Co, Ni, and Cu activity shown in Cited figure 1, while Ni remained electrochemically inert (*Angew. Chem. Int. Ed.* 2023, 62, e202215865). As shown in Cited figure 2, Prof. Yan Yu’s team utilized ex-situ XPS to demonstrate complete electrochemical inertness of Ni and partial activity of Cu (*Adv. Mater.* 2024, 2405458). According to the citation list by reviewer (Reference 3), Prof. Yunhui Huang’s group electrochemically demonstrated ≈0.5 mol Cu redox activity in CuHCF-P with specific morphology, while

simultaneously affirming that conventional CuHCF-C exhibit solely 1 mol Na⁺ transfer by active Fe^{LS} and Cu is electrochemical inertness (Cited figure 3).

Figure Redacted

Cited figure 1 In situ XANES spectra at the transition metal K-edges of HE-HCF (*Angew. Chem. Int. Ed.* 2023, 62, e202215865)

Figure Redacted

Cited figure 2 c–f) Ex situ XPS analysis of (c) Mn, (d) Fe, (e) Cu, and (f) Ni at initial, C 3.3, C 3.7, C 4.2, D 3.6, D 3.2, and D 2 V states for the MnCuNi-PBA electrode. (*Adv. Mater.* 2024, 2405458)

Figure Redacted

Cited figure 3 The XPS Cu 2p spectra of (a) CuHCF-P and (c) CuHCF-C, and the XPS Fe 2p spectra of (b) CuHCF-P and (d) CuHCF-C at different redox states including as-prepared samples. The electrodes charged to 4.2 V and discharged to 2.0 V. (<https://doi.org/10.1016/j.ensm.2020.08.008>)

Based on the mentioned researches under similar condition with our work, we conclude that within non-aqueous sodium-ion batteries, **Ni in PBAs exhibits definitive electrochemical inertness, while the activity of Cu remains debated.** At the instance of reviewer's recommendation, **we supplemented Cu K-edge XANES analysis for M5-PBA as followed in supplement figure 1.**

Supplement figure 1 Cu K-edge XANES analysis for M5-PBA

Results confirm valence changes in Cu during charging and discharging process, but the precise quantification of electron transfer numbers proves unattainable. **However, in this work, we**

primarily investigated the effects of cyanide coordination electronic structure modulation on structural stability and capacity, while the origin of electrochemical activity in PBAs was not the central research focus. As rightly suggested by the reviewer, future studies will emphasize mechanistic investigations of how individual transition metal ions influence PBA activity through combined in situ characterization approaches. We sincerely appreciate the reviewer’s valuable suggestions, and we have implemented the following manuscript revisions to ensure scholarly rigor:

“The resulting uniform electronic structure enhances the reactivity of the transition metals, which enables the full activation of electrochemical activity for transition metals.” (Page 2 Line 21-23)

“Consequently, the optimized PBA achieves the simultaneous improvement of capacity and cycling lifetime, delivering the discharge capacity of $142.4 \text{ mAh}\cdot\text{g}^{-1}$ at 0.1 C and retaining 91.7% of the reversible capacity after 1000 cycles at 5 C.” (Page 4 Line 81-84)

“The initial discharge capacities of M2-PBA, M4-PBA and M5-PBA are 142.7, 142.4 and 109.5 $\text{mAh}\cdot\text{g}^{-1}$, respectively.” (Page 11 Line 239-240)

Fig. S24 Radar chart of electrochemical performance comparison for PBAs in this work. (Supporting Information)

Reviewer #2 (Remarks to the Author):

We sincerely thank Reviewer #2 and editors for their constructive feedback, which has significantly strengthened the rigor and clarity of our manuscript.